# Analytical solutions for the advective-diffusive ice column in the presence of strain heating

Daniel Moreno-Parada[1,2], Alexander Robinson[3], Marisa Montoya[1,2], and Jorge Alvarez-Solas[1,2]

[1]Departamento de Física de la Tierra y Astrofísica, Universidad Complutense de Madrid, Facultad de Ciencias Físicas, 28040 Madrid, Spain
[2]Instituto de Geociencias, Consejo Superior de Investigaciones Científícas-Universidad Complutense de Madrid, 28040 Madrid, Spain
[3]Alfred Wegener Institute, Helmholtz Centre for Polar and Marine Research, Potsdam, Germany

**Correspondence:** Daniel Moreno-Parada (danielm@ucm.es)

**Abstract.** A thorough understanding of ice thermodynamics is essential for an accurate description of glaciers, ice sheets and ice shelves. Yet there exists a significant gap in our theoretical knowledge of the time-dependent behaviour of ice temperatures due to the inevitable compromise between mathematical tractability and the accurate description of physical phenomena. In order to bridge this shortfall, we have analytically solved the 1D time-dependent advective-diffusive heat problem including additional terms due to strain heating and depth-integrated horizontal advection. Newton's Law of Cooling is applied as a Robin-type top boundary condition to consider potential non-equilibrium temperature states across the ice-air interface. The solution is expressed in terms of confluent hypergeometric functions following a separation of variables approach. Non-dimensionalisation reduces the parameter space to four numbers that fully determine the shape of the solution at equilibrium: surface insulation, effective geothermal heat flow, the Peclét number and the Brinkman number. The initial temperature distribution exponentially converges to the stationary solution. Transient decay timescales are only dependent on the Peclét number and the surface insulation, so that higher advection rates and lower insulating values imply shorter equilibration timescales, respectively. On the contrary, equilibrium temperature profiles are mostly independent of the surface insulation parameter. We have extended our study to a broader range of vertical velocities by using a general power-law dependence on depth, unlike prior studies limited to linear and quadratic velocity profiles. Lastly, we present a suite of benchmark experiments to test numerical solvers. Four experiments of gradually increasing complexity capture the main physical processes for heat propagation. Analytical solutions are then compared to their numerical counterparts, upon discretisation over unevenly-spaced coordinate systems. We find that a symmetric scheme for the advective term and a three-point asymmetric scheme for the basal boundary condition best match our analytical solutions. A further convergence study shows that $n \geq 15$ vertical points are sufficient to accurately reproduce the temperature profile. The solutions presented herein are general and fully applicable to any problem with an equivalent set of boundary conditions and any given initial temperature distribution.

# 1 Introduction

The study of ice thermodynamics is of crucial importance for understanding the behaviour of glaciers, ice sheets and ice shelves. Ice temperatures control both the rate at which ice deforms (LeB. Hooke, 1981) and the occurrence of sliding when the base reaches melting (Iken and Bindschadler, 1986). Precisely, ice softens one order of magnitude as temperature increases from -10ºC to the melting point (e.g., Greve and Blatter, 2009; Cuffey and Paterson, 2010) and velocities can increase by 2-3 orders of magnitude over a temperate base that yields rapid sliding. However, accurate ice temperature estimations are challenging, since heat transfer balance is the result of a complex interplay between advection, diffusion and various heat sources. Only an accurate representation of these processes will allow for a robust assessment of ice flow, mass balance and overall stability. In this context, the development of analytical solutions for ice thermodynamics can provide deeper comprehension of the fundamental physics of ice, as they are intuitively interpretable, reveal hidden symmetries and further serve as a verification tool or benchmark for numerical models.

Robin (1955) and Lliboutry (1963) first laid the groundwork for understanding ice-column thermodynamics in the presence of vertical advection and diffusion by providing analytical solutions for stationary scenarios. These seminal works offered valuable insights into the steady-state behaviour of ice columns subject to advective-diffusive processes. Nevertheless, they did not consider the time-dependent evolution of ice temperatures. Hence, their applicability was limited to situations involving steady-state ice flow and fixed environmental conditions.

Steady-state ice temperature distribution studies also provide analytical solutions in bounded spatial domains, but fall short if the transient nature of the solution is to be captured. This is the case of the studies on the shear heating margins of West Antarctic ice streams (e.g., Perol and Rice, 2011, 2015) for which a more refined one-dimensional thermal model was produced, first introduced by Zotikov (1986). Meyer and Minchew (2018) later solved a similar advective-diffusive problem under stationary conditions accounting for a constant strain-heating rate and further neglecting lateral (horizontal) advection after a scaling analysis.

More recently, Rezvanbehbahani et al. (2019) proposed an improved temperature solution that considers a power-law vertical velocity profile derived from the Shallow Ice Approximation. The authors showed the importance of the strain heating term and demonstrated that including it as an additional basal heat source yields good results for the interior regions of an ice sheet. Nevertheless, horizontal advection is absent in their analytical solutions and a further comparison with numerical solutions reveals that their analytical results are only applicable to slowly moving regions (mostly below 20 m/yr). As with prior studies, steady-state conditions are also assumed and thus no information about the time evolution of ice temperatures can be obtained.

Despite these simplifications, heat transfer is well-known to be a three-dimensional process with a higher level of complexity that encompasses several mechanisms such as horizontal and vertical advection, the potential presence of liquid water within the ice, a varying ice thickness, internal heat deformation and frictional heat production among others (e.g., Greve and Blatter, 2009; Cuffey and Paterson, 2010). Full numerical models are therefore also essential if a simultaneous consideration of such mechanisms needs to be achieved (Winkelmann et al., 2011; Pattyn, 2017).

However, numerical models require caution as their accuracy and consistency must be previously assessed. Intercomparison projects are thus fundamental since they can provide consensus in benchmark experiments that further serve as a reference solution for validation. In this context, analytical descriptions are extremely useful as they provide a control irrespective of the resolution or discretisation schemes. For instance, Huybrechts and Payne (1996) already noted the lack of analytical temperature solutions for more realistic vertical velocity profiles. Previously obtained solutions relied on strong assumptions regarding the particular vertical velocity profile (linear profile, Robin 1955; quadratic, Raymond 1983) and therefore an independent analytical description of the temperatures was not available.

Traditional approaches both from numerical and analytical perspectives assume the simplest heat-flux boundary condition at the ice surface: the imposition of the air temperature at the uppermost ice layer. Knowing that glacial ice forms through snow densification, this imposition appears to be an oversimplification, given that thermal conductivity increases with density (e.g., Sturm et al., 2002; Calonne et al., 2011, 2019). Therefore, in view of the surface fraction of the Greenland and Antarctic Ice Sheets covered by a firn layer (90% and ∼100%, respectively, Medley et al., 2022; Noël et al., 2022), a more sophisticated description of the energy balance between the ice and the atmosphere may be beneficial. Already noted by Carslaw and Jaeger (1988), prescribing a fixed temperature is in fact a limit case of a broader set of boundary conditions known as 'linear heat transfer' or 'Newton's law of cooling' that accounts for a more realistic heat flux across the interface given by the temperature difference between the two media.

The 1D advective-diffusive equation has been thoroughly studied in a wide range of fields, particularly in dispersion problems. In early studies, the basic approach was to reduce the advection-diffusion equation to a purely diffusive problem by eliminating the advective terms. This was achieved via a moving coordinate system (e.g., Ogata and Banks, 1961; Harleman and Rumer, 1963; Bear, 1975; Guvanasen and Volker, 1983; Aral and Liao, 1996; Marshall et al., 1996) or through the introduction of another dependent variable (e.g., Banks and Ali, 1964; Ogata, 1970; Lai and Jurinak, 1971; Marino, 1974; Al-Niami and Rushton, 1977). To solve the equations, quite diverse mathematical methods are employed, such as the Laplace transformation (McLachlan, 2014), the Hankel transform (Debnath and Bhatta, 2014), the Aris moment method (Merks et al., 2002), Green's function (Evans, 2010) or superposition approaches (Lie and Scheffers, 1893) among others. More recent studies (e.g., Selvadurai, 2004) provide time-dependent analytical solutions for which Darcy flow is applicable, yet they lack an appropriate set of boundary conditions given the infinite length of the domain.

Ice temperatures are not only critical to understand the dynamics and an ice body's evolution in time, but also in ice-sheet initialisation of numerical models. Poorly known parameter fields such as the ice temperature are estimated minimising the mismatch between observations and model output variables. Traditional approaches compute a steady-state temperature field, incorrectly assuming that the ice is at thermal equilibrium (e.g., Morlighem et al., 2010, 2011; Pralong and Gudmundsson, 2011; Perego et al., 2014). This issue can be mitigated via transient optimisation approaches that incorporate available data that accounts for the transient nature of observations and the model dynamics (e.g., Goldberg et al., 2015), though this method is significantly more expensive. Nonetheless, time integration with transient optimisation that includes all relevant model processes is not feasible for high-resolution, large-scale ice sheet models. As a result, a time-dependent description of ice temperatures would strongly reduce the computational demands in modelling exercises.

It is thus clear that a time-dependent analytical description would be valuable, in spite of the inevitable compromise of designing a model that is both mathematically solvable and accurate. It is thus of utmost importance to carefully navigate this trade-off, deciding the appropriate level of analytical tractability and physical realism based on the specific goals of any given study. Attaining the right balance allows for meaningful insights, while avoiding excessive computational demands or oversimplification that may hinder accurate representation and understanding of the real-world system. Despite all the effort in previous works, there is still a clear gap in the understanding of the analytical nature of time-dependent ice temperatures. As a result, there are no available benchmark experiments to test numerical solvers extensively employed in ice-sheet models.

The current study presents an analytical formulation of the transient ice temperature equation and provides useful insight in two ways. First, allowing for a simplified way of studying the physics of heat transfer in ice (as demonstrated by an equilibrium timescales analysis) and secondly, by providing a way of benchmarking numerical solvers for heat transfer. Our approach accounts for the temporal evolution of the temperature profile rather than assuming an equilibrated state, thus taking a step towards a more accurate representation of the ice thermal behaviour. The formulation of the problem is given in Section 2; the approach followed in this work is presented in Section 3; analytical solutions are shown in Section 4; results are presented in Section 5, benchmark experiments are detailed in Section 6, results are discussed in Section 7 and concluding remarks are given in Section 8.

## 2   Advective-diffusive ice column

We consider a one-dimensional ice column with diffusive heat transport, vertical advection, strain heat and depth-integrated horizontal advection. Our domain is defined as the interval $z \in [0, L] \equiv \mathcal{L}$ and we further impose a Robin-type boundary condition at the top of the column, $z = L$ (Fig. 1). The aim of this section is to provide a rigorous mathematical formulation of the physical mechanisms involved in the heat problem necessary to obtain an exact solution of the ice temperature $\theta(z, t)$.

In the simplest physical scenario, the ice surface temperature is set to the air temperature value $\theta(L, t) = T_{\text{air}}$. However, surface temperatures are in fact the result of the energy balance between the ice and the atmosphere. To address this limitation, we refine the surface boundary condition by allowing for a potential deviation from the air temperature, accounting for the thermal insulating effect in the uppermost region of the ice column. This insulation effect is a direct consequence of the reduction in ice density towards the surface (e.g., Stevens et al., 2020) and, as a result, the reduced ice thermal conductivity (Sturm et al., 2002; Calonne et al., 2011, 2019). This surface energy balance falls within the so-called linear heat-transfer boundary conditions or 'Newton's law of Cooling' (Carslaw and Jaeger, 1988, Chapter § 1.9). Briefly, Newton's law of cooling states that the heat flux across the interface is proportional to the temperature difference between the surface and the surrounding medium. It is applicable to a large variety of conditions such as a body cooling by forced convection (i.e., a fluid forced rapidly past the surface of a solid) or a a thin surface layer of a poor conductor (such as a low density firn or snow layer above the glacial ice). Moreover, Newton's law of cooling captures the two simpler boundary conditions as limit cases: (1) prescribed surface temperature and (2) no heat flux across an interface.

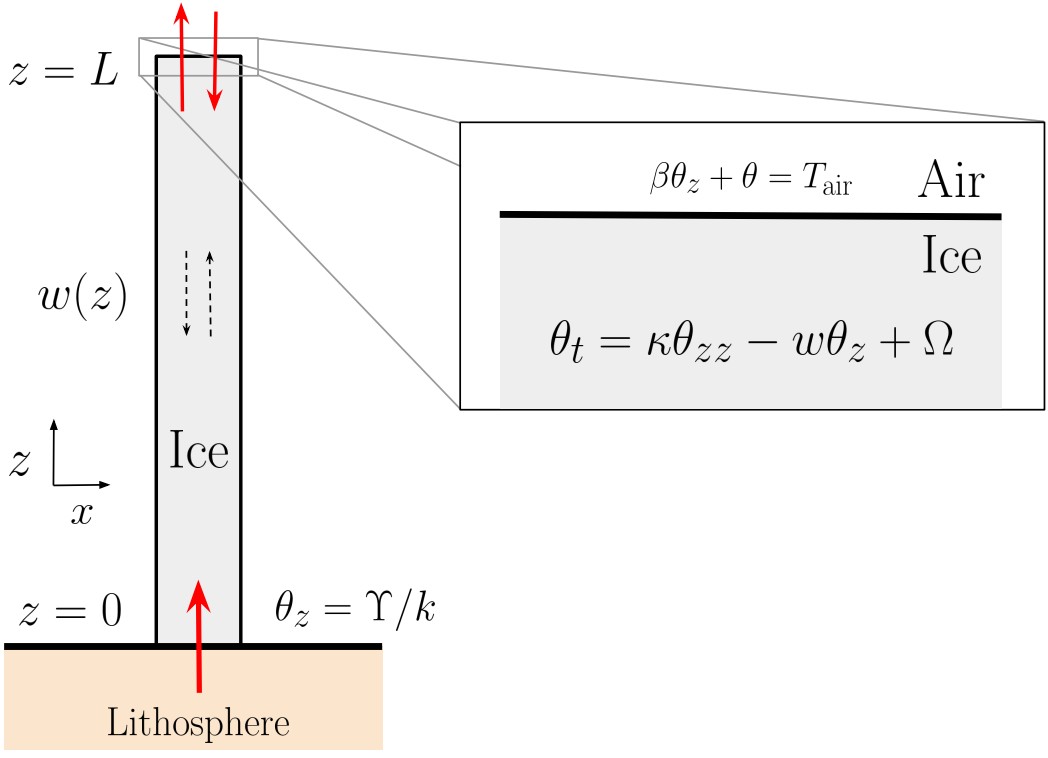

**Figure 1.** Schematic view of the one-dimensional ice column with vertical advection $w(z)$ and inhomogeneous term $\Omega$ (here, we independently consider both strain heating and depth-integrated horizontal advection). Temperature evolution is dictated by the heat equation and an appropriate set of initial and boundary conditions. Subscripts denote partial differentiation. At the top, both the ice temperature and the vertical gradient can vary in time, thus allowing for non-equilibrium thermal states across the ice-air interface. At the base, the vertical gradient is fixed to the value given by the combined contribution of geothermal heat flow and potential basal frictional heat $\theta_z = -\Upsilon/k$. Note that our formulation is one-dimensional so that the $x$-axis is solely introduced for visualization.

This refinement enables a more accurate representation of the surface heat transfer dynamics and contributes to a comprehensive understanding of the energy balance within the ice column. In this description, both the surface ice temperature $\theta(L,t)$ and its vertical gradient $\theta_z(L,t)$ can vary in time:

$$\beta\theta_z + \theta = T_{\text{air}}, \quad z = L, \ t > 0, \tag{1}$$

where italic subscripts denote partial differentiation and $\beta$ is a parameter with length dimensions that modulates the permissible deviation between ice and air temperatures, often referred to as the surface thermal resistance (per unit area). We physically interpret $\beta$ as the thermal insulation of the ice-air interface. In other words, $\beta$ is a length-scale over which the ice column feels the air temperature. A zero value corresponds to an ideal conductor $\theta(L,t) = T_{\text{air}}$, whereas $\beta \to \infty$ represents a perfect thermal insulator characterized by a null heat exchange across the interface. In the limit case $\beta = 0$, the interface ice-air is always at

thermal equilibrium (i.e., $\theta = T_{\text{air}}$). For $\beta \neq 0$, we allow for a heat exchange across the ice surface driven by the temperature difference between the two media. The thermal equilibrium is only reached if the ice surface and the atmosphere temperatures are identical. In such conditions, the heat flux across the interface is null and the vertical gradient at the top the ice column vanishes regardless of the value of $\beta$.

Considering diffusive heat transport, vertical advection, and a potential heat source, the ice temperature $\theta(z, t)$ satisfies an initial value problem given by the heat equation:

$$
\begin{cases}
\theta_t = \kappa \theta_{zz} - w\theta_z + \Omega, & z \in \mathcal{L},\ t > 0, \\
\theta = \theta_0(z), & z \in \mathcal{L},\ t = 0, \\
\theta_z = -\Upsilon/k, & z = 0,\ t > 0, \\
\beta\theta_z + \theta = T_{\text{air}}, & z = L,\ t > 0,
\end{cases}
\tag{2}
$$

where the heat source $\Omega$ is an inhomogeneous term that captures strain heat and horizontal advection, $\Upsilon = G + Q$ is the combined contribution of geothermal heat flow $G$ and potential basal frictional heat $Q$, $k$ is the ice conductivity and $\kappa$ is the ice diffusivity, both assumed to be constant. We further consider a $z$-dependent vertical velocity component given by $w(z)$.

In order to solve this problem, we must first provide the particular form of the vertical velocity term. As in Clarke et al. (1977) and Zotikov (1986), we first assume a linear variation of $w(z)$ with depth:

$$
w(z) = w_0 \frac{z}{L},
\tag{3}
$$

where $w_0$ is the vertical velocity at the ice surface $z = L$.

Standard values for $w_0$ usually read from $-0.1$ to $-0.3$ m/yr (Glovinetto and Zwally, 2000; Spikes et al., 2004). Positive values of $w_0$ imply an upward movement of ice and are physically plausible, though quite rare. Dahl-Jensen (1989) calculated steady temperature distributions (Fig. 5 therein) and found that profiles near the terminus position resemble those predicted for an ablation zone ($w_0 > 0$). Solutions herein presented are applicable to both positive and negative values of $w_0$, though we will focus on the downward movement of ice (i.e., $w_0 < 0$). The linear dependency is widely used in the literature (e.g., Joughin et al., 2002, 2004; Suckale et al., 2014). Nonetheless, we will also explore a more general power-law relationship that better describes vertical velocities modeled by Glen's flow law (see Appendix C).

     The inhomogeneous term $\Omega$ can encompass a number of heat sources and sinks. Here we focus on strain heating $\mathcal{S}$ and horizontal advection $\mathcal{H}$, so that $\Omega = \mathcal{S} + \mathcal{H}$. In general, the strain heating term can be expressed as $\mathcal{S} = \sigma_{ij}\dot{\epsilon}_{ij}$, where $\sigma_{ij}$ is the Cauchy stress tensor and $\dot{\epsilon}_{ij}$ is the strain rate tensor (expressed in index notation). Upon application of Glen's law, the rate of strain heating is solely a function of the second invariant of the strain rate tensor:

$$
\mathcal{S} = \sigma_{ij}\dot{\epsilon}_{ij} = 2A^{-1/n}\, \dot{\epsilon}_e^{\,(n+1)/n},
\tag{4}
$$

where $\dot{\epsilon}_e = (\dot{\epsilon}_{ij}\dot{\epsilon}_{ij}/2)^{1/2}$ is the second invariant of the strain rate tensor and summation is implied over repeated indexes (Einstein notation). This formulation does not impose any conditions on the strain rate regime (i.e., the dominant components)

and only assumes $\dot{\epsilon}$ to be constant in depth. This requirement ensures the analytical tractability of the solution while including a potential strain contribution throughout the ice column.

The horizontal advection term $\mathcal{H}$ can imply a heat source or a sink, depending on the sign of the horizontal temperature gradient along a particular direction. We herein consider such a contribution by defining a depth-averaged lateral advection term (Meyer and Minchew, 2018):

$$\mathcal{H} = \frac{1}{L} \int\limits_0^L (\mathbf{u} \cdot \hat{\mathbf{n}}) \theta_{\hat{\mathbf{n}}} dz, \tag{5}$$

where $\mathbf{u}$ is the horizontal velocity vector, $\hat{\mathbf{n}}$ is the normal vector along an arbitrary direction contained in the horizontal plane and $\theta_{\hat{\mathbf{n}}} = \partial\theta/\partial\hat{\mathbf{n}}$ denotes the directional derivative along $\hat{\mathbf{n}}$.

This assumptions allow us to include a potential strain heat source $\mathcal{S}$ and a horizontal advection of heat term $\mathcal{H}$ while keeping the analytical tractability of Eq. 2. The limitations of these simplifications are discussed in Section 7.

## 3 Analytical solution

We next outline our analytical approach. We first non-dimensionalise our problem and exploit the linearity of the differential operator by further decomposing the solution as a sum of stationary and transient components to deal with the inhomogeneity. Lastly, we apply separation of variables to obtain a solution of the time-dependent problem and impose the corresponding initial and boundary conditions. Derivation details are elaborated in Appendix A.

It is natural to non-dimensionalise our problem by defining the following variables:

$$\xi = \frac{z}{L}, \ \tau = \frac{\kappa}{L^2}t, \ \theta = \frac{T}{T_{\mathrm{air}}}, \ \tilde{w} = \frac{L}{\kappa}w, \ \tilde{\beta} = \frac{\beta}{L}, \ \tilde{\Omega} = \frac{L^2}{\kappa T_{\mathrm{air}}}\Omega \tag{6}$$

over the domain $\tilde{\mathcal{L}} = [0, 1]$. Tildes are hereinafter dropped to lighten the notation.

Hence, we can express our Problem 2 as:

$$
\begin{cases}
\theta_\tau = \theta_{\xi\xi} - \mathrm{Pe}\,\xi\theta_\xi + \Omega, & \xi \in \mathcal{L}, \ \tau > 0, \\
\theta = \theta_0(\xi), & \xi \in \mathcal{L}, \ \tau = 0, \\
\theta_\xi = \gamma, & \xi = 0, \ \tau > 0, \\
\beta\theta_\xi + \theta = 1, & \xi = 1, \ \tau > 0,
\end{cases}
\tag{7}
$$

where $\gamma = -T_{\mathrm{air}}\Upsilon/(kL)$, $w = \mathrm{Pe}\,\xi$ and $\theta_0(\xi)$ are the non-dimensional geothermal heat flow, vertical velocity and initial profile respectively. The vertical velocity is thereby conveniently expressed in terms of the Peclét number $\mathrm{Pe} = w_0 L/\kappa$ (i.e., the ratio of advective to diffusive heat transport). The non-dimensional strain heat source term $\mathcal{S}$ can be identified with the Brinkman number Br, which represents the ratio of deformation heating to thermal conduction (see Table 1). The non-dimensional number $\gamma$ is the combined contribution of geothermal heat flow and potential basal frictional heat, normalised by the vertical

**Table 1.** Non-dimensional definitions and characteristic ranges. Summation is implied over repeated indices. Pe and Br are the Peclét and Brinkman numbers, respectively. $\Lambda$ is the normalised horizontal advection, $\beta$ is the surface insulation parameter and $\gamma$ is the dimensionless combined contribution of geothermal heat flow and basal frictional heat. Physical magnitudes employed to obtain these ranges are give in Table 2.

| Symbol | Definition | Characteristic range |
|:---:|:---:|:---:|
| Pe | $\dfrac{L}{\kappa}w_0$ | $0.0 - 30.0$ |
| Br | $\dfrac{L^2}{\kappa T_{\mathrm{air}}}\sigma_{ij}\dot{\epsilon}_{ij}$ | $0.0 - 20.0$ |
| $\Lambda$ | $\dfrac{L^2}{\kappa T_{\mathrm{air}}}\int_0^1 (\mathbf{u}\cdot\hat{\mathbf{n}})\,\theta_{\hat{\mathbf{n}}}\,d\xi$ | $0.0 - 10.0$ |
| $\gamma$ | $-\dfrac{T_{\mathrm{air}}}{kL}\Upsilon$ | $0.1 - 5.0$ |
| $\beta$ | $\dfrac{\beta}{L}$ | $0.0 - 1.0$ |

temperature gradient that would exists for a column thickness $L$ and temperature $T_{\mathrm{air}}$. It provides the relative strength of the basal inflow of heat compared to the ice-column extent and the air temperature.

The dimensionless problem clearly shows that five numbers completely determine the shape of the stationary solution: $\gamma$, $\beta$, Pe, $\Lambda$ and Br. Their particular impact on the temperature distributions is discussed below.

**Table 2.** Physical parameters values employed to determine the non-dimensional range shown in Table 1.

| Parameter | Definition | Explored range | Units | Reference |
|:---:|:---:|:---:|:---:|:---:|
| $L$ | Ice thickness | 1 - 3 | km | Greve and Blatter (2009) |
| $T_{\mathrm{air}}$ | Air temperature | 223.15 - 263.15 | K | Cuffey and Paterson (2010) |
| $\kappa$ | Thermal diffusivity | $1.4 \cdot 10^{-6}$ | $\mathrm{m^2\ s^{-1}}$ | Ritz (1987) |
| $k$ | Thermal conductivity | 2.0 | $\mathrm{W\ m^{-1}\ K^{-1}}$ | Ritz (1987) |
| $\beta$ | Surface insulation | 0 - 3 | km | N/A |
| $G$ | Geothermal heat flow | 0.01 - 0.05 | $\mathrm{W\ m^{-2}}$ | Hooke (2005) |
| $Q$ | Frictional heat | 0 - 0.5 | $\mathrm{W\ m^{-2}}$ | Karlsson et al. (2021) |
| $u$ | Horizontal velocity | 0 - 500 | $\mathrm{m\ yr^{-1}}$ | Greve and Blatter (2009) |
| $\theta_{\hat{n}}$ | Horizontal temperature gradient | 0 - 1 | $\mathrm{K\ km^{-1}}$ | Dahl-Jensen (1989); Funk et al. (1994) |
| $\dot{\epsilon}_e$ | Effective strain rate | 0 - 0.1 | $\mathrm{yr^{-1}}$ | Meyer and Minchew (2018) |
| $A$ | Ice rate factor | $10^{-25}$ - $10^{-24}$ | $\mathrm{Pa^{-3}s^{-1}}$ | Cuffey and Paterson (2010) |

Given that Eq. 7 is inhomogeneous, we will decompose the solution as a sum of a transient $\mu(\xi, \tau)$ and a stationary $\vartheta(\xi)$ components, so that $\theta(\xi, \tau) = \mu(\xi, \tau) + \vartheta(\xi)$. As a result, the transient and stationary problems are subject to homogeneous and inhomogeneous boundary conditions, respectively:

$$
\begin{cases}
\mu_\tau = \mu_{\xi\xi} - w\mu_\xi, & \xi \in \mathcal{L}, \ \tau > 0, \\
\mu = \mu_0, & \xi \in \mathcal{L}, \ \tau = 0, \\
\mu_\xi = 0, & \xi = 0, \ \tau > 0, \\
\beta\mu_\xi + \mu = 0, & \xi = 1, \ \tau > 0,
\end{cases}
\tag{8}
$$

and

$$
\begin{cases}
\Omega = \vartheta_{\xi\xi} - w\vartheta_\xi, & \xi \in \mathcal{L}, \\
\vartheta_\xi = \gamma, & \xi = 0, \\
\beta\vartheta_\xi + \vartheta = 1, & \xi = 1,
\end{cases}
\tag{9}
$$

where $\mu_0 = \theta_0(\xi) - \vartheta(\xi)$ is the initial profile of the transitory solution.

The solution to the stationary component (Eq. 9) already differs from previous analytical works as Robin (1955) and Lliboutry (1963). First, they considered a homogeneous version of the problem (i.e., $\Omega = 0$) so that potential strain heating or horizontal advective contributions are neglected. Moreover, they simplified the top boundary condition since they imposed a prescribed constant temperature value at $\xi = 1$ (see also Clarke et al., 1977). However, our refinements still allow for analytically tractability and thus the stationary solution is (see Appendix B for derivation details):

$$
\vartheta(\xi) = \Omega \frac{\xi^2}{2} \, {}_2F_2\left(1, 1; \frac{3}{2}, 2; -\zeta\right) + A \, \mathrm{erf}\,[a\xi] + B
\tag{10}
$$

where ${}_2F_2(a_1, a_2; b_1, b_2, x)$ is the generalised hypergeometric function, $\zeta = (a\xi)^2$, $a = (w_0/2)^{1/2}$, $A = -\gamma (\pi/(4a))^{1/2}$ and $B = 1 - A\left(2a\pi^{-1}\beta e^{-a^2} + \mathrm{erf}\,[a]\right)$. Note that if the inhomogeneous term is zero (i.e., $\Omega = 0$), the stationary temperature profile reduces to the well-known error function previously obtained by Robin (1955) and Lliboutry (1963). Even so, the temperature distribution would still differ as the boundary condition considered herein reflects a potential surface thermal insulation unlike prior studies.

We now take a step further and allow for time evolution by solving Eq. 8 and building our solution as the sum of both contributions. Namely, the general solution of the transient problem $\mu(\xi, \tau)$ is (see Appendix A for derivation details):

$$
\mu(\xi, \tau) = \sum_{n=0}^{\infty} \left[A_n \Phi\left(\alpha_n; \delta; \zeta\right) + B_n \Psi(\alpha_n; \delta; \zeta)\right] e^{-\lambda_n \tau}
\tag{11}
$$

where $\Phi\left(\alpha; \delta; \zeta\right)$ and $\Psi\left(\alpha; \delta; \zeta\right)$ are the Kummer (Kummer, 1836) and Tricomi confluent hypergeometric functions respectively (also known as confluent hypergeometric functions of the first and second kind). $\alpha_n = -\lambda_n/(2w_0)$ and $\delta = 1/2$. As the solution must be bounded at the origin, we set $B_n = 0$.

The full solution $\theta(\xi, \tau) = \vartheta(\xi) + \mu(\xi, \tau)$ thus reads:

$$\theta(\xi, \tau) = \Omega \frac{\xi^2}{2} \, {}_2F_2\left(1, 1; \frac{3}{2}, 2; -\zeta\right) + A \, \mathrm{erf}\left[a\xi\right] + B + \sum_{n=0}^{\infty} A_n \Phi\left(\alpha_n; \delta; \zeta\right) e^{-\lambda_n \tau} \tag{12}$$

where the coefficients $A_n$ are obtained from the initial temperature profile (Eq. A.13 in Appendix A).

## 4 Stationary solutions

Before displaying the results of the full time-dependent problem, it is worth describing the temperature solutions at equilibrium.

Figure 2 shows our steady-state solutions as vertical profiles for a subset of the permutations of the non-dimensional numbers Pe, Br, $\gamma$, $\Lambda$ and $\beta$. It is illustrative to compare the shape of our temperature solutions with Clarke et al. (1977) (Fig. 1 therein). We must stress that a one-to-one comparison is not readily possible since they imposed a simpler top boundary condition in which the ice surface temperature is fixed to a given value, though the exact same solutions can be simply obtained by setting $\beta = 0$ in our case (see Eq. 1).

The non-dimensionalization of our analytical model provides simplicity and further reduces the parameter dimensionality of the solutions to solely five numbers, each corresponding to one column in Fig. 2. The Peclét number produces significant changes in the equilibrium solutions, as colder ice is advected from the uppermost part of the column, consequently cooling down the profile with increasing Pe values (Fig. 2a), in contrast to the well-known linear profile resulting for the purely diffusive case (i.e., Pe $\to 0$). The combined contribution of geothermal heat flow and friction heat dissipation $\gamma$ also yields large temperature amplitudes within the explored range. Nevertheless, the impact is clearly limited to the lower half of the column, thus leaving the upper regions nearly unperturbed as shown in Fig. 2c. Likewise, for the surface insulation parameter $\beta$ in the presence of downwards advection (Pe $= 7$), the entire temperature profile is left unchanged despite varying values of $\beta$ (Fig. 2b). This can be understood as the heat exchange at the ice-air interface is not relevant for strong downward transport of colder ice, which is a far more effective heat transport compared to dissipation. Unlike $\gamma$, the strain heat dissipation Br influences the upper region of the ice temperature as its contribution is distributed throughout the column (Fig. 2d), rather than being a basal heat source. Even so, the impact is most notable near the base given that the temperature therein can freely evolve so long as the geothermal heat flow condition is met (Eq. 2). Similarly, the vertically-averaged lateral heat advection $\Lambda$ also affects upper regions of the column (Fig. 2e). Here we have chosen positive $\Lambda$ values, implying advection of colder ice. As a result, for sufficiently large values of $\Lambda$, the temperature within the column can be lower than at the surface, reaching a local minimum therein and gradually increasing as the base is approached. For negative values of $\Lambda$, we would find temperature profiles as those obtained in Fig. 2d.

## 5 Full solutions

We now present the results of the full problem presented in Eq. 2 by including the time-dependent solution. This transient nature depends on the initial state of the system, although it exponentially converges to the steady state as the transient component

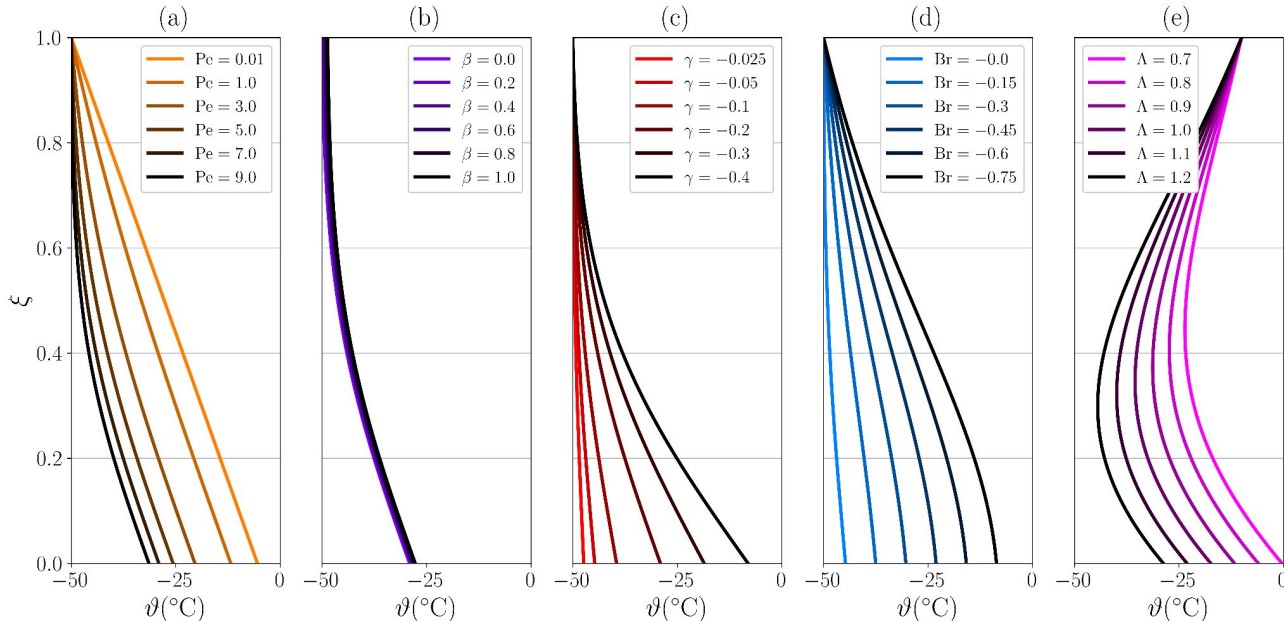

**Figure 2.** Stationary temperature profiles $\vartheta(\xi)$. Solutions are fully determined by five non-dimensional numbers: Pe, $\beta$, $\gamma$, Br and $\Lambda$, corresponding to each panel respectively. Default values are: Pe $= 5$, $\beta = 0$, $\gamma = -0.2$, Br $= 0$ and $\Lambda = 0$, except for panel (e), where $\gamma = -0.4$.

vanishes under the assumption of constant boundary conditions. We further overcome the arbitrariness on the initial temperature profile by directly calculating the eigenvalues of the problem and their corresponding decay times as an estimation of the time scale of our system in different physical scenarios.

To illustrate the full solutions, we show the explicit time evolution from an initial profile as it approaches the corresponding stationary solution (Fig 3). In this instance, we employ constant initial temperature profiles for simplicity, $\theta_0(\xi) = 0.5$ and $\theta_0(\xi) = 2.5$ in panels Fig 3a and b, respectively. With these particular choices, we ensure that the initial temperature profile is below and above the stationary solution for two strong advective scenarios: vertical and lateral. Fig. 3a shows how temperature both at the ice surface and most notably at the base start to increase for $\tau > 0$, while at the central region of the column

remains constant until heat propagates along the column. It is worth noting how the surface temperature gradually relaxes to the equilibrium profile since instead of imposing the air temperature, a more realistic heat exchange at the ice-air interface is considered via $\beta = 0.5$. On the contrary, Fig. 3b shows an instantaneous change at the surface by an oversimplified top boundary condition if $\beta = 0$ (i.e., a perfectly conductive ice-air interface). As a result, the cold air temperature rapidly propagates into the uppermost region of the ice column rapidly, whereas the geothermal heat flow contribution requires a longer time to propagate

from the base. On the contrary, the lower part of the domain increases its temperature notwithstanding the sudden decrease of the upper region. As the column evolves in time, the rate of change gradually diminishes and it approaches zero as the transient solution asymptotically reaches the temperature profile given by the stationary temperature profile $\vartheta(\xi) = \lim_{\tau \to \infty} \theta(\xi, \tau)$.

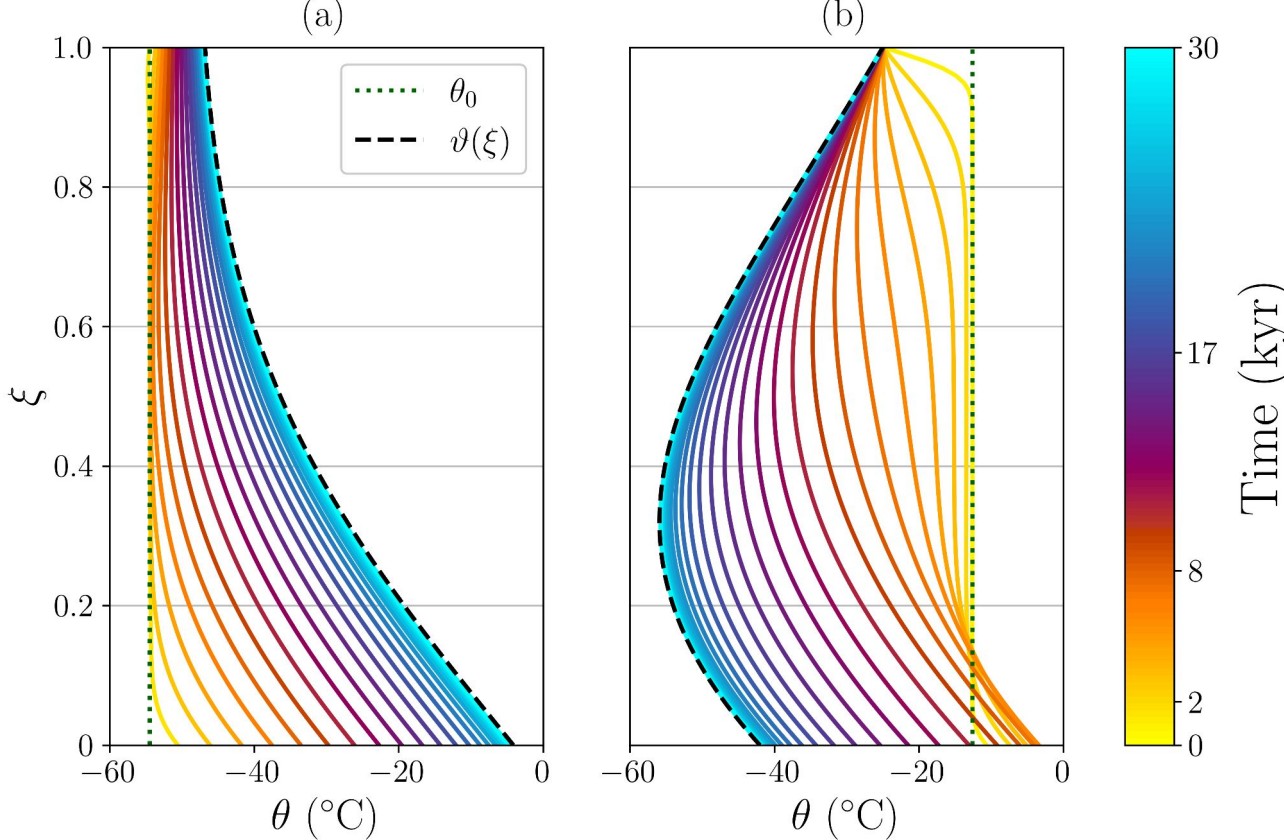

**Figure 3.** Time-dependent solution $\theta(\xi, t)$ given an initial temperature profile $\theta_0(\xi)$ (vertical dotted line). Dimensionless values: (a) $\beta = 0.5$, $\Lambda = 0$ and (b) $\beta = 0.0$, $\Lambda = 1.0$. Default values: $\mathrm{Pe} = 5.0$, $\gamma = -0.35$, $\mathrm{Br} = 0$. Black dashed lines represent the stationary solution $\vartheta(\xi)$. To ease visualization, the time variable is quadratically spaced as indicated in the colourbar.

To examine closely the transient nature of the solutions, we present the temperature evolution of a given initial profile for a certain range of the non-dimensional parameters (Fig. 4). This gives us information about the time-dependent effects of each parameter, unlike Fig. 2 that was restricted to equilibrium states. Addtionally, the continuous representation (i.e., colourbar in Fig. 4), as opposed to the discrete number of vertical profiles in Fig. 3 facilitates comparison among particular parameter choices.

The particular parameter values were selected so that we obtain four physically distinct scenarios: (a) high geothermal heat flow under a large advection regime, (b) high strain heat dissipation in a low vertical advection regime, (c) strong lateral advection of colder ice under surface insulating conditions and (d) weak geothermal heat flow under a low vertical advection regime. This setup allows us to separately determine the role played by each mechanism during the transient regime of the solution.

Figure 4a shows that the thermal equilibration begins by an increase of the basal temperature that gradually propagates upwards until the it is balanced by the downward advection ice from the colder surface. A similar transient behaviour is found if strain heat dissipation is additionally considered (Fig. 4b). Even though the geothermal heat flow is significantly smaller in this scenario, the heat travels further upwards as a result of a low vertical advection regime ($\mathrm{Pe} = 2$) combined with a source of strain heat throughout the column ($\mathrm{Br} = 6$). If we instead consider a scenario where heat is removed by lateral advection of colder ice $\Lambda = 6$ (Fig. 4c), we note two different timescales: the geothermal heat flow first warms the ice base, then the lateral removal of heat takes over with a consequent reduction of temperature in the entire column. Lastly, a low basal inflow of heat combined with a weak vertical advective regime (Fig. 4d) yields the smallest temperature gradients within the column.

We can also predict the behaviour of the transitory component directly from the eigenvalues of the problem. By calculating the inverse of the eigenvalues $\lambda_n^{-1}$, we obtain a magnitude that can be expressed with time dimensions and represents the decay time of each Fourier mode (Fig. 5a). Physically, this is the time required for the transient component to be reduced a factor $e^{-1}$ at any point and it further allows us to estimate the equilibration time from an arbitrary initial state. As we would expect, higher order modes have a shorter life time. Notably, the eigenvalue equation solely depends on $\mathrm{Pe}$ and the surface insulation parameter $\beta$ (Eq. A.8, Appendix A). This implies that the time to reach equilibrium exclusively depends on these two numbers. The remaining dimensionless parameter values yield the exact same equilibration time, despite playing a role in the particular form of the solution. In other words, the five dimensionless numbers shape the temperature profile, but only the vertical advection and the surface insulation parameter influence the exponential decay of the transitory component and therefore, the timescale to reach equilibrium from an arbitrary initial state (Fig. 5b). Particularly, scenarios with a high advective regime yield shorter equilibration times (Fig. 5b) $\sim$ 2-10 kyr, unlike highly insulating scenarios at the surface, characterized by long decay times ($\sim$ 25-40 kyr).

## 6 Benchmarks for numerical solvers

The analytical solutions obtained herein are valuable tools for testing numerical solvers. We thus propose a suite of benchmark experiments with gradually increasing complexity to test the representation of each physical process involved in ice temperature evolution (see Table 3).

First, we simply consider the well-known purely diffusive case (Exp-1). Then, vertical advection is additionally included (Exp-2). Lastly, strain heating (Exp-3) and the vertically-averaged horizontal advection (Exp-4) are considered. Given the analytical nature of our solutions, spatial and temporal resolutions can be set arbitrarily high as there are neither convergence nor stability constraints. This allows for a comparison against spatial and temporal resolutions found in numerical solvers. We must stress that the initial temperature profile and all other parameters can be set by the user to test the solution at any desired scenario. We also note that these are simply proposed benchmarks, but the solutions developed here can be used for any type of benchmark test that is desired and fits the limitations of the equations.

We develop a numerical model for testing by performing a finite differences discretisation of Eq. 7 and the basal boundary condition over a sigma coordinate system, where grid points are unevenly-spaced. This nonuniform grid can follow either a

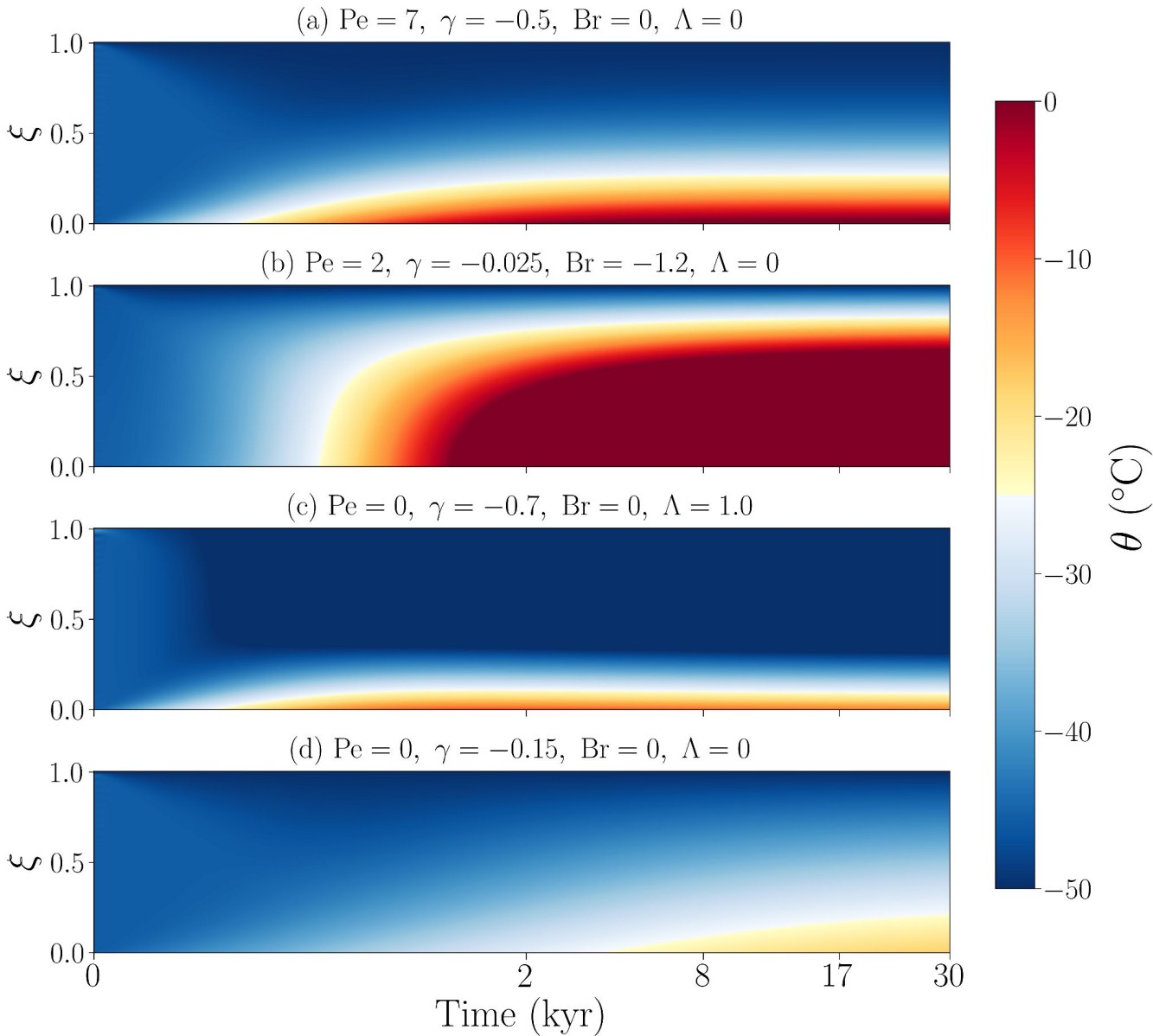

**Figure 4.** Time-dependent solution $\theta(\xi, \tau)$ given an initial temperature profile. For simplicity, here the initial temperature profile is $\theta_0(\xi) = -40$°C at all depths and in all cases.

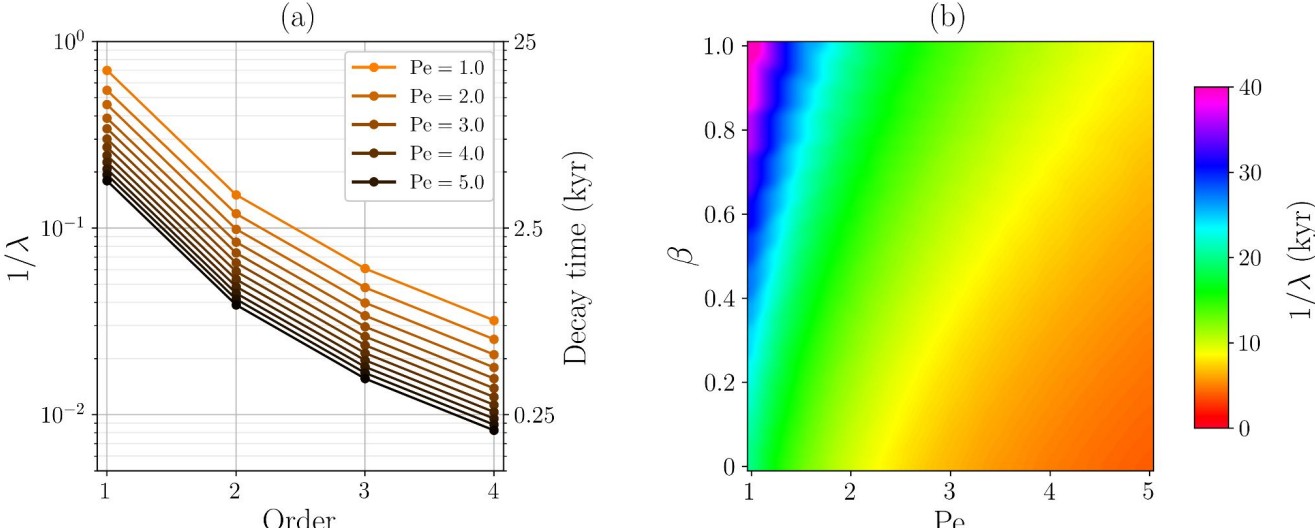

**Figure 5.** Decay time and corresponding eigenvalues. (a) First four eigenvalues for the set of $\beta$ values shown in Fig. 2b. (b) Decay time (kyr) of the first eigenvalue as a function of Pe and $\beta$.

**Table 3.** Benchmark experiments for numerical solvers and main physical processes considered for heat propagation. The experiments are named in increasing complexity order.

| Experiment name | Physical processes | | | |
|---|---|---|---|---|
| | Diffusion | Vertical adv. | Strain heating | Horizontal adv. |
| Exp-1 | Yes | No | No | No |
| Exp-2 | Yes | Yes | No | No |
| Exp-3 | Yes | Yes | Yes | No |
| Exp-4 | Yes | Yes | Yes | Yes |

quadratic or an exponential relation, set by the user. This yields higher resolutions near the base for a fixed number of points, thus minimising the computational costs. Several discretisation schemes are employed with varying orders of convergence, summarised in Table 4. Numerical solutions are then compared at equilibrium with their analytical counterpart (Fig. 6).

As could be expected, Figure 6 illustrates that spatial discretisation becomes a fundamental piece to obtain an accurate temperature solution, particularly at the base of the ice. The purely diffusive scenario (Exp-1, Fig. 6a) shows the smallest (negligible) errors for all discretisation schemes given its mathematical simplicity. If vertical advection is further introduced (Exp-2, Fig. 6), the particular choice by which the temperature first derivative $\theta_\xi$ is discretised becomes important, as temperature gradients can be transported via non-zero vertical velocities. Forward stencils slightly overestimates (F-2p) and underestimates (F-3p) the solution as shown in Fig. 6b. On the contrary, symmetric stencils S-2p provides a numerical solution significantly

**Table 4.** Finite-difference approximations employed in the numerical study (Fig. 6) for unevenly-spaced grids $\zeta_i$, as detailed in Appendix D. Distance between two adjacent points is defined as $h_i = \zeta_{i+1} - \zeta_i$. Note that vertical velocities are negative (downwards movement of ice) and the advection stencils are consequently adjusted. Discretization coefficients for the S-5p scheme are given in Appendix D.

| Quantity | Continous | Discrete approx. | Stencil name | Order |
|---|---|---|---|---|
| Diffusion | $\theta_{\xi\xi}$ | $\dfrac{2\left[h_{i-1}\theta_{i+1}-\left(h_i+h_{i-1}\right)\theta_i+h_i\theta_{i-1}\right]}{h_i h_{i-1}\left(h_i+h_{i-1}\right)}$ | Three-point symmetric (S-3p) | $\mathcal{O}(\varepsilon^2)$ |
| | | $c_{i+2}\theta_{i+2} + c_{i+1}\theta_{i+1} + c_i\theta_i + c_{i-1}\theta_{i-1} + c_{i-2}\theta_{i-2}$ | Five-point symmetric (S-5p) | $\mathcal{O}(\varepsilon^4)$ |
| Vert. advection | $w\theta_\xi$ | $-w_i \dfrac{\theta_{i+1}-\theta_i}{h_i}$ | Two-point forward (F-2p) | $\mathcal{O}(\varepsilon^1)$ |
| | | $-w_i \dfrac{\theta_{i+1}-\theta_{i-1}}{h_i+h_{i-1}}$ | Two-point symmetric (S-2p) | $\mathcal{O}(\varepsilon^2)$ |
| | | $-w_i\left[\dfrac{2h_{i-1}+h_i}{h_{i-1}\left(h_{i-1}+h_i\right)}\theta_i - \dfrac{h_{i-1}+h_i}{h_{i-1}h_i}\theta_{i+1} + \dfrac{h_{i-1}}{h_i\left(h_{i-1}+h_i\right)}\theta_{i+2}\right]$ | Three-point forward (F-3p) | $\mathcal{O}(\varepsilon^2)$ |
| Basal BC | $\theta_\xi$ | $\dfrac{\theta_1-\theta_0}{h_0}$ | Two-point forward (F-2p) | $\mathcal{O}(\varepsilon^1)$ |
| | | $\dfrac{2h_0+h_1}{h_0(h_0+h_1)}\theta_0 - \dfrac{h_0+h_1}{h_0 h_1}\theta_1 + \dfrac{h_0}{h_1(h_0+h_1)}\theta_2$ | Three-point forward (F-3p) | $\mathcal{O}(\varepsilon^2)$ |

closer to the analytical profile, particularly near the base. The next benchmark experiment (Exp-3, Fig. 6c), where the in-homogenous term captures a source of heat throughout the column due to strain deformation, presents a similar behaviour, where the F-3p stencil undersetimates the solution. Again, the symmetric scheme outperforms the asymmetric ones. Lastly, the inhomogeneous term is introduced, physically capturing a vertically-averaged source or sink of heat as a consequence of the advected ice in the horizontal dimension. We thus considered a negative contribution that physically describes a downstream advection of colder ice (Exp-4, Fig. 6d). Numerical solutions overestimate the analytical solution for the assymetric discreti-sation schemes (i.e., F-2p and F-3p), unlike the two-point symmetric scheme (S-2p). It is worth noting that the closest result to the analytical solution is obtained using S-2p for the advective term and F-3p for boundary condition discretisation. In the remaining experiments, the particular scheme employed in the basal boundary condition does not modify the solution.

For all experiments tested, results are identical irrespective of the particular discretisation of the diffusion term (Table 4), so that both a three-point and a five-point symmetric stencils yield the same stationary temperature profiles. Overall, all finite differences stencils herein presented successfully converge (Fig. 6e) for all benchmark experiments, yielding the smallest residual error for the purely diffusive scenario (Exp-1).

Additionally, we perform a resolution convergence test for the best discretisation choice (Table 4): a F-3p for the diffusive term, a S-2p for vertical advection and a F-3p basal boundary condition. In order to quantify the residual error as a function of the spatial resolution for each benchmark experiment (Fig. 7), we compute the $\ell^2$-norm of the difference between the

numerical and the analytical solutions $\varepsilon = ||\vartheta_{\text{num}} - \vartheta||_{\ell^2}$, defined as $||\boldsymbol{x}||_{\ell^2} = \left(\sum_i x_i^2\right)^{1/2}$. The larger deviations from the analytical solutions are found for the lower half of the ice column and are strongly dependent on the vertical resolution. Results show that a coarse resolution tends to overestimate the equilibrium temperature for all benchmark experiments. The residual error between the analytical and numerical solution exponentially decays, reaching values of $\varepsilon < 10^{-2}$ for $n > 15$.

## 7  Discussion

The adoption of dimensionless variables results in enhanced generality and mathematical convenience, albeit at the expense of veiling the practical significance to real glaciers and ice sheets. We have consequently tabulated data for characteristic values to ease interpretation (Table 1), thus showing that the explored range encompasses realistic values found in ice caps (Table 2).

We first start by comparing our results with a previously obtained solution for a simpler case (e.g., Clarke et al., 1977). We obtain identical results by setting the ice surface temperature to a fixed value given by the air temperature, i.e., imposing $\beta = 0$ in Eq. 2 (Figs. 2c and 2f). Prominently, note that a non-zero $\beta$ value is fundamental in the transitory regime (Fig. 11), though it leads to negligible changes at equilibrium (Fig. 2).

The transient behaviour of the solution is intricate given the freedom to choose an arbitrary initial state. This issue can be overcome by direct inspection of the eigenvalues of the problem. An estimation of the decay time of the analytical solution shows that the advection and the surface insulation are the only parameters that determine the timescale to reach thermal equilibrium. This approach has some limitations, some of which we now discuss. The decay time dependency is subjected to the mathematical form of our problem (Eq. 2). If an analytical solution could be obtained with an additional explicit horizontal advection term (rather than a vertically-averaged contribution), then the eigenvalues, and consequently the decay times, would also depend on $\Lambda$. A second limitation concerns the boundary conditions. This solution required time-independent conditions and therefore the decay time estimations do not hold if, for instance, the surface temperature changes over time. Even so, the approach developed here provides estimates of relaxation times under different physical conditions and gives an explicit expression for the time-dependent temperature profile from any arbitrary initial state.

The tractability of the analytical solution does not allow for further complexity and hence additional numerical methods would be necessary if such a physical description is desired. Nonetheless, a constant horizontal advection term $\Lambda$ was also introduced as part of the inhomogeneous term $\Omega$, for which the sign of the horizontal temperature gradients must be chosen a priori. Even though horizontal variability of temperature distributions can vary greatly, we account for this effect assuming a constant term (throughout the ice column) entering the heat equation, thus not reflecting much of the non-local features of the thermal structure of the ice sheets.

It must be stressed that our analytical solutions are not limited to regions with negligible horizontal velocities, since the true constraining quantity is the vertical gradient of the horizontal velocity $u_z$. Hence, rapidly sliding regions with a small vertical gradient of the horizontal velocity are also suitably described by our solutions, for that $u_z \simeq 0$ implies that the temperature profile is merely transported along the flow direction, while compressing the temperature gradient as the ice stream thins (Robel et al., 2013). One can argue that the additional source of heat due to frictional dissipation should be therein also

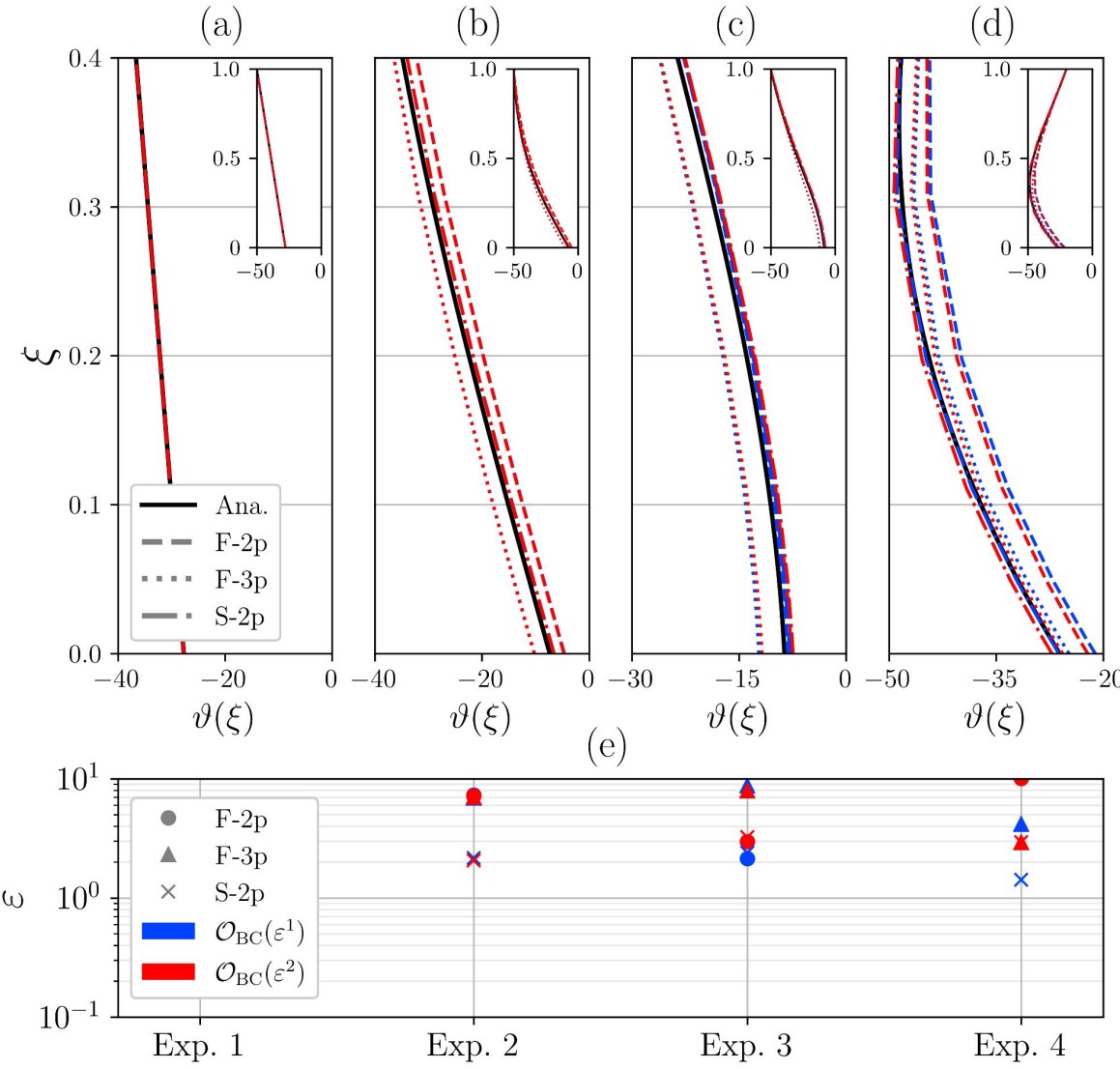

**Figure 6.** Numerical steady-state solutions (red, blue) for all discretisations shown in Table 4 compared with the analytical solution (solid black). Colour code represents the two asymmetric discretisation schemes for the basal boundary condition: F-2p (blue) and F-3p (red). Marker and line styles denote the discretisation stencil of the vertical advective term. The number of vertical points $n = 10$ is fixed for all cases. Numerical solutions are identical upon spatial discretisation of the difussive term at orders $\mathcal{O}(\varepsilon^2)$ and $\mathcal{O}(\varepsilon^4)$ (see Table 4). The purely diffusive case (Exp. 1) yields negligible errors $\varepsilon < 10^{-5}$.

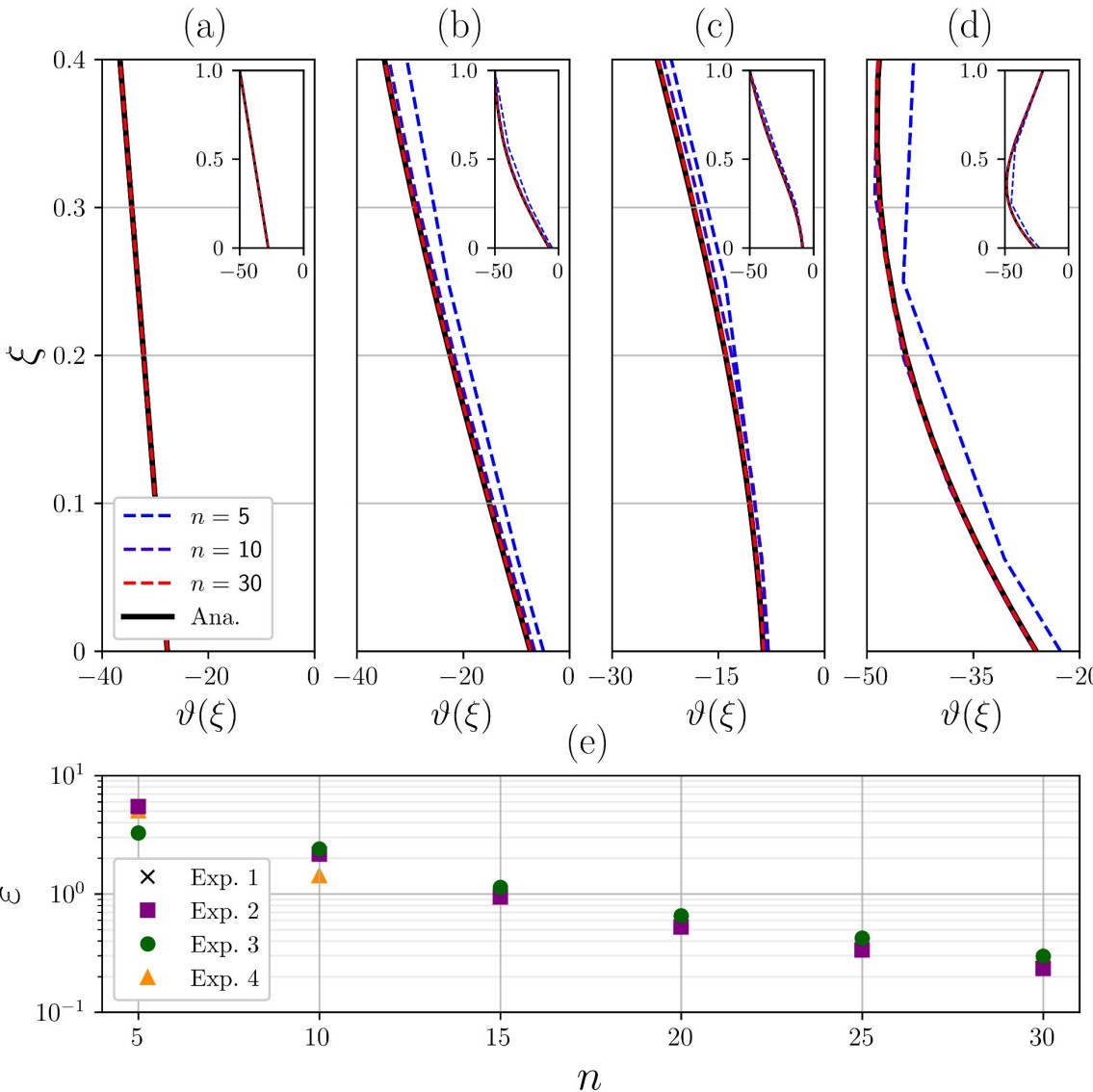

**Figure 7.** Convergence study of benchmark experiments. Steady-state analytical solutions shown in black solid line. (a) Exp-1, (b) Exp-2, (c) Exp-3 and (d) Exp-4, (e) Residual error defined as $\varepsilon = ||\vartheta_{\mathrm{num}} - \vartheta||_{\ell^2}$. For all experiments, $\gamma = 2$ and $\beta = 0$.

considered. Nonetheless, in terms of the temperature distribution, this effect is equivalent to an increased geothermal heat flow, as it is purely restricted to the column base and therefore already encompassed in Eq. 7.

The strain rate regime pose further limitations on the applicability of the solution. Particularly, regions where vertical shear dominates and the strain heat dissipation is concentrated near the base, a vertically-averaged contribution appears to be inaccurate. Nevertheless, as already noted by Rezvanbehbahani et al. (2019), this effect is instead well captured by an increase in the inflow of heat from the base (i.e., equivalent to a larger geothermal or frictional heat term) under conditions where most of the vertical shear is concentrated in the basal layers (Fowler, 1992).

It is worth noting that phase changes are not herein considered, so that temperature evolution is strictly confined to values below the pressure-melting point. Unlike a numerical solver, where temperature is manually limited, these solutions must be taken with caution as we are describing a frozen ice column. Results are still compatible with a potential heat contribution due to basal frictional heat Eq. 2, even though fast sliding regions are often related with temperate basal conditions. Nevertheless, an additional heat contribution would imply an increased vertical temperature gradient even if the column base eventually reached the pressure-melting point.

Knowing that ice forms by snow densification through time (Stevens et al., 2020), we find layers of progressively increasing ice density descending from the surface. Likewise, snow thermal conductivity increases with density (e.g., Sturm et al., 1997, 2002; Calonne et al., 2011, 2019), resulting in a poorer heat conductor as the snow-air interface is approached. As already noted by Carslaw and Jaeger (1988), if the flux across a surface is proportional to the temperature difference between the surface and the surrounding medium, the appropriate boundary condition takes the form of Eq. 1, rather than the oversimplified version $\theta(L,t) = T_{\text{air}}$. Here we explicitly describe the ice column with a constant thermal conductivity to keep analytical tractability, but we aim at describing the fact that the thermal conductivity of glacial ice $k(\rho)$ is reduced towards the surface. Following Carslaw and Jaeger (1988), we apply a general "Newton's Law" that also captures the traditional approach (i.e., imposing a particular ice surface temperature given by the air temperature) as a limit case if $\beta \to 0$.

Our suite of benchmark experiments allows us to test numerical solvers and assess reliability for different discretisation schemes and resolutions. The basal boundary condition is sensitive to the particular discretisation scheme, as the geothermal flux is the main source of heat in the ice column and is considered via a Neumann boundary condition. The simplest two-point stencil does not correctly represent the equilibrium temperatures, yielding larger deviations at the base (Fig. 6). Higher order discretisations are necessary to obtain a more reliable temperature distribution. In our benchmark experiments, we find significant improvement between $\mathcal{O}(\varepsilon^1)$ and $\mathcal{O}(\varepsilon^2)$ schemes for the basal boundary condition (Fig. 6), particularly for scenarios with large strain heating values or strong horizontal heat advection. Results for the different vertical advection schemes show that forward stencils (both F-2p and F-3p) deviate further from the analytical solution when compared to a symmetric scheme. Despite the fact that symmetric advective schemes might show some instabilities, we have not found any numerical issues in the present study. On the contrary, such schemes appear to outperform the asymmetric counterparts for all benchmark experiments.

Resolution plays a fundamental role to obtain a reliable temperature profile. A sigma coordinate system with quadratic spacing accurately ($\varepsilon < 10^{-2}$) reproduces the analytical solution for $n \geq 15$ grid points provided our best numerical scheme choice. Additional calculations performed for an exponential grid spacing (not shown) reveal consistent results with the quadratic de-

pendency (Figs. 6 and 7). This shows robustness of our numerical schemes, from which the symmetric advective stencil (S-2p) and the three-point basal boundary conditions (F-3p) again outperform the remaining choices.

## 8 Conclusions

We have determined the analytical solution to the 1D time-dependent advective-diffusive heat problem including additional terms due to strain rate deformation and depth-integrated horizontal advection. A Robin-type top boundary condition further considers potential non-equilibrium temperature states across the ice-air interface. The solution was expressed in terms of confluent hypergeometric functions following a separation of variables approach. Non-dimensionalisation reduced the parameter space to five numbers that fully determine the shape of the solution at equilibrium. We further overcome the arbitrariness on the initial temperature profile by directly calculating the eigenvalues of the problem and their corresponding decay times as an estimation of the time scale of our system in different physical scenarios. The transient component exponentially converges to the stationary solution with a decay time that solely depends on vertical advection and surface insulation.

The sign of vertical advection is of utmost importance as it determines the direction along which temperature gradients are transported. We have focused in the present study on the downward advective scenario, given the implausibility of an upward advection of ice. At equilibrium, basal temperatures are particularly sensitive to four physical quantities: vertical advection, geothermal heat flow, strain heat and lateral advection. On the contrary, the surface insulation yields negligible changes in the stationary solution. This is true even for highly insulating conditions at the ice surface, so long as colder ice is transported more efficiently than heat travels upwards due to diffusion.

The transient regime shows a strongly distinct behaviour. The arbitrariness of the initial state is overcome by a direct inspection of the eigenvalues of the problem. We then obtain a magnitude that represents the decay time of each Fourier mode that provides information about the equilibration time of the system. We find that the decay time of the transient component solely depends on two magnitudes: advection (Pe) and surface insulation ($\beta$). The remaining dimensionless parameters shape the temperature solution, though they have no influence in the timescale to reach thermal equilibrium. Strong advective regimes (Pe $\sim 5$) yield $\sim$ 2-10 kyr decay times under null and strong surface insulation conditions, $\beta = 0$ and $\beta = 1$ respectively. On the contrary, weak advective regimes are characterised by longer timescales $\sim$ 20-40 kyr, also depending on the particular insulating scenario.

Our suite of benchmark experiments are convenient for assessing accuracy and reliability of numerical schemes. We have employed unevenly-spaced grid discretisations to obtain higher resolution near the base whilst minimising the total number of grid points, thus reducing computational costs. A symmetric discretisation of the advective term combined with a three-point basal boundary condition yields the best agreement compared to analytical solutions. In terms of convergence and grid resolution, we find that $n \geq 15$ is the lower limit to obtain accurate temperature profiles. These results are robust both for a quadratic and an exponential grid spacing.

Lastly, we note that our analytical solutions are general and can be applied to any initial boundary value problem that fulfils the conditions herein described. They can provide temperature distributions for any 1D problem at arbitrarily high spatial and

temporal resolutions, that considers the combined effects of diffusion, advection and strain heating without any additional numerical implementation. Furthermore, they present a reliable benchmark test for any numerical thermomechanical solver to quantify accuracy losses and necessary spatial and temporal resolutions.

*Code availability.* TEXT

*Data availability.* TEXT

*Code and data availability.* All scripts to obtain the results herein presented and to further plot figures can be found in: https://github.com/d-morenop/Suplementary_ice-column-thermodynamics

*Sample availability.* TEXT

*Video supplement.* TEXT

**Appendix A: Separation of variables and full solution**

Let us briefly outline the separation of variables technique before elaborating on the solutions of our general problem. Consider the following initial/boundary problem on an interval $\mathcal{L} \subset \mathbb{R}$,

$$
\begin{cases}
\mu_\tau = \mu_{\xi\xi} - w\mu_\xi, & \xi \in \tilde{\mathcal{L}}, \ \tau > 0, \\
\mu = \mu_0, & \xi \in \tilde{\mathcal{L}}, \ \tau = 0, \\
\mu_\xi = 0, & \xi = 0, \ \tau > 0, \\
\beta\mu_\xi + \mu = 0, & \xi = 1, \ \tau > 0,
\end{cases}
\tag{A.1}
$$

This technique looks for a solution of the form:

$$
\mu(\xi, \tau) = X(\xi)T(\tau),
\tag{A.2}
$$

where the functions $Y$ and $T$ are to be determined. Assuming that there exists a solution of A.5 and plugging the function $\mu = XT$ into the heat equation, it follows:

$$
\frac{T_\tau}{T} = \frac{X_{\xi\xi}}{X} - w\frac{X_\xi}{X} = -\lambda,
\tag{A.3}
$$

for some constant $\lambda$. Thus, the solution $\mu(\xi, \tau) = X(\xi)T(\tau)$ of the heat equation must satisfy these equations. In order for a function of the form $\mu(\xi, \tau) = X(\xi)T(\tau)$ to be a solution of the heat equation on the interval $\mathcal{I} \subset \mathbb{R}$, $T(\tau)$ must be a solution of the ODE $T_\tau = -\kappa \lambda T$. Direct integration leads to:

$$T(\tau) = Ae^{-\kappa \lambda \tau}, \tag{A.4}$$

for an arbitrary constant $A$.

Additionally, in order for $\mu(\xi, \tau)$ to satisfy the boundary conditions, we arrive to a second-order linear ordinary differential equation:

$$\begin{cases} X_{\xi\xi}(\xi) - w(\xi)X_\xi(\xi) + \lambda X(\xi) = 0, & \xi \in \tilde{\mathcal{L}}, \\ X_\xi = 0, & \xi = 0, \\ \beta X_\xi + X = 0, & \xi = 1, \end{cases} \tag{A.5}$$

It is necessary to provide the particular shape of the the function $w(\xi)$. First, we will employ the linear profile $w(\xi) = w_0\xi$ so that the differential equation now reads $X_{\xi\xi}(\xi) - w_0\xi X_\xi(\xi) + \lambda X(\xi) = 0$. This equation can be easily identified with the well-known confluent hypergeometric differential equation (e.g., Abramowitz and Stegun, 1965; Evans, 2010) defined as:

$$\xi X_{\xi\xi} + (\delta - \xi)X_\xi - \alpha X = 0, \tag{A.6}$$

Simply by defining $\alpha = -\lambda/(2w_0)$, $\delta = 1/2$ and $\zeta = w_0\xi^2/2$, we can write our solution in terms of the two independent Kummer and Tricomi functions:

$$X(\xi) = C_1 \Phi(\alpha, \delta, \zeta) + C_2 \Psi(\alpha, \delta, \zeta) \tag{A.7}$$

where $C_1$ and $C_2$ are constants to be determined from the boundary conditions. At the base, the solution must be finite, so we set $C_2 = 0$ given that Tricomi function $\Psi(\alpha, \delta, \zeta)$ diverges at the origin. The second boundary condition (i.e., at $\xi = 1$) allows us to determine the eigenvalues $\lambda_n$ of the problem as we look for all values of $\alpha_n$ that satisfy:

$$\beta \Phi_\xi(\alpha_n, \delta, \zeta) + \Phi(\alpha_n, \delta, \zeta) = 0, \text{ at } \xi = 1, \tag{A.8}$$

and then we compute the eigenvalues $\lambda_n = -2w_0\alpha_n$. This is in fact a trascendental equation with no algebraic representation and therefore, the values of $\alpha_n$ are numerically determined.

Thus, for each eigenfunction $X_n$ with corresponding eigenvalue $\lambda_n$, we have a solution $T_n$ such that:

$$\mu_n(\xi, \tau) = X_n(\xi)T_n(\tau), \tag{A.9}$$

is a solution of the heat equation on our interval $\mathcal{I}$ which satisfies the BC. Moreover, given that the problem A.5 is linear, any finite linear combination of a sequence of solutions $\{\mu_n\}$ is also a solution. In fact, it can be shown that an infinite series of the form:

$$\mu(\xi, \tau) \equiv \sum_{n=0}^{\infty} \mu_n(\xi, \tau), \tag{A.10}$$

will also be a solution of the heat equation on the interval $\mathcal{I}$ that satisfies our BC, under proper convergence assumptions of this series. The discussion of this issue is beyond the scope of this work.

We can then express the transitory solution as:

$$\theta(\xi,\tau) = \sum_{n=0}^{\infty} A_n \Phi\left(\alpha_n;\delta;\zeta\right) e^{-\lambda_n \tau} \tag{A.11}$$

where the coefficients $A_n$ are given by the initial condition.

Since the confluent hypergeometric functions are orthogonal, the normalized eigenfunctions form an orthonormal basis under the $\varrho(\xi)$-weighted inner product in the Hilbert space $L^2$, thus allowing to write the coefficients $A_n$ as:

$$A_n = \frac{1}{||\Phi_n||^2} \int_0^1 \left(\theta(\xi,0) - \vartheta(\xi)\right) \varrho(\xi) \Phi\left(\alpha_n;\delta;\zeta\right) d\xi. \tag{A.12}$$

where $\theta(\xi,0)$ is the initial temperature distribution, $\varrho(\xi) = e^{-w_0 \xi^2/2}$ and $||\Phi_n||^2$ is defined by the inner product:

$$||\Phi_n||^2 = \langle \Phi_n, \Phi_n \rangle = \int_0^1 \Phi\left(\alpha_n;\delta;\zeta\right) \varrho(\xi) \Phi\left(\alpha_n;\delta;\zeta\right) d\xi. \tag{A.13}$$

**Appendix B: Stationary solution**

For the stationary regime, we do not need to apply separation of variables for that the problem reduces to a second-order ordinary differential equation in only one independent variable $\xi$:

$$\begin{cases} \Omega = \vartheta_{\xi\xi} - w\vartheta_\xi, & \xi \in \mathcal{L}, \\ \vartheta_\xi = \gamma, & \xi = 0, \\ \beta\vartheta_\xi + \vartheta = 1, & \xi = 1, \end{cases} \tag{B.1}$$

Even though we have increased the complexity of the problem with a refined top boundary condition and non-homogeneous term $\Omega$, the solution can still be found analytically:

$$\vartheta(\xi) = \Omega \frac{\xi^2}{2} \, _2F_2\left(1,1;\frac{3}{2},2;-\zeta\right) + A \operatorname{erf}\left[a\xi\right] + B \tag{B.2}$$

where $_2F_2(a_1,a_2;b_1,b_2,x)$ is the generalised hypergeometric function, $\zeta = (a\xi)^2$, $a = (w_0/2)^{1/2}$, $A = -\gamma\left(\pi/(4a)\right)^{1/2}$ and $B = 1 - A\left(2a\pi^{-1}\beta e^{-a^2} + \operatorname{erf}\left[a\right]\right) - \Omega\left((\beta + 1/2) \, _2F_2(1,1;3/2,2,a^2) + \beta a^2 \, _2F_2(2,2;5/2,3,a^2)/3\right)$ is a constant given by the top boundary condition. Note that hypergeometric function can be easily differentiated following e.g., Eq. 15.2.1 in Abramowitz and Stegun (1965).

## Appendix C:  General power-law velocity profiles

In this section, we also assume thermal equilibrium, thus reducing again the problem to a second-order ordinary differential equation in only one independent variable $\xi$:

$$
\begin{cases}
0 = \vartheta_{\xi\xi} - w\vartheta_\xi, & \xi \in \mathcal{L}, \\
\vartheta_\xi = \gamma, & \xi = 0, \\
\beta\vartheta_\xi + \vartheta = 1, & \xi = 1,
\end{cases}
\tag{C.1}
$$

where we have set $\Omega = 0$ to ensure analytical tractability for a general power-law velocity profiles. This solution is consequently limited to regions where Pe, $\gamma \gg \Lambda$, Br.

Unlike the general stationary solution shown in Eq. B.2, we allow for a general power-law vertical velocity profile of the form $w(\xi) = w_0\xi^m$. The solution can be then expressed as:

$$
\vartheta^-(\xi) = \frac{p\gamma}{(pw_0)^p}\, \Gamma\left(p, pw_0\xi^{m+1}\right) + C
\tag{C.2}
$$

where $p = (m+1)^{-1}$, $C = 1 - [2\beta\,(pw_0)^p\, e^{-pw_0} + \Gamma(p, w_0p)]\, p\gamma/(pw_0)^p$ is a constant given by the top boundary condition and $\Gamma(\cdot, \cdot)$ is the upper incomplete gamma function defined as:

$$
\Gamma(a, x) = \int_x^\infty e^{-t} t^{a-1} dt
\tag{C.3}
$$

Additionally, the solution can be also expressed in terms of Kummer confluent hypergeometric function $\Phi$ given the relation (Abramowitz and Stegun, 1965, Eqs. 6.5.3 and 6.5.12):

$$
\Gamma(a, x) = \Gamma(a) - a^{-1} x^a e^{-x} \Phi(1, 1+a; x)
\tag{C.4}
$$

Hence, the stationary solution is equivalent to $\sim \Phi\left(1, p+1; pw_0\xi^{m+1}\right)$.

## Appendix D:  Discretisation schemes

Our finite differences discretisation considers unevenly-spaced grids, commonly used in the glaciological community where higher resolutions are desired near the base whilst minimising the required number of points to reduce computational costs. We thus build a new coordinate system $\zeta$ considering two types of nonuniform grid spacing: polynomial and exponential. Given that our original variable $\xi \in [0, 1]$, these relations can be expressed as:

$$
\zeta = \xi^n
\tag{D.1}
$$

where $n$ is the spacing order, and:

$$
\zeta = \frac{e^{s\xi} - 1}{e^s - 1}
\tag{D.2}
$$

where $s$ is the spacing factor for the exponential grid. In this study, we have employed $n = 2$ and $s = 2$.

We now present the numerical schemes necessary to account for non-homogeneous grids $\zeta$. The distance between two adjacent points is defined as $h_i = \zeta_{i+1} - \zeta_i$. The five-point symmetric second-order derivative then reads:

$$\theta_{\xi\xi}(\xi_i) \simeq \frac{-2h_i(2h_{i+1} + h_{i+2}) + 2h_{i+1}(2h_{i+1} + h_{i+2})}{h_{i-1}(h_{i-1} + h_i)(h_{i-1} + h_i + h_{i+1})H_i}\theta_{i-2} + \frac{2(2h_{i-1} + h_i)(2h_{i+1} + h_{i+2}) - 2h_{i+1}(h_{i+1} + h_{i+2})}{h_{i-1}h_i(h_{i-1} + h_{i+1})(h_i + h_{i+1} + h_{i+2})}\theta_{i-1}$$
$$+ \frac{2h_i(h_{i-1} + h_i) - 2(h_{i-1} + 2h_i)(2h_{i+1} + h_{i+2}) + 2h_{i+1}(h_{i+1} + h_{i+2})}{(h_{i-1} + h_{i+1})h_ih_{i+1}(h_{i+1} + h_{i+2})}\theta_i$$
$$+ \frac{2(2h_{i-1} + 2h_i)(h_{i+1} + h_{i+2}) - 2h_i(h_{i-1} + h_i)}{(h_{i-1} + h_i + h_{i+1})(h_i + h_{i+1})h_{i+1}h_{i+2}}\theta_{i+1} + \frac{2(h_{i-1} + h_i)h_i - 2(2h_{i-1} + h_i)h_{i+1}}{H_i(h_i + h_{i+1} + h_{i+2})(h_{i+1} + h_{i+2})h_{i+2}}\theta_{i+2}$$

(D.3)

where $H_i = h_{i-2} + h_{i-1} + h_i + h_{i+1} + h_{i+2}$. This result is consistent with Singh and Bhadauria (2009).

*Author contributions.* Daniel Moreno-Parada formulated the problem, derived the analytical solutions, coded the numerical solvers, analysed the results and wrote the paper. All other authors contributed to analyse the results and writing the paper.

*Competing interests.* Alexander Robinson is an editor of The Cryosphere. The peer-review process was guided by an independent editor, and the authors have also no other competing interests to declare.

*Disclaimer.* TEXT

*Acknowledgements.* This research has been supported by the Spanish Ministry of Science and Innovation (project IceAge, grant no. PID2019-110714RA-100), the Ramón y Cajal Programme of the Spanish Ministry for Science, Innovation and Universities (grant no. RYC-2016-20587) and the European Commission, H2020 Research Infrastructures (TiPES, grant no. 820970).

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
