# Peer review of "Analytical solutions for the advective-diffusive ice column in the presence of strain heating"

_The Cryosphere, 2022_

## Referee Comment (RC2)

**Referee Report: "On the periodicity of free oscillations for a finite ice column" by Moreno et al.**

**General Comments**

In this manuscript, the authors consider how the length of a one-dimensional finite ice column modifies the timescales associated with heat transfer through the column, when compared to a semi-infinite column. They apply a Robin type boundary condition at the upper surface, which permits heat transfer across the ice surface to be accounted for. They apply Fourier analysis to the heat equation to derive a general form of the solution and present numerical evaluations of this solution. They focus in particular on the effect of the column length, surface temperature flux, and initial conditions on the temperature profile and the melting timescale (the time taken for the base to reach melting point). They conclude that considering a finite, rather than semi-infinite, column has important consequences for the temperature profile and melting timescale, suggesting that the previous results for the Heinrich event timescale based on one dimensional semi infinite ice columns could be quite wrong.

Overall, I found this article interesting, but somewhat lacking. I appreciated the simplicity of the approach and use of analytic techniques. However, I felt that the analysis was a little thin, and the system was not explored in a great level of detail. In addition, I felt that the authors made a number of general statements that are based on the small range of parameters considered (outlined in the individual comments below). There are also a number of places where the language is difficult to follow; in these places, I have tried to infer the scientific content of the sentence, rather than judging the written English of the authors. It is possible that I have incorrectly inferred what the authors intend to communicate; in these places I would welcome the authors to respond critically. There are also many modelling assumptions which are not justified and steps jumped, while the method of separation of variables (which is fairly standard) is set out in great detail.

While the analysis that the authors have conducted is fairly standard, that does not preclude it from representing an important advance. While I have an interest in them, I am by no means an expert in Heinrich events and Daansgaard Oeschger cycles and therefore I am not able to fully comment on the implications of this work in this context. However, I do think that the authors could do a better job at expressing the implications of the work more clearly and its place within the field. In addition, I believe that many of the main conclusions are heavily reliant on modelling assumptions which are not justified, most notably the initial temperature profile.

Below I have set out my responses as scientific comments and technical comments (typos etc), however, there may be overlap between the two sets of comments.

**Scientific Comments**

- Title: I do not think the title really gives an indication of what the paper is about. In particular, the authors do not consider periodicity in their model and it is not clear (at least to me) what free oscillations refers to. I think heat transfer should at least be mentioned.
- Line 4: "general boundary condition problem": this is a stretch. The authors consider a Robin boundary condition at the surface, which is more general than a Dirichlet or Neumann BC, but certainly not every BC can be expressed in this form.
- Line 7: "depends on several factor: the ice column thickness L...". The time required depends on more than just these things (as the authors themselves show), e.g. the diffusivity. The phrasing makes it sounds as though the time depends only on L, the initial temp profile and the BC.
- Line 14: This may be my own ignorance, but clarification on what the authors mean by "thermomechanical instabilities" would be appreciated (this phrase appears frequently throughout the ms, and I was unable to infer its meaning from the context in any place).
- Line 16: The solutions presented in this paper are not, to my mind, analytic solutions. While I agree that the form of the solution has been expressed analytically, the eigenvalues must be determined numerically (as the authors point out) and the infinite sum presumably is also evaluated numerically. This should be clarified and the many statements about deriving analytic solutions should be adjusted to reflect.
- Line 36-37: can you elaborate on the statement "since it would be prohibited by the ice sheet heat transfer physics". I guess you mean that if the ice sheet is sufficiently thick, the base is insulated from the surface?
- Line 42: need to explain what the low order model is — an ice sheet model?
- Line 57: what does a 'unique spatial element' mean?
- Line 57: "the surface temperature and the geothermal heat flux were found to determine the character of the ice flow": I don't think that's true: these two factors are important controls, but don't together determine the *entire* ice flow.
- Line 61: what is a zero dimensional spatial model? One spatial dimension?
- Line 80: I disagree that this is the most general approach, what if there is e.g. precipitation at the surface which transfers heat?
- Line 86: I think it would be useful to explain the physical interpretation of \beta. What is a typical value? This would help a lot with interpretation of the numerical solutions (see below). The authors say it "modulates the permissible deviation between ice and air temps", which I agree with, but I don't think adds anything beyond what the equation says directly.
- Line 90: "the ice surface will consequently evolve in time towards $T_{air}$" - I don't think that's true? e.g. in the limit of a very short ice column, the heat flux at the surface will match that at the base (i.e. -G/k) and thus the surface temp will evolve towards $T = T_{air} + \beta G/k$? More generally, it is possible to have a heat flux out of the domain while the system is in steady state.
- Line 92: This formulation assumes that the diffusivity is constant, which should be stated. Is that a reasonable assumption to make? I am not an expert, but it seems like the firn layer at the surface might have a vastly different diffusivity to the rest of the column.
- Line 93: (Key): where does this initial profile come from? Why is it appropriate? It is simply introduced with no justification! The authors go on to show that the time to melting is sensitive to the particular linear profile chosen, so presumably it is also sensitive to the type of profile. Shouldn't the initial condition be set by the temperature profile immediately after an event in the binge purge cycle?

- The authors should also make it clear that this initial profile is something they impose on the model. In addition, surely the initial profile is only compatible with the BC for specific values of theta_l and theta_b, related to G and k etc?
- General comment on modelling: I am not sure why the authors did not non-dimensionalize their model. That seems like a straightforward way to reduce the number of parameters in the system, as well as gain insight into which processes are important in which locations. Reformulating the system in terms of dimensionless variables might help with the lack of generality of the numerical results which I point to later in my review. I also found it quite surprising that the authors did not even mention the heat equation by name.
- Line 96: I would not call this method "separation of variables", which I understand to be more general, and others may also have this confusion. The authors might consider removing mention of separation of variables (does it add anything?).
- Line 112: the authors mention using a numerical method, but provide no link to code required to solve equations or produce figures. Code and data should be held in (e.g.) an open repository.
- Line 113: what does the tolerance refer to here?
- Line 112-115: see above: this equation requires a numerical method, so the solution is not analytic!
- Line 122: what is $\tilde{G}$?
- General note on appendices: Appendix A is entirely standard (e.g. [https://courses.maths.ox.ac.uk/pluginfile.php/22143/mod_resource/content/1/FS-PDE-Slides-Week-4.pdf](https://courses.maths.ox.ac.uk/pluginfile.php/22143/mod_resource/content/1/FS-PDE-Slides-Week-4.pdf)) and does not need to be included (a reference to a textbook would suffice). Appendices B and C could also be described as standard, but could also be quite easily incorporated into the main text.
- Figure 2: need to state the values of kappa, T_air, k, G (etc?) used to generate these figures. Also, there is no need for a legend in both figures, and the final legend entry is not necessary (imo). Labels (a) and (b) are missing from the panels.
- The solutions are infinite sums, which must naturally be truncated somewhere. The authors should discuss their choice of truncation and justify that the terms ignored are not important (e.g. by considering their order of magnitude).
- Line 125: "the second time frame is chosen…": it would be nice to explain to the reader what they are looking for in the figure to show this (I think it is the blue line going through zero?)
- Important question: based on the previous comment, it seems that the authors are setting the freezing point (on which their timescale calculations are based) to zero? Is that correct? If so, why is there no pressure dependence in melting? Including pressure dependence would mean that larger columns don't need to get to as high a basal temperature in order to start melting, making the dependence of the timescale on L stronger, I think.
- Line 128: "this rate is in fact proportional to…". I think you have to work a bit harder to get to this. I agree that each mode has the quoted rate, but it is not obvious what happens when only takes the sum over all the modes (the value quoted is dependent of the dummy variable n!).
- Line 128-129: what are L1 and L2? I guess you are saying that if we have two columns of lengths L1 and L2, then the temp gradient at the base is larger in the longer one? If so, I don't think you can make this statement based on the rate argument made previously (see above comment). In any case, it is not obvious that the rate is increasing in L, particularly since lambda_n are complicated functions of L.
- Line 131-132: "the specific non-zero $\beta$ does not alter this behaviour" - this requires justification, you have only shown the results for one non zero beta (and one value of T_air, kappa etc).

- Line 134: so beta is the lengthscale over which the ice column feels the surface? This should be stated earlier.
- Is the value of \beta = 50m large or not? More generally, is the close agreement between the curves for \beta =0 and \beta = 50 because the system is not sensitive to beta, or because of the particular value of beta chosen? Non-dimensionalizing the system would help with this.
- Line 142: "it is clear that the column thickness is the primary factor…". I don't think you can make as strong a statement as this based on the figure. It is clear that the system is sensitive to L, but it is not clear that this is the most important factor (is it even possible to rank importance of factors?), and this figure is only for one particular set of T_{air}, kappa etc.
- Line 148: "…solely depends on L, G, T_{air}". That can't be true! You even said yourself that the rate depends on kappa (line 127: "in fact proportional to kappa lambda_n e^{-kappa lambda_n t})". There are also potentially other constants in the prefactor.
- Line 151: "dependency…on both the initial and boundary conditions". This is crucial: there is a very strong dependence on the initial conditions, which the authors have not justified. Why are they a sensible choice physically? Why is the timescale determined as the time from this (seemingly arbitrary) initial profile?
- Line 153: "though solely for ice thicknesses below ~ 2km": the figure only shows this for this particular set of beta, T_{air} etc. The following statement ("we therefore find L = 2km is a threshold value…") has not been shown in general. How does figure 4 change for different parameter values is not explained or pursued.
- Figure 4: caption: this is picky, but really theta_l and theta_b are parameters rather than boundary conditions. You should also state the values of all parameters used to create these figures. 'Initial basal temperature' in the final line should not be capitalized.
- Figure 4: I find it surprising that that the timescale T is large for small theta_b for L < 1.25 or so. Is this because the geothermal heat flux is too strong? Is there a limit in which the base never reaches thawing, or does it always happen just on longer and longer timescales?
This figure is very interesting but not really discussed. It raises many questions, e.g. the non-monotonic behaviour in (d) is also quite interesting, why is that? Also, there is a very sharp transition in (b): why is this transition so sharp (what sets this boundary)?
- Line 180: this is absolutely not clear, and took me about 10 minutes to figure out that the right hand side of the square bracket goes to zero, so you're looking for roots of tan(theta)=0.  This should be explained in more detail.
- Line 185-186: you have shown that the results are asymptotically equivalent, rather than identical (they differ by some small amount for any finite L).
- Line 190: so MacAyeal considered periodic forcing in semi infinite domain, and this work looks at a finite length domain with non-periodic forcing. Can the two changes (periodic vs non periodic and finite vs semi infinite) be decoupled?
- Line 192: Again, the strength is not determined solely by L, beta will also be important (for example)
- Line 194: It would be helpful to make it clear that the new development is the consideration of a finite length domain.
- Line 195: again, I disagree that these solutions are analytic, for the reasons outlined above. 'Analytic approach' also mentioned on line 205.
- Line 198: again, I don't think this is a general boundary condition considered. It's a Robin type (as discussed above).

- Line 202: You actually showed something stronger: you showed that the semi-infinite domain is an oversimplification provided that the column length is not large (within the context of the assumptions made).
- Figure 5: this figure should appear earlier in the text. It also has two black solid lines. Again, fixed parameter values should be stated. \tilde{G} appears again here. The behaviour of the blue and black solid lines in particular suggests that there is a critical value of the ice thickness below which the potential periodicity goes to infinity (as hinted at by figure 4). Is this the case?
- Line 207: It was certainly not shown analytically that L = 2km is an upper bound! That was a numerical result and there is no proof that this value holds in general (numerically or otherwise).
- Line 215: what is the estimate of a "medium" size based on?
- Line 216: "the explicit consideration of distinct initial temperature profile manifests a high sensitivity…to its initial state". Indeed…see earlier comments on the initial condition. Why is the initial conditions correct, or even sensible? And why is the timescale the time from this initial condition to thawing?
- Line 222-223: "the periodicity of such events cannot be imposed by the frequency of an external forcing". How do the authors come to this conclusion? There is no periodicity in their model.
- Line 227: "this double fold nature….is considered": I don't understand what the authors mean by thermomechanical instabilities here. Such things have not been discussed in the paper. Or is this a result from elsewhere? If so, what is the relevance?
- While I appreciate that the authors intend to improve upon the work of MacAyeal (1993) by considering more general conditions, it would also be nice to discuss the many simplifications made on a real ice ice sheet, of which there are many: heat transfer is three dimensional in practice, ice columns do not all have uniform length etc.

**Technical Corrections**

- Line 13: "these results ultimately manifest a": results are not active, they do not manifest things. "These results suggest/show"?
- Line 35: need a reference for "appear to be dependent on environmental factors"
- Line 38-39: the sentence "In fact…, for a motionless ice column" is very confusing to anyone who has not read the MacAyeal paper in question. Could the authors clarify? Do they mean that MacAyeal showed that a periodicity of T = 7000yrs corresponds to an e-folding decay length of 14m?
- Line 50: SIA has not yet been defined.
- Line 52: causes -> caused
- Line 53: Is this comment 'thus'? I don't think it necessarily immediately follows from the examples above…
- Line 72-75: this description of the paper's structure doesn't actually match what appears in the following (e.g. section 7 has the limit L \to \infty)
- Figure 1: I don't see why it is necessary to show the x axis? The problem is one dimensional.
- Figure 1 caption: specify that non-equilibrium thermal states refers to equilibrium across the ice-air interface.
- Line 84: "as a result": I don't think this sentence is a direct consequence of the previous one.
- Line 90: "Thus": this is not appropriate: the heat equation doesn't follow from the description of the BC.

- Line 98: there is a backwards quote mark. Also, what is 'shifting the data' an example of?
- Line 108: to my mind, that is an initial boundary value problem, rather than just a boundary problem.
- Line 127: I would avoid making statements like "the implications of a finite domain are clear"…It was not clear to me at this point! Your job is to explain the implications.
- Line 140: referred as -> referred to as
- Line 171: "excess" has a backwards quote mark at its beginning
- Line 222: which events does 'such events' refer to?

---

## Author Comment (AC1)

**Authors final response to Reviewer 1**
**TC-2022-97**

Daniel Moreno, Alexander Robinson, Marisa Montoya, and Jorge Alvarez-Solas.

September 20, 2022

Note: Reviewers comments are given in blue font whereas the author response reads in black.

**General comments:**

This manuscript highlights the computational efficiency of using the method of separation of variables to solve the diffusion equation rather than direct numerical methods. While this is well-known, the application to the basal temperature evolution in an ice column had previously only been done for infinitely deep ice; this manuscript considers the case of a shallow ice sheet and performs some example calculations for different initial conditions and boundary conditions. Overall, I was somewhat disappointed that the authors did not go into more depth in analysing and describing their results, in particular exploring the wealth of curious trends shown in figure 4 - most of the paper is instead given over to a routine description of the method of separation of variables. In particular, given the stated threshold of 2km for the solution to approach the infinite depth limit, it would be nice to explore what factors set this threshold. Looking at figure 5 there seems to be a rather narrow band of depth values for which T is finite but larger than the MacAyeal solution. I think figure 4b also shows this rather sudden regime change.

We thank the reviewer for such a comment. The authors have accordingly elaborated in the manuscript to understand the physics behind this threshold.

Briefly, we can understand the process as follows. Since we focus on the time required for the base to thaw, it is essential to consider the temperature gradient between the base and the top. The vertical temperature gradient must be supported by the geothermal heat flux. If the surface is too cold, the heat provided by G may not be sufficient to hold a

temperature difference large enough (within the column) so that the base reaches the melting point. For a given choice of $G$, $k$ and $T_{\text{air}}$, there exists a minimum ice thickness $L_{\text{min}}$ that yields a temperature gradient that allows the base to thaw. For thinner columns, the base will remain below the melting point. This further translates into a sudden jump in the potential periodicity shown in Fig. 4b.

**Specific comments:**

If Equation (6) were given as $\cot\left(L\sqrt{\lambda}\right) = \beta\lambda$, there would be no need to treat $\beta = 0$ as a special case. We thank the reviewer for such a comment. We will express Eq. 6 as $\cot\left(L\sqrt{\lambda}\right) = \beta\lambda$. Figure 4 - the values of the parameters held fixed are not given. Indeed, we will now include a table with the most relevant parameter values. Figure 4d - interesting that $T$ is non-monotonic with $L$ at $-14°C$. Why is this?

This is a quite complex behaviour since there are several factors that must be considered simultaneously. It is illustrative to look at the vertical profiles shown in Fig. 2. The fact that the temperature appears to be non-monotonic with $L$ at $-14C$ is a consequence of two factors: the necessary energy budget to warm an ice column and the vertical temperature gradient. The former increases with $L$, whereas the latter decreases with $L$.

For slight variations of the thickness $\delta L$ near $L = 1.5$ km (while fixing $T = -14C$), the time required to thaw the base is larger regardless of its sign. In other words, it takes longer to reach the pressure melting point both for a thinner and a thicker column. This local minima is a balance between the total energy necessary to heat a column and the fact that a thinner one implies a larger vertical gradient for a fixed temperature difference between the base and the top. Namely, we could consider the effect of these factors explicitly. First, a thinner column requires a smaller amount of total energy to increase the temperature of the column. However, considering the second factor, a thinner column would yield a larger vertical temperature gradient (ultimately yielding a slowdown in the warming rate as the geothermal heat flux is fixed in the BC). The combination of both effects allows for local minima.

The manuscript will be changed to address and discuss this behaviour explicitly.

Figure 4c - this figure shows the most interesting trends, but is barely discussed in the text. Perhaps using $\theta_L/L$ as the primary variable instead would clarify the impact of the temperature gradient on the basal evolution.

This comment agrees with the other two referees. A detailed discussion will be included in the manuscript addressing the most interesting trends. Using $\theta_L/L$ will be also considered to clarify the impact of the vertical temperature gradient on the basal evolution.

Line 162 - where T saturates to above 25kyr, are we in fact in a limit where T is infinite?

Yes, for certain boundary conditions the base would never thaw. The text will be updated for clarification.

Convergence towards no dependence on the detailed surface boundary conditions as $L \xrightarrow{\infty}$ could be moved to an appendix for better flow of the manuscript.

We have considered the possibility of moving this part to an appendix, but the comparison with previous work (MacAyeal, 1993a; b) fully relies on the limit $L \xrightarrow{\infty}$. In fact, we recover the well-known 7000-yr-periodicity widely used in the literature. While moving it might lead to a better flow of the manuscript, we feel that an important point would be lacking in the main text. We have made modifications to the text however, to try to nonetheless improve the flow.

**Technical corrections:**

Figure 4 colorbar caption could be oriented to match the axis label.

We thank the reviewer for such a comment. Figure 4 will be changed.

---

## Author Comment (AC2)

**Authors final response to Reviewer 2**
**TC-2022-97**

Daniel Moreno, Alexander Robinson, Marisa Montoya, and Jorge Alvarez-Solas.

September 20, 2022

Note: Reviewers comments are given in blue font whereas the author response reads in black.

**General comments:**

In this manuscript, the authors consider how the length of a one-dimensional finite ice column modifies the timescales associated with heat transfer through the column, when compared to a semi-infinite column. They apply a Robin type boundary condition at the upper surface, which permits heat transfer across the ice surface to be accounted for. They apply Fourier analysis to the heat equation to derive a general form of the solution and present numerical evaluations of this solution. They focus in particular on the effect of the column length, surface temperature flux, and initial conditions on the temperature profile and the melting timescale (the time taken for the base to reach melting point). They conclude that considering a finite, rather than semi-infinite, column has important consequences for the temperature profile and melting timescale, suggesting that the previous results for the Heinrich event timescale based on one dimensional semi infinite ice columns could be quite wrong.

Overall, I found this article interesting, but somewhat lacking. I appreciated the simplicity of the approach and use of analytic techniques. However, I felt that the analysis was a little thin, and the system was not explored in a great level of detail. In addition, I felt that the authors made a number of general statements that are based on the small range of parameters considered (outlined in the individual comments below). There are also a number of places where the language is difficult to follow; in these places, I have tried to infer the scientific content of the sentence, rather than judging the written English of the authors. It is possible that I have incorrectly inferred what the authors intend to

communicate; in these places I would welcome the authors to respond critically. There are also many modelling assumptions which are not justified and steps jumped, while the method of separation of variables (which is fairly standard) is set out in great detail. While the analysis that the authors have conducted is fairly standard, that does not preclude it from representing an important advance. While I have an interest in them, I am by no means an expert in Heinrich events and Daansgaard Oeschger cycles and therefore I am not able to fully comment on the implications of this work in this context. However, I do think that the authors could do a better job at expressing the implications of the work more clearly and its place within the field. In addition, I believe that many of the main conclusions are heavily reliant on modelling assumptions which are not justified, most notably the initial temperature profile. Below I have set out my responses as scientific comments and technical comments (typos etc), however, there may be overlap between the two sets of comments.

The authors are deeply grateful for the elaborated and constructive comments of the reviewer. This work will strongly benefit from them.

**Scientific Comments:**

Title: I do not think the title really gives an indication of what the paper is about. In particular, the authors do not consider periodicity in their model and it is not clear (at least to me) what free oscillations refers to. I think heat transfer should at least be mentioned.

Our model builds upon MacAyeal (1993a). In particular, we are exploring the theoretical periodicity of a binge-purge oscillator (MacAyeal, 1993a, b). Note that the solution provided by eqs 5-7 describes the evolution of the temperature profile and does not show a periodic behaviour in itself; the periodic behaviour emerges once this solution is considered within MacAyeal's oscillator.

As stated in the introduction of this manuscript: "Free oscillations were first proposed in MacAyeal (1993a) as manifestations of the Laurentide Ice Sheet (LIS) purging excess ice volume. This interpretation rests on the assumption that a transition exists between two potential states of basal lubrication (Alley and Whillans, 1991; Hughes, 1992) and it is known as the binge-purge hypothesis. Namely, when the basal ice temperature is below the pressure melting point, the ice sheet is assumed to be stagnant and it simply thickens due to snow accumulation. As a result of the geothermal heat flow, the ice column is expected to warm and the base eventually yields melting. At this point, the ice sheet is no longer at rest and begins to slide over a lubricated sediment bed".

Line 4: "general boundary condition problem": this is a stretch. The authors consider a

Robin boundary condition at the surface, which is more general than a Dirichlet or Neumann BC, but certainly not every BC can be expressed in this form.

This is a fair point, the manuscript will be changed to reflect it.

Line 7: "depends on several factor: the ice column thickness L...". The time required depends on more than just these things (as the authors themselves show), e.g. the diffusivity. The phrasing makes it sounds as though the time depends only on L, the initial temp profile and the BC.

We agree. This sentence will be changed accordingly.

Line 14: This may be my own ignorance, but clarification on what the authors mean by "thermomechanical instabilities" would be appreciated (this phrase appears frequently throughout the ms, and I was unable to infer its meaning from the context in any place).

A clarification will be included for self consistency in the abstract. Besides, an additional paragraph will be expanded in the conclusion section.

Line 16: The solutions presented in this paper are not, to my mind, analytic solutions. While I agree that the form of the solution has been expressed analytically, the eigenvalues must be determined numerically (as the authors point out) and the infinite sum presumably is also evaluated numerically. This should be clarified and the many statements about deriving analytic solutions should be adjusted to reflect.

It is true that eigenvalues are given by a transcendental equation in our particular case and must be evaluated numerically. Nevertheless, the technique is analytical and the solution is presented as a convergent infinite series (considered an analytical expression). Truncation is only needed for visualisation purposes.

Following the definition, a closed-form expression uses a finite number of standard operations. The analytical expression is slightly less restrictive and allows for an infinite number provided that convergence is ensured. Our solutions fall within the latter, since they are expressed as a convergent infinite series. The eigenvalues may be evaluated numerically as it is the case of the Bessel functions where the zero-th of such function must be computed (e.g., Dirichlet laplacian in a circle), yet they are still considered analytical solutions.

Line 36-37: can you elaborate on the statement "since it would be prohibited by the ice sheet heat transfer physics". I guess you mean that if the ice sheet is sufficiently thick, the base is insulated from the surface?

Indeed, the reviewer is right. This statement is further elaborated on in the sentences thereafter, but it will be rephrased to avoid confusion.

Line 42: need to explain what the low order model is — an ice sheet model? Here we are referring to MacAyeal (1993b)'s model. It is in fact a 2D conceptual model of the Laurentide Ice Sheet built to verify the accuracy of the predicted periodicity of the binge-purge oscillator (MacAyeal, 1993a). Ice flow mechanics and mass balance are combined in a manner that yields null horizontal ice flow when the base is frozen whereas deforming sediments allow for rapid sliding when the base is melted.

Our current work further expands the theoretical prediction of MacAyeal (1993a) by considering a more realistic domain (a finite ice column rather than a semi-inifinite domain), yet it builds upon the binge-purge oscillator hypothesis.

Line 57: what does a 'unique spatial element' mean?

Here we refer to the absence of spatial dimensions, quoting Robel et al. (2013): "single lumped spatial element". This will be further clarified in the text.

Line 57: "the surface temperature and the geothermal heat flux were found to determine the character of the ice flow": I don't think that's true: these two factors are important controls, but don't together determine the entire ice flow.

Robel et al. (2013) precisely wrote "In our simple model, we find that geothermal heat flux and surface temperature control vertical basal heat budget and determine the character of the ice flow, of which we found two potential modes of behaviour." Of course, these factors alone do not determine the entire flow, but they constrain it so that it is enough for our purposes.

Line 61: what is a zero dimensional spatial model? One spatial dimension?

No, there is one single lumped spatial element so that we cannot talk about "spatial dimension". This comment referred specifically to Robel et al. (2013); this will be made more explicit in the text.

Line 80: I disagree that this is the most general approach, what if there is e.g. precipitation at the surface which transfers heat?

The reviewer is right. The text will be changed accordingly.

Line 86: I think it would be useful to explain the physical interpretation of $\beta$. What is a typical value? This would help a lot with interpretation of the numerical solutions (see below). The authors say it "modulates the permissible deviation between ice and air temps", which I agree with, but I don't think adds anything beyond what the equation says directly.

The authors thank the reviewer for such a comment. Additional text will be added to link this to the physics of the ice column.

Line 90: "the ice surface will consequently evolve in time towards $T_{\mathrm{air}}$" - I don't think that's true? e.g. in the limit of a very short ice column, the heat flux at the surface will match that at the base (i.e. $-G/k$) and thus the surface temp will evolve towards $T = T_{\mathrm{air}} + \beta G/k$? More generally, it is possible to have a heat flux out of the domain while the system is in steady state.

We are grateful for this comment. The reviewer is right.

Line 92: This formulation assumes that the diffusivity is constant, which should be stated. Is that a reasonable assumption to make? I am not an expert, but it seems like the firn layer at the surface might have a vastly different diffusivity to the rest of the column.

We could include a deviation of the thermal diffusivity for a range of temperatures (e.g., Cuffey, Greve. . . ). However, for the purposes of this work, this is well known and a fair approximation. In the firn, the density approaches zero as we reach the top and so does the diffusivity. Nevertheless, we are not explicitly considering the firn layer above the ice. A statement on this simplification will be added in the text.

Line 93: (Key): where does this initial profile come from? Why is it appropriate? It is simply introduced with no justification! The authors go on to show that the time to melting is sensitive to the particular linear profile chosen, so presumably it is also sensitive to the type of profile. Shouldn't the initial condition be set by the temperature profile immediately after an event in the binge purge cycle?

The authors should also make it clear that this initial profile is something they impose on the model. In addition, surely the initial profile is only compatible with the BC for specific values of $\theta_L$ and $\theta_b$, related to G and k etc? The linear profile is introduced for simplicity. It allows us to explicitly determine the impact of the initial basal/surface ice temperature independently.

Indeed, this profile is imposed as the initial condition of the model and then, a broad range of $\theta_b$ and $\theta_L$ values is explored. Ideally, the initial condition should be set by the temperature profile immediately after an event in the binge purge cycle, yet such a profile is not available. A linear profile assumes that the temperature in the ice reflects a linear lapse rate in the atmosphere as the ice thickness builds up over time. The manuscript will be updated to account for this comment.

General comment on modelling: I am not sure why the authors did not non-dimensionalize their model. That seems like a straightforward way to reduce the number of parameters in the system, as well as gain insight into which processes are important in which locations. Reformulating the system in terms of dimensionless variables might help with the lack of generality of the numerical results which I point to later in my review.

We thank the reviewer for such a comment. Non-dimensionalization provides some clarity, but the results will not change. We believe the equations are simple enough that keeping dimensionality will help the reader.

I also found it quite surprising that the authors did not even mention the heat equation by name.

The heat equation is named in several places throughout the text.

Line 96: I would not call this method "separation of variables", which I understand to be more general, and others may also have this confusion. The authors might consider removing mention of separation of variables (does it add anything?).

As the reviewer later includes some notes in the standard derivation (e.g., `https://courses.maths.ox.ac.uk/pluginfile.php/22143/mod_resource/content/1/FS-PDE-Slides-Week-4.pdf`), these are also referred to as "separable solutions". Other references also referred to this as separation of variables (e.g., Arfken, G. "Separation of Variables" and "Separation of Variables–Ordinary Differential Equations." §2.6 and §8.3 in Mathematical Methods for Physicists, 3rd ed. Orlando, FL: Academic Press, pp. 111-117 and 448-451, 1985.)

Line 112: the authors mention using a numerical method, but provide no link to code required to solve equations or produce figures. Code and data should be held in (e.g.) an open repository.

We thank the reviewer for such a comment. A github repository will be created where all scripts will be uploaded.

Line 113: what does the tolerance refer to here?

This is a root-finding algorithm. In our particular case, it is applied to the eigenvalue equation where we are solving for $\lambda_n$. This iterative algorithm continues until numerical convergence is ensured by setting a certain tolerance, as is standard procedure (`https://rdrr.io/rforge/pracma/man/brentdekker.html`).

Line 112-115: see above: this equation requires a numerical method, so the solution is not analytic!

A closed-form expression uses a finite number of standard operations. The analytical expression is slightly less restrictive and allows for an infinite number provided that convergence is ensured. Our solutions fall within the latter, since they are expressed as a convergent infinite series. The eigenvalues may be evaluated numerically as it is the case of the Bessel functions where the zero-th of such function must be computed (e.g., Dirichlet laplacian in a circle), yet they are still considered analytical solutions.

Line 122: what is $\tilde{G}$?

Yes, this is a typo, it should be just $G$.

General note on appendices: Appendix A is entirely standard (e.g. `https://courses.`
`maths.ox.ac.uk/pluginfile.php/22143/mod_resource/content/1/FS-PDE-Slides-Week-4.`
`pdf`) and does not need to be included (a reference to a textbook would suffice). Appendices
B and C could also be described as standard, but could also be quite easily incorporated
into the main text.

We thank the reviewer for such a comment. A standard reference will be included. We
prefer to keep Appendices B and C separate from the main text to make the main message
clear and leave the mathematical derivation/subtleties aside for the interested reader. We
will cite the standard reference, but prefer to keep Appendix A, so that the method is clear
without further searching.

Figure 2: need to state the values of kappa, $T_{\text{air}}$, $k$, $G$ (etc?) used to generate these figures.
Also, there is no need for a legend in both figures, and the final legend entry is not necessary
(imo). Labels (a) and (b) are missing from the panels.

We agree, the manuscript will be changed accordingly.

The solutions are infinite sums, which must naturally be truncated somewhere. The authors
should discuss their choice of truncation and justify that the terms ignored are not important
(e.g. by considering their order of magnitude).

The reviewer is right. We actually kept more than 60 terms in the series, but never mentioned
it in the text. From what we have seen, if $n = 13$ terms, the error reads:

$$\frac{||u_n - u_{n-1}||}{||u_{n-1}||} \leq 0.03\% \tag{1}$$

where $u_n$ denotes the temperature solution truncated at order $n$.

Line 125: "the second time frame is chosen...": it would be nice to explain to the reader
what they are looking for in the figure to show this (I think it is the blue line going through
zero?)

Yes, this needs a clearer explanation and the manuscript will be changed accordingly.

Important question: based on the previous comment, it seems that the authors are setting the freezing point (on which their timescale calculations are based) to zero? Is that correct? If so, why is there no pressure dependence in melting? Including pressure dependence would mean that larger columns don't need to get to as high a basal temperature in order to start melting, making the dependence of the timescale on $L$ stronger, I think.

We thank the reviewer for this comment. Indeed, including a pressure dependence would make the dependence on L stronger, though changes will be generally small. Quantitatively, it follows that $\tilde{\theta} = \theta + \alpha P$, where $\alpha = 9.8^{-8}$ K/Pa. Knowing that $P = \rho g L$ and $L = [1.0, 3.5]$. If $\Delta L \simeq 3.0$ km, then $\Delta \theta \simeq 2.622$ ºC.

However, this dependency needs to be omitted in the comparison section with MacAyeal (1993a) since it was therein disregarded (if we want to check convergence to the 7000-yr periodicity as $L \to \infty$).

Line 128: "this rate is in fact proportional to...". I think you have to work a bit harder to get to this. I agree that each mode has the quoted rate, but it is not obvious what happens when only takes the sum over all the modes (the value quoted is dependent of the dummy variable $n$!).

The reviewer is right. We here meant the behaviour at leading order. We will elaborate on this more rigorously.

Line 128-129: what are $L_1$ and $L_2$? I guess you are saying that if we have two columns of lengths $L_1$ and $L_2$, then the temp gradient at the base is larger in the longer one? If so, I don't think you can make this statement based on the rate argument made previously (see above comment). In any case, it is not obvious that the rate is increasing in $L$, particularly since $\lambda_n$ are complicated functions of $L$.

We agree that it is not obvious, this is why it was explicitly worded. However, we do think that the reasoning is sufficient even if $\lambda_n$ is a complicated function. REVISE THIS!!!!!!!

Line 131-132: "the specific non-zero $\beta$ does not alter this behaviour" - this requires justification, you have only shown the results for one non zero beta (and one value of $T_{\mathrm{air}}$, $\kappa$, etc).

I actually proved (mathematically) that solutions are identical in the limit $L \to \infty$ for an arbitrary $\beta$. I kept everything else general so it should work in my opinion. I'll go over the derivation again though.

Line 134: so beta is the lengthscale over which the ice column feels the surface? This should be stated earlier.

Yes, the manuscript will be changed to make this more clear.

Is the value of $\beta = 50$ m large or not? More generally, is the close agreement between the curves for $\beta = 0$ and $\beta = 50$ because the system is not sensitive to beta, or because of the particular value of beta chosen? Non-dimensionalizing the system would help with this.

This is a great point. To the authors' knowledge, we have not seen any $\beta$ values in the literature since nobody has addressed the finite problem analytically. It is in fact sensitive to $\beta$ if I give larger values, but 50 m is probably "small". We will add some discussion of the choice of $\beta$ and its impact.

Line 142: "it is clear that the column thickness is the primary factor. . .". I don't think you can make as strong a statement as this based on the figure. It is clear that the system is sensitive to L, but it is not clear that this is the most important factor (is it even possible to rank importance of factors?), and this figure is only for one particular set of $T_{\mathrm{air}}$, $\kappa$ etc.

We agree, the text will be changed accordingly.

Line 148: ". . . solely depends on $L$, $G$, $T_{\mathrm{air}}$". That can't be true! You even said yourself that the rate depends on kappa (line 127: "in fact proportional to $\kappa \lambda_n e^{-\kappa \lambda_n t}$". There are also potentially other constants in the prefactor.

The sentence will be modified to account for additional dependencies.

Line 151: "dependency. . . on both the initial and boundary conditions". This is crucial: there is a very strong dependence on the initial conditions, which the authors have not justified. Why are they a sensible choice physically? Why is the timescale determined as the time

from this (seemingly arbitrary) initial profile?

The referee suggested using a profile right after an event, but such a temperature profile is unknown. Yet it is interesting to justify why the linear profile is a sensible choice: it is in fact the simplest scenario where base and surface temperatures may differ and their impact can be independently quantified. The timescale was defined by MacAyeal's binge-purge oscillator (MacAyeal, 1993a). Since he assumed a semi-infinite domain, his initial temperature profile is constant along the column and reads -10 ºC (a linear case with identical base and surface temperature).

Line 153: "though solely for ice thicknesses below 2km": the figure only shows this for this particular set of beta, $T_{\mathrm{air}}$ etc. The following statement ("we therefore find L = 2km is a threshold value...") has not been shown in general. How does figure 4 change for different parameter values is not explained or pursued.

This is true. I haven't proved it mathematically. Maybe we need a more detailed discussion of Fig 4 to avoid this comment?

Figure 4: caption: this is picky, but really $\theta_L$ and $\theta_b$ are parameters rather than boundary conditions. You should also state the values of all parameters used to create these figures. 'Initial basal temperature' in the final line should not be capitalized.

We agree that they are not boundary conditions but it should be noted in Figure 4 we refer to them (panels c, d) as "Initial conditions". Boundary conditions refer solely to panels a and b. This will be modified accordingly.

Figure 4: I find it surprising that the timescale T is large for small $\theta_b$ for $L < 1.25$ or so. Is this because the geothermal heat flux is too strong? Is there a limit in which the base never reaches thawing, or does it always happen just on longer and longer timescales?

If the boundary conditions are such that the surface is cold enough (here, $T_{\mathrm{air}} = -25°C$) and the column is thin enough the base will never melt. This means that the temperature at the base does not reach 0°C at infinite times but may rather cool down. The manuscript will be updated to avoid confusion.

This figure is very interesting but not really discussed. It raises many questions, e.g. the

non-monotonic behaviour in (d) is also quite interesting, why is that? Also, there is a very sharp transition in (b): why is this transition so sharp (what sets this boundary)?

The question regarding the monotonic behaviour was also asked by Referee 1 and will be thoroughly included in the manuscript. Briefly, the mechanism is the following:

This is a quite complex behaviour since there are several factors that must be considered simultaneously. It is illustrative to look at the vertical profiles shown in Fig. 2. The fact that the temperature appears to be non-monotonic with $L$ at $-14C$ is a consequence of two factors: the necessary energy budget to warm an ice column and the vertical temperature gradient. The former increases with $L$, whereas the latter decreases with $L$.

For slight variations of the thickness $\delta L$ near $L = 1.5$ km (while fixing $T = -14C$), the time required to thaw the base is larger regardless of its sign. In other words, it takes longer to reach the pressure melting point both for a thinner and a thicker column. This local minima is a balance between the total energy necessary to heat a column and the fact that a thinner one implies a larger vertical gradient for fixed temperature difference between the base and the top. Namely, we could consider the effect of these factors explicitly. First, a thinner column requires a smaller amount of total energy to increase the temperature of the column. However, considering the second factor, a thinner column would yield a larger vertical temperature gradient (ultimately yielding a slowdown in the warming rate as the geothermal heat flux is fixed in the BC). The combination of both effects allows for local minima.

Concerning the sharp transition, we could understand the result knowing that at some point, the geothermal heat flux is not enough to sustain the amount of heat being lost at the surface (while warming the base). If we cross that threshold, the base will no longer reach melting.

Line 180: this is absolutely not clear, and took me about 10 minutes to figure out that the right hand side of the square bracket goes to zero, so you're looking for roots of tan(theta)=0. This should be explained in more detail.

We agree that this may need more explanation, which will be provided in the revised manuscript.

Line 185-186: you have shown that the results are asymptotically equivalent, rather than identical (they differ by some small amount for any finite L).

We completely agree. We were not fully mathematically rigorous here. This will be rephrased.

Line 190: so MacAyeal considered periodic forcing in semi infinite domain, and this work looks at a finite length domain with non-periodic forcing. Can the two changes (periodic vs non periodic and finite vs semi infinite) be decoupled?

MacAyeal actually dismissed external (atmospheric) periodic forcing as the source of the oscillation. Here we are not saying otherwise, we are saying that in the binge-purge oscillator a non-period forcing with a more realistic (finite thickness) description of the system would yield a broad range of periodicities for the same set of parameters that MacAyeal used.

Line 192: Again, the strength is not determined solely by L, beta will also be important (for example).

Yes, this will be rephrased for clarity.

Line 194: It would be helpful to make it clear that the new development is the consideration of a finite length domain.

Yes, we will make this more explicit in the revised manuscript.

Line 195: again, I disagree that these solutions are analytic, for the reasons outlined above. 'Analytic approach' also mentioned on line 205.

We discussed this in the comments above, and prefer to stick with the terminology as used.

[A closed-form expression uses a finite number of standard operations. The analytical expression is slightly less restrictive and allows for an infinite number provided that convergence is ensured. Our solutions fall within the latter, since they're expressed as a convergent infinite series. The eigenvalues may be evaluated numerically as it is the case of the Bessel functions where the zero-th of such function must be computed (e.g., Dirichlet laplacian in a circle), yet they are still considered analytical solutions.]

Line 198: again, I don't think this is a general boundary condition considered. It's a Robin type (as discussed above).

We agree, it is rather a more general approach. The text will be changed.

Line 202: You actually showed something stronger: you showed that the semi-infinite domain is an oversimplification provided that the column length is not large (within the context of the assumptions made).

The authors thank the referee for bringing a positive statement that was overlooked while writing the manuscript. This will be expressed clearly in the conclusions.

Figure 5: this figure should appear earlier in the text. It also has two black solid lines. Again, fixed parameter values should be stated. $\tilde{G}$ appears again here. The behaviour of the blue and black solid lines in particular suggests that there is a critical value of the ice thickness below which the potential periodicity goes to infinity (as hinted at by figure 4). Is this the case?

This figure is just a particular case of our problem in which we set those values of MacAyeal (1993). This is now precisely stated in the caption. And yes, it approaches infinity below a critical ice thickness value.

Line 207: It was certainly not shown analytically that $L = 2$ km is an upper bound! That was a numerical result and there is no proof that this value holds in general (numerically or otherwise).

It's true tha the particular threshold value is in fact parameter dependent, though we also expect its existence for a distinct choice. The manuscript will be updated to reflect this.

Line 215: what is the estimate of a "medium" size based on?

An explicit quantification of the will be given (i.e., 1-4 km).

Line 216: "the explicit consideration of distinct initial temperature profile manifests a high sensitivity. . . to its initial state". Indeed. . . see earlier comments on the initial condition. Why is the initial conditions correct, or even sensible? And why is the timescale the time from this initial condition to thawing?

The timescale was defined by MacAyeal (1993a). As stated before, we built upon his binge-purge oscillator, in which the initial temperature followed an atmospheric lapse rate (since his domain was semi-infinite, setting a particular surface temperature was not possible). Our approach is more general: we keep the linear dependence of a lapse rate, though we can explicitly set a particular initial surface temperature.

Line 222-223: "the periodicity of such events cannot be imposed by the frequency of an external forcing". How do the authors come to this conclusion? There is no periodicity in their model.

We build upon MacAyeal (1993a), who determined the e-fold decay. The periodicity in that study results from his conceptual model and is numerically demonstrated in MacAyeal (1993b). We here extend the implications for the timescale required for the base to reach thawing to a finite column. In that sense, the rest of the model would remain identical.

Line 227: "this double fold nature....is considered": I don't understand what the authors mean by thermomechanical instabilities here. Such things have not been discussed in the paper. Or is this a result from elsewhere? If so, what is the relevance?

A clarification will be included and its relevance related to the binge-purge oscillator.

While I appreciate that the authors intend to improve upon the work of MacAyeal (1993) by considering more general conditions, it would also be nice to discuss the many simplifications made on a real ice ice sheet, of which there are many: heat transfer is three dimensional in practice, ice columns do not all have uniform length etc.

This is in fact a caveat of both MacAyeal (1993a, b) and our current work. The problem is approached by using analytical techniques and so the complexity of the system is critical if a solution is to be found. A better discussion of the simplifications will be included in the manuscript, which was also requested by other reviewers. Nevertheless, the simplicity of our work provides new insight from a theoretical perspective.

Let us note that we are not claiming this is the mechanism underlying HEs. What we are saying is that, if it were so, the 7 kyr periodicity would be tightly linked to the size of the ice sheet, something that has been ignored even by all 3D ice-sheet models. Our results could be interesting for those too.

**Technical Corrections:**

Line 13: "these results ultimately manifest a": results are not active, they do not manifest things. "These results suggest/show"?

Thank you, we will rephrase this as suggested.

Line 35: need a reference for "appear to be dependent on environmental factors"

Thank you, we will include a reference to MacAyeal (1993a).

Line 38-39: the sentence "In fact..., for a motionless ice column" is very confusing to anyone who has not read the MacAyeal paper in question. Could the authors clarify? Do they mean that MacAyeal showed that a periodicity of $T = 7000$ yrs corresponds to an e-folding decay length of 14m?

Yes, MacAyeal showed that such a surface signal would have a 314 m e-fold decay. As a result, such periodicity can not be imposed by an external forcing. The text will be modified to avoid confusion.

Line 50: SIA has not yet been defined.

SIA stands for Shallow Shelf Approximation. The text will explicitly include it.

Line 52: causes -> caused

This typo will be corrected.

Line 53: Is this comment 'thus'? I don't think it necessarily immediately follows from the examples above...

This will be changed.

Line 72-75: this description of the paper's structure doesn't actually match what appears in the following (e.g. section 7 has the limit L → ∞)

This will be changed to improve consistency.

Figure 1: I don't see why it is necessary to show the x axis? The problem is one dimensional.

This is just a conceptual view of the system to facilitate visualisation of the processes involved. It will be explicitly stated in the caption that the problem is one dimensional to avoid potential confusion.

Figure 1 caption: specify that non-equilibrium thermal states refers to equilibrium across the ice-air interface.

We will add this.

Line 84: "as a result": I don't think this sentence is a direct consequence of the previous one.

This will be changed.

Line 90: "Thus": this is not appropriate: the heat equation doesn't follow from the description of the BC.

We will change the text accordingly.

Line 98: there is a backwards quote mark. Also, what is 'shifting the data' an example of?

First is a typo. Shifting the data is an example of a change of variable that leaves our initial boundary problem with homogeneous boundary conditions so that the corresponding eigenvalue problem can be solved. This will be clarified.

Line 108: to my mind, that is an initial boundary value problem, rather than just a boundary

Yes, we will make this more clear in the revised text.

Line 127: I would avoid making statements like "the implications of a finite domain are clear"... It was not clear to me at this point! Your job is to explain the implications.

We thank the reviewer for such a comment. We will modify the text as suggested.

Line 140: referred as -> referred to as

This will be changed.

Line 171: "excess" has a backwards quote mark at its beginning

This will be changed.

Line 222: which events does 'such events' refer to?

We refer to the Heinrich Events (HE) as previously stated throughout the manuscript. We will modify this sentence to avoid confusion.

---

## Author Comment (AC3)

**Authors final response to Reviewer 3**
**TC-2022-97**

Daniel Moreno, Alexander Robinson, Marisa Montoya, and Jorge Alvarez-Solas.

September 20, 2022

Note: Reviewers comments are given in blue font whereas the author response reads in black.

Recommendation: This manuscript, in anything like its current form, does not seem to contain a publishable idea. The most generous interpretation is that other researchers, over decades of analysis of temperature conduction in a solid rod, have failed to notice an intrinsic timescale which might relate to ice sheet binge-purge cycles. If that is so, something this reader thinks is not true, then the way the article is written must be completely redone. Critically, issues of incoherent definition ("potential periodicity" is here meaningless) and essentially-disregarded parameter dependence (the assumed initial basal temperature and geothermal rates are in fact dominant) must be somehow overcome. (It would be a different paper if so.) In any case, the many time scales potentially associated to full, physically-clear binge-purge mechanisms must be carefully considered if the claimed special time scale here is to be taken seriously.

It is clear from this review that our intended message has not come across clearly in the text, and this is something we will aim to improve in the revised manuscript. In short, we believe that our intended message is actually quite consistent with what the reviewer expects and the rather negative comments largely stem from misunderstanding of that message. We hope that addressing these comments will clarify any misconceptions and help us improve the manuscript. First, we would like to respond to the comments from this first paragraph briefly:

- *The apparent fact that the paper does not contain publishable ideas.* This statement is in disagreement with the other two referees. Referee 2 even considered the paper

as "interesting" and "appreciated the simplicity of the approach and use of analytic techniques". We believe that this statement results from a thorough misinterpretation of our intended message. We will ensure that our message is much more clear in the revised text.

- *Incoherent definitions (e.g., "potential periodicity").* Our paper is framed within the binge-purge oscillator framework, even though it reaches far beyond it for its generality and simplicity. This term merely emphasises the fact that it is not a pure oscillation, but rather relates to the "binge" phase timescale that would yield a binge-purge oscillator, as defined in MacAyeal (1993a) model. It is clear that this phraseology gives the reader a false impression of our aims, and we will maintain the more accurate phrase "time to reach the pressure melting point" whenever possible.

- *Disregarded parameter dependency.* Parameter dependency is not disregarded - in fact, one of our main aims is to show precisely that there is no special timescale intrinsic to the system, but rather that it generally depends quite strongly on the boundary and initial conditions. The precise 6944-year value arises only when employing the exact parameter values as MacAyeal (1993) and we use this to show that our system behaves consistently when considering the same (over)simplification. Otherwise, as Fig. 4 shows, we have exposed a broad range of response timescales that depend very much on the choice of several parameter values.

Summary of the manuscript: The Introduction ties binge-purge (Heinrich event) cycles to ice temperature (which is fine) and concludes by asserting that 7ka periodicity is widely used in the literature. Section 2 sets up an initial-boundary value problem for a motionless ice column of finite length, with geothermal (Neumann) basal and Robin surface boundary conditions, and linear-in-height initial temperature. Sections 3 and 4 sketch, with details in the Appendices, a Fourier series solution of the problem, in which (generally) the eigenvalues solve a transcendental equation requiring numerical solution. Section 5 visualizes the temperature profiles and their time-dependence, with an emphasis on how they depend on the ice thickness L and on beta, an insulation coefficient in the surface Robin condition. Section 6 starts by defining a certain solution time as "potential periodicity"—there is no given justification for connecting *this* solution time to periodicity!—and then illustrates and discusses dependence of this time on parameters. Section 7 then focuses on the dependence of the time on $L$, as $L$ becomes large, revealing a time 6944a in the limit. (This value, conveniently near 7ka, entirely depends on the assumed conditions at the base, namely the initial basal temperature $\theta_b$ and the geothermal rate $G/k$.) Finally the Conclusion again emphasizes the role of $L$. Appendices then give details of the standard Fourier series analysis.

The connection to periodicity is justified by the definition in MacAyeal (1993a) as the time required to thaw the base (Sections 4.2 and 4.3, MacAyeal, 1993a). For a binge-purge oscillator, as expected, the 6944-yr limit depends on the conditions at the base (since it is an initial boundary problem) and MacAyeal (1993a) identical values are employed in that section for a one-to-one comparison. Our results now show that for a finite domain there is an additional dependence on the ice sheet thickness (Fig. 5) that did not appear in MacAyeal's original papers since the heat solution assumed a semi-infinite domain. As mentioned above, we will generally use the phrase "time to reach the pressure melting point" to avoid confusion.

**Major concerns:**

Understanding the consequences of conservation of energy in ice sheets is a nontrivial matter, thus it is included as a 3D partial differential equation into most modern ice sheet modeling efforts, and it is important because internal energy (e.g. temperature) is tied to the long time-scales at which ice sheets change. Because ice sheets are thin, variations in the vertical are generally larger than in the horizontal, but nonetheless the problem is advection-dominated. In ice columns near the divide the strongest direction of ice advection is typically vertical, but over large areas of ice sheets this direction is horizontal so that column-wise temperature distributions are commonly far from what any isolated vertical-column model might generate. Furthermore the bases of ice sheets are usually near or at the pressure-melting point. The thermo-mechanical condition of near-basal ice can dominate overall ice sheet dynamics because the presence of pressurized liquid water facilitates ice deformation and basal sliding. The near-basal thermal regime is dominated by geothermal flux, dissipation heat from sliding, and at times the transport of liquid water from elsewhere (e.g. ice surface or through subglacial hydrology). Because of the strong role of liquid water, it follows that conservation of energy is a two-phase problem, thus not one which can be well-modeled by temperature alone.

We agree with the conservation of energy reasoning. And it is clear that a more realistic and sophisticated description of the thermomechanical processes at the base of an ice sheet is possible and is employed by 3D ice sheet models. However, conceptual studies also have great value in helping to understand the importance of different processes, and mathematical simplicity allows for analytical solutions that facilitate the analysis. The main aim of our paper is to reevaluate an important foundational piece of literature in the binge-purge hypothesis (MacAyeal, 1993a, b) and advance our understanding of how the thickness of an ice sheet influences its thermal evolution. The context of the problem here relates to a region where the ice is initially frozen to the bed. Thus horizontal advection can be expected to be low, and there should be no liquid water at the base. The question addressed is how long would it take such a column of ice to reach the pressure melting point. The subsequent evolution of the ice sheet would indeed be more complex, but is outside the scope of this simple scenario. It can be argued, as indeed MacAyeal (1993a, b) did, that this initial time to reach

the pressure melting point is related to the binge timescale of the binge-purge mechanism. We also note that the solutions calculated here are furthermore not restricted to a particular problem and can be used in any physical system that satisfies the initial boundary problem.

The current manuscript considers none of these realities, nor does it provide this reader any insights about ice sheet thermodynamics. Instead it examines a conduction-only isolated column model. Within this narrow, unpromising model it proceeds to ignore the dominant parameter dependencies and instead extract a special 7ka time scale, a time scale for temperature change at the base, by surreptitiously fixing some dominant, but unexamined, values. Then it confusingly discusses dependence on less-dominant parameters, especially ice thickness L and surface conduction beta, simultaneously arguing that L is important and irrelevant.

As mentioned above, the aim of the paper is not to provide a full description of all processes concerning the energy conservation within an ice sheet. It is well known that there exists no analytical solution to describe such a system. Nonetheless, parameter dependencies are not ignored in our description. They are considered in Fig. 4, where a broad range of values are employed to compute the time required for the base to thaw (i.e., periodicity, as defined by MacAyeal, 1993a). Moreover, ice thickness is never simultaneously argued as important and irrelevant. A careful look at the paper reveals the subtleties of such degrees of freedom (even in this idealised system).

Thus the manuscript first fails to consider the actual thermodynamics of ice sheets, and then it makes unreasonable claims for the relevance of its very-simplified model. An extremely well-trod mathematical analysis, namely Fourier series applied to conduction in an interval, a problem already addressed by Fourier and Kelvin, is offered as new and insightful, which it is not. The modeled time evolution of a column's basal temperature profile simply does depend strongly on the column thickness L, despite the "strongly dependent" claim in the abstract (line 5). The particular 7ka time scale revealed herein, and unconvincingly tied to binge-purge oscillations and Heinrich events, actually does have strong dependence on particular basal parameters in the model, namely the assumed geothermal flux rate and initial basal temperature. However, this special time scale would in any case be destroyed by any (here missing) advection mechanism including sliding, critical to any serious discussion of binge-purge.

Fourier analysis is not presented as new, but rather as a standard approach (appendices were only included for clarity with readers not familiar with it and will be deleted considering the comments of the other two referees). Yet, to the authors' knowledge, this method has not

been applied by the glaciological community to address the current problem. In particular, MacAyeal (1993a) did not use it but instead resorted to considering an infinite domain to simplify calculations, arguing this could be justified. We here show that considering a finite domain leads to a dependency on ice thickness that is not present in MacAyeal's solution. In addition, we demonstrate that the time scale also depends on the initial and boundary condition of this problem as expected (but ignored in the original work). As posed by MacAyeal (1993a), no horizontal advection is considered in this problem, though vertical advection is neglected by estimating the e-fold decay of a sinusoidal signal at the surface with a constant vertical velocity comparable to the accumulation rate at the summit of the GIS (e.g., Alley et al., 1993).

A key sentence (lines 138-140) is that "We further calculate the time required for the column base to reach the melting point ..., hereinafter referred as potential periodicity". There is no offered justification for why this solution time is a "periodicity" for anything! Indeed binge-purge is a periodic mechanism, one of great interest and importance, but there is not even an attempt to explain why this time is related to the desired periodicity.

There is clear justification in Sections 4.1, 4.2 and 4.3 (MacAyeal, 1993a): "once the basal temperature reaches the melting point, the ice sheet begins to move". This is in fact the end of the growth phase (binge) of the cycle. We had assumed that the ideas presented in such a paper are known and fully understood by the reader, but we will explicitly address this to make our link to that work more clear. The 6944-yr periodicity is then elaborated in Section 5.0 of the same paper.

This "potential periodicity" time is completely dependent on a parameter which is completely arbitrary, namely $\theta_b = -10°$ C as the starting point at time 0. It also depends strongly on the geothermal flux rate, which is known to vary substantially over a continent. (Geothermal flux rates are available for modern North America and thus could be used to explore this parameter dependence.) As shown in Figure 4(d), stably across a broad range of ice thicknesses $L$, variation of $\theta_b$ from -15C to -5C implies "potential periodicity" which ranges from about 4ka to about 20ka. Lines 161-162 actually mention this but the rest of the manuscript drops it: "the potential periodicity appears to be rather sensitive to the initial basal temperature, rapidly saturating to values above 25 kyr for $\theta_b < -11$ C". Attempting to interpret time scales as depending on L seems to deliberately ignore that they depend much more strongly on an uninspected parameters $\theta_b$ and $G/k$. Possibly $\theta_b$ should be regarded here as a proxy for the coldness of the cold part of the atmospheric-driver temperature cycles, but (as far as I can tell) even this is not argued.

Indeed, the timescale to reach the pressure melting point depends on $\theta_b$ (already noted in

MacAyeal, 1993a). Nowhere in the paper is the contrary stated and, the fact there exist additional dependencies (e.g., $L$, $\beta$, $G$, $k$...), does not say otherwise. We first explore a broad range of $\theta_b$ values (Fig. 4d), so we do not fully understand why the referee stated that this parameter had been ignored. Then, to perform a one-to-one comparison with MacAyeal's 6944-yr estimation, we employed an identical value of $\theta_b = -10°$C as we did in our Section 7. Additionally, geothermal heat flux values span those available for North America, thus exploring a realistic range. Likewise, $\theta_L$ is not ignored and presented in Fig. 4c.

Finally I want to describe two important figures, so as to illustrate the inappropriateness of the manuscript's analysis. Figure 4: What the parts of this Figure actually show, though this is ignored, is that the strongest dependence of the "potential periodicity" time is on the geothermal flux rate and the initial basal temperature. The discussion of dependence on air temperature and ice thickness is mostly a distraction.

As expected, there exists a dependence on both the geothermal heat flux and the initial basal temperature. Nevertheless, Fig 4b is enough to understand that the air temperature and ice thickness are of paramount importance. It is quite interesting to notice the sharp dependency with thickness for air temperatures below -20ºC. Even more, all panels in Fig. 4 share the x-axis where the strong L-dependency is clearly shown. These dependencies will be further elaborated in the text for completeness.

Figure 5: Here is my attempt to say what is shown in this Figure; note that Figure 2b in particular supports my interpretation. A geothermal rate and ice conductivity are fixed, giving a fixed value G/k. An initial basal temperature ($\theta_b$) is fixed, most likely as -10C consistently with Figures 2a and 3, though its value is unstated. Then the time for the base to warm to 0C (the mis-named "potential periodicity") is shown as a function of ice thickness $L$. Different surface boundary condition treatments give several curves, but for $L > 2.5$ km they all coincide at a time about 7ka. I observe that the explanation for this value of 7ka is actually quite clear! Namely, as long as the top of the ice is far away, the chosen values of the initial basal temperature and the geothermal flux rate will determine the time taken for the base of the ice to warm up to 0C; this is a balance of upward conduction with the delivered heat in the time interval. Thus the special value 7ka is actually (and strongly, and entirely as L goes to infinity) a function of $\theta_b$, $G$, and $k$, which were all fixed at certain values for no stated reason. This dependence should be examined, but instead the paper looks elsewhere, at L and beta, and then it spins the results as related to Heinrich events.

In the manuscript, our intention was essentially to give the conclusion of the reviewer here. The point is that the widely cited value of 7ka is not special at all. Indeed, as $L$ goes to infinity the time required to reach melting is a function of $\theta_b$, $G$ and $k$. In this section,

the values used here were chosen to replicate those of MacAyeal (1993) and in that way show that our solution converges to his in such a limit. Meanwhile, the dependence of the timescale on various parameters has already been examined in our more realistic analysis with a finite $L$: different values of geothermal heat flux and the initial temperature profile of the column are tested (that is precisely the message of Fig. 4).

In the revised manuscript, we have made an extra effort to clarify the new results we present, namely that more realistic treatment of the problem demonstrates a strong relationship of timescales to boundary and initial conditions, and therefore no special timescale of 7ka should be expected to exist. We maintain the section demonstrating how the 7ka timescale can be obtained under the assumptions made by MacAyeal (1993) and emphasise that there is no reason a priori to expect those assumptions to hold universally. We hope that with these changes, it will be clear to the reviewer that the message of our paper is quite consistent with their expectations.

---

## Referee Report (RR1)

I am concerned that the authors have not actually taken on board a number of the referee comments in a significant way. I still think it could be a stronger paper than in current form. Frequently, the authors have only amended the text to acknowledge limitations of their study even if the suggested improvements seemed quite minor. I am fine with the authors explaining that a suggestion was not helpful/beyond the scope/did not reveal anything new but I would like to hear that. Multiple times in the responses to referees, the authors state that a comment 'has been considered' with neither further explanation, nor updated text in the manuscript. For example, I suggested a plot showing the effect of $\theta_L/L$ on the timescale, which seems to have been ignored.

Since all the referees highlighted the strong dependence on initial conditions as an issue, perhaps the authors could show some results with something other than a linear initial profile - e.g. piecewise linear or exponential. This would really help to clarify how the finite depth of the ice column plays a role in the time evolution, compared to by changing the initial temperature profile (which, as another referee highlights, is really the most uncertain part of the proposed application). Also, what is a 'lapse rate' as referred to several times when discussing the temperature profiles?

Both referee 2 and I asked about the 2km transition depth, and I think the authors could do better than their current heuristic explanation. They should ideally predict how the transition depth would vary with the parameters that they say it should depend on - it seems like from their argument the transition to infinite time should only require an analysis of heat fluxes in the steady state? Just because a value depends on parameters doesn't mean it cannot be quantitatively analysed.

Similarly, the depth of 3km to return to the infinite depth solution - is this found by eye? What is the threshold being applied, as it looks more like 2.5km for some surface temperatures? What parameters does this depend on? The suggestion by another referee to non-dimensionalise the problem would clarify these points significantly.

Paragraphs at lines 225 to 237: much of this a description of the non-monotonicity, in quite a verbose way, rather than a physical explanation. This could certainly be condensed, probably into a single paragraph. Consider what the important points are. If it possible to explain why some of the curves are monotonic and others are not, that would strengthen the argument.

Minor comments:

Throughout the text, 'reads' is used instead of 'is'

The term 'periodicity' persists in a few places - double-check and replace.

When discussing the convergence towards the $L \to \infty$ case, variables with tildes are not defined.

The structure of the introduction does not reflect the change in title of the paper. Either find a title that is more of a common ground, or at least pretend the paper is about the more general thermodynamics of ice sheets by adding an introductory paragraph.

---

## Referee Report (RR2)

**Referee Report for TC2022-97 (Thermodynamic evolution of a finite ice column: analytical solution and basal-melting timescales) by Moreno et al.**

This referee report follows a major revision of the paper by the authors. I believe that the changes made to this manuscript have improved it, but their remain serious issues with it and I outline these below.

Please note (both author and editor): should this paper be revised and resubmitted, I would be happy to provide another review but only in the case that the **authors completely remove the 'binge-purge' terminology used in their ms.** I appreciate that this name is not their invention, but such language is outdated, trivialises a serious condition and is potentially triggering to a reader (see e.g. https://www.hopkinsmedicine.org/health/conditions-and-diseases/eating-disorders/bulimia-nervosa for an overview). I also strongly encourage anyone external to this review process who might see this review, to refrain from using such terminology. I do not hold the authors use of this terminology against them in the present review.

As I see it, the authors essentially take the study of MacAyeal and consider how the timescale derived therein would change should I different upper boundary condition be applied. They consider the study of MacAyeal to be infallible, a fact that is evident in both their replies and in their ms (in particular, they dedicate an entire section of the ms to describing how the MacAyeal result can be recovered from their model.) They assume that, because MacAyeal suggested that the timescale of HE events is related to the thawing timescale, then this can safely be assumed. My main issue with the ms (and as raised by reviewer 3 in the previous round) is that the MacAyeal model describes largely the wrong physics for ice temperatures: ice sheet temperatures are set almost entirely by horizontal advection of heat (see e.g. https://doi.org/10.1002/jgrf.20054), rather than diffusion of heat; however, the authors assume that temperatures are *entirely* set by diffusion. To me, this means that no inference can be drawn between the model in the ms and the timescales of HE (even if this comparison was made in previously published work). Since this is the central premise of the paper (there is even a section 'a new period for the binge/purge oscillator') I am not sure if this is surmountable, but at the least I think the paper needs to be entirely reframed with the connection to the HE timescale removed.

I find the authors response to my comment on non-dimensionalization wholly unsatisfying. In their system, there are 7 parameters: kappa, $T_{air}$, L, G/k, beta, theta_L, theta_B, which represents a massive parameter space. Non-dimensionalizing the system (see below for an example on how this would look), reduces the dependence of the 'thawing timescale' to only four dimensionless parameters. This would also allow an 'apples with apples' comparison in many of the figures: in figure 2, for example, configurations with different values of L should be evaluated at different times to properly understand the effect of changing L (in particular, the solution with L= 2.5km should be evaluated $1.5^2 = 2.25$ times later than that with L = 1km because the diffusive timescale is $L^2/kappa$). The authors say that 'the simple enough that keeping dimensionality will help the reader.': I think the opposite is true — it is a complicated system with 7 parameters; reducing the dimensionality will help the reader understand which parameters truly matter.

Regarding the initial condition: the authors responded by saying that 'the linear profile is introduced for simplicity'. That is OK, but, if so, drawing analogy with actual observed timescales is even more tenuous. In addition, they say that it 'allows us to explicitly determine the impact of the initial basal/surface ice temperature independently'. In fact, the authors shouldn't be considering the initial basal temp and surface temp independently; as the non-dimensionalization shows, these quantities are coupled, e..g changing the surface temperature without varying the basal temperature changes the temperature gradient in the column. This is another advantage of using the dimensionless approach.

I would also stress reviewer three's comment that the timescale determined is most sensitive to the basal temperature. I will not repeat this point here, but I do not think this comment has been satisfactorily addressed in the updated ms.

The authors mention that they have placed the code in a repository (by the way, the repository mentioned in the updated ms is different to that mentioned in the referee responses) but this repository is not open.

I do not provide another line by line breakdown of the paper as I feel their are more fundamental problems to be overcome before another such critique is useful.

One dimensional heat diffusion through a column.

$$z = h : \theta = T_{air} + \beta \theta_z$$

$$\theta_t = k \theta_{zz}$$

+ initial condition:

$$\theta(z,0) = \theta_b + \frac{z}{L}(\theta_s - \theta_b)$$

$$z = 0 : \theta_z = G/k$$

introduce scaled variables: $[z] = L$, $[t] = L^2/k$ (diffusive time scale)

(dimensionless variables w/ hats) $[\theta] = T_{air}$

Then: $\theta_t = k\theta_{zz} \longmapsto \hat{\theta}_{\hat{t}} = \hat{\theta}_{\hat{z}\hat{z}}$

bottom bc: $\theta_z = G/k \longmapsto \hat{\theta}_{\hat{z}} = \frac{G}{k} \cdot \frac{[z]}{[\theta]} = \frac{GL}{kT_{air}}$

top bc: $\theta = T_{air} + \beta\theta_z \longmapsto \hat{\theta} = 1 + \frac{\beta}{L}$

IC: $\theta = \theta_b + z(\theta_s - \theta_b) \longmapsto \hat{\theta} = \frac{\theta_b}{T_{air}} + \hat{z}\frac{\theta_s - \theta_b}{T_{air}}$

4 dimensionless parameters:

$\frac{G}{k}\frac{L}{T_{air}}$ : dimensionless geothermal heat flux

$\frac{\beta}{L}$ : dimensionless insulation length (already identified by authors)

$\frac{\theta_b}{T_{air}}$ : dimensionless initial surface temp

There could also be ... term ...

expressed as a temp gradient and temp difference.

$\frac{\theta_s - \theta_b}{T_{air}}$ : dimensionless initial temp diff

---

## Referee Report (RR3)

Analytical solutions for the advective-diffusive ice column in the presence of strain heating, Moreno-Parada et al.

This is a significantly different manuscript to the previous version, so my comments are starting afresh. As a general note, the paper is over-long, and if the editor's decision is another round of revisions, would benefit from a full re-write with a more systematic layout and less repetition in both set-up and description of results.

In this work the authors calculate some steady states and some examples of the time-evolution of a temperature profile in a column of ice, subject to imposed profiles of vertical advection and, at times, internal heat sources and/or surface insulation.

I am not sold that this study is novel or valuable. In part this is because the structure of the paper, in particular the introduction, makes it difficult to understand what precisely is being done in this work, and where the gaps in the literature previously existed. As I understand it, the manuscript claims three points of distinction from previous work: considering surface insulation and internal heating (i.e. different forcing for the advection-diffusion equation, and dropped in section 6 - I don't feel this is enough for a paper unless the output is notable), including time-evolution (unclear from both the introduction and analysis if this would have much impact on realistic ice sheet dynamics, and only included in section 5), and sticking to analytic solutions of a highly simplified model.

By seeking analytical solutions (although the calculation of the eigenvalues, and the evaluation of the constituent functions must be done numerically, so it is more semi-analytic), the authors claim to avoid having to assess the 'accuracy and consistency' of numerical models. However, by limiting to solutions in 1D where analytic results can be obtained, they are also in a regime where numerical solvers for advection-diffusion equation have been validated for decades. I do agree that analytical results have value, e.g. for easily extracting characteristic timescales as functions of parameters, but this has not been done and instead the work presented here, inspecting the solutions by eye, could have been based on purely numerical results with no difference in the discussion. I think this speaks volumes to the lack of real depth in the analysis.

I won't make too many specific points about the analysis as I feel my concerns are broader, but some illustrative examples:

- Around l.255 it is stated that there are two different timescales visible in the results. One way to quantify this would be to relate these timescales to the different eigenvalues (which are the exponential decay rates). This is not done.

- Figure 5 is just thrown in to the mix with no attempt to explain the trends. In general there is no quantification of results based on the dimensionless parameters, only demonstration of output.

- Figure 6 claims to show a favourable comparison to the EISMINT results for $m = 1.5$. I would expect to see the EISMINT results actually plotted on the same graph.

I also have major concerns about the set-up of the equations:

- The idea of upwards advection (from ice being created at the base?) is unreasonable, yet presented throughout as though it is one of the two equally plausible regimes.

- The strain-heating term dependent on $u_x$ should, by the incompressibility of Stokes flow, also be expressible in terms of $w_z$ - a quantity which is explicitly calculable from the vertical advection profiles $w(z)$ - yet is instead taken as constant throughout the paper and then is neglected completely in section 7, when $w_z$ varies in depth.

- There is no reason why advection of temperature should appear as a uniform source term in this model (the paper cited depth-integrates variables, but the present manuscript only considers the vertical structure, which depth-integration wipes out). Gratingly, in the discussion section the authors extol their inclusion of horizontal advection.

- Please stop calling the surface boundary condition 'sophisticated' - it's not that exciting.

Finally, there are consistent spelling and grammatical errors throughout the manuscript, and the discussion section needs editing for clarity.

---

## Referee Report (RR4)

**Review: "Analytical solutions for the advective-diffusive ice column in the presence of strain heating"**

by Daniel Moreno-Parada et al.

Submitted to *The Cryosphere*

**1 General**

In this paper, the authors analyze the temperature structure of glaciers as a function of advection, shear heating, and surface forcing. They develop analytical solutions using classical applied mathematics yielding interesting, unwieldy, and dumbfounding expressions. It is not clear that their analysis presents new applied mathematics, except for a new solution to an ODE, and does not clearly answer a glaciological problem. The work is of publishable quality and I won't stand in the way, but I am not sure what goal this paper is achieving.

**2 Specific comments**

1. line 1 in the abstract: 'of paramount importance' is 'paramount'

2. line 5 in the abstract: 'sophisticated' can be replaced by 'Robin-type'

3. line 6 in the abstract: non-equilibrium temperature and non-equilibrium thermodynamics are very different, but I think the authors are referring to the former, here and throughout. I suggest clarifying the language.

4. line 8 in the abstract: the surface insolation number? As I note later, it is typically called the Biot number.

5. line 9 in the abstract: there is a typo in 'Brinkman' in many places throughout the text as 'Brikman'.

6. equation (7): I would put $\mathrm{Pe}\,\xi$ into the equation for $w$ so that the system is closed and all of the parameters are clear.

7. seemingly much of this analysis is presented in Hills et al. (2023).

8. in figure 2, why does changing $\beta$ have little effect?

9. There are positive and negative vertical velocities presented, what is the physical mechanism of positive vertical velocity?

10. Brainstorming a few ideas: this analytical solution is likely relatively fast to compute, could this be a good initialization for an ice sheet model? I know there is an interest in Heinrich events from the coauthors, could it be a better analytical model to use in a thermodynamically coupled ice sheet model?

**References**

B. H. Hills, K. Christianson, R. W. Jacobel, H. Conway, and R. Pettersson. Radar attenuation demonstrates advective cooling in the Siple Coast ice streams. *J. Glaciol.*, 69(275):566–576, 2023. doi: 10.1017/jog.2022.86.

---

## Referee Report (RR5)

**Referee report for "Analytic solutions for the advective-diffusive ice column in the presence of strain heating"**

This paper has changed significantly since the last revision, which I welcome. After consulting with the editor, I am writing this review as if the paper is a major revision, rather than a new submission.

In this updated paper, the authors present a modified model, in which vertical advection, strain heating and horizontal advection are included. They non-dimensionalize the problem as suggested, identifying four dimensionless parameters which describe the behaviour: a dimensionless geomthermal heat flux, a dimensionless 'insulation coefficient', a vertical Peclet number, and a dimensionless source term. They solve this system analytically, and, firstly, present steady state configurations for a range of parameter combinations and, then, present some transient solutions from an arbitrary initial condition. They then present another analytic solution which corresponds to a vertical velocity profile with a more general power law dependence on depth; comparing this vertical velocity profile to that seen in numerical experiments gives a temperature profile which they claim can be used as an analytic control on thermodynamic ice sheet model behaviour.

However, despite the changes, I find that the current manuscript still has major flaws. These flaws are different to those in the previous manuscript, but they are major flaws nonetheless. It feels a bit unfair to be pointing out 'new' major flaws upon third revision, but I think that is perhaps inevitable given how much the paper has changed. The motivation (although different from before) is very tenuous, the analysis is very limited and includes mistakes (which affect the rest of the paper), their new section (the EISMINT experiments) doesn't, to my mind, offer what they claim, and they make many broad, sweeping statements based on very limited number of results in a small region of parameter. Below I expand on these in more detail.

I find the motivation to be lacking. The authors claim that the paper builds upon the works of Robin and Lliboutry by including a time-dependent component, but I don't see what the benefit of the time dependent component is (more on this below). They say that this "allow[s] for a more accurate representation of the ice behaviour in response to changing external conditions", but then impose boundary conditions which are constant in time. They also claim that "transient solutions offer the potential to refine the interpretation of ice core data", but this seems to be a stretch to me (and is not elaborated on): the author's timescale on which solutions approach the steady state is kappa/L^2, which is on the order of minutes, suggesting that the ice column is always in quasi-equilibrium with the top boundary condition and the time-dependent state is not important.

While the authors have included horizontal advection in their model in a way, they do so via an awkward source term which is then completely ignored in their analysis (they focus only on a strain heating source term). The authors claim that the dimensionless horizontal advection term is in the range of 0-0.01, which I disagree with. To demonstrate this, I have quickly plotted Lambda = L^2 / (kappa*|T_air|) * V over the Antarctic ice sheet, where kappa = 36m^2/year is the thermal diffusivity, T_air = -20C is the air temperature, L is the ice thickness and V is the horizontal ice velocity, a proxy for the integral they consider. This plot shows that there are very few regions where the quantity Lambda is < 1; in fact, over most of the ice sheet it is very large, suggesting that horizontal advection is dominant. I had mentioned horizontal

advection in my previous review and, while the authors need credit for trying to include it, they have not done so satisfactorily. The authors also mention the role of horizontal advection in their discussion, but it is so central to the problem that it cannot be ignored in a model that attempts to say anything useful about ice sheet temperatures.

[Figure]

Their equation (2), now updated to include a vertical advection term, is still missing a term from flow divergence. The first of equation (2) should read $\theta_t = \kappa\theta_{zz} - w\theta_z - w_z\theta + \Omega$. I am not sure how this would change the rest of the analysis but, given that they claim vertical advection is very important, this term could potentially be very important.

The solutions shown in figure 2 are not actually solutions to the problem (9): the solutions shown have the wrong boundary condition at the upper surface. You can see this from panel (f), for example: for beta = 0, the temperature should be 1 at the upper surface (just from reading off the boundary condition), whereas these plots show 0 temperature there. I suspect the authors have solved with a boundary condition $\beta v_\xi + v = 0$ at the upper surface. Whilst this doesn't seem to change the qualitative behaviour of the solutions (I coded this problem up myself and solved it numerically, see figure below.), it does call into question their analytic solution: is this only valid for the boundary condition $\beta v_\xi + v = 0$? In addition, the rest of their discussion in this section is based on these solutions, which are wrong. There are also sign errors in equation (9), Omega has the wrong sign (based on its definition in equation (2)) and so does gamma. The Brinkmann number is also referred to frequently as the Pe (see e.g. the caption of figure 2). It's also very confusing to introduce *parameters* and then not change their names when they are non-dimensionalized.

[Figure]

The time-dependent aspect of the model is not, too my mind, physically relevant: there is no situation where a 1km thick block of ice simply appears with a uniform temperature and then relaxes to an equilibrium. The authors also claim that "the time required to reach the stationary state is considerably shorter for w0 < 0", which is based on a single solution. Clearly, if you started at the w0 > 0 steady state, the opposite would be true, and so this statement cannot hold in practice. There may well be some physical reason why solutions with w0 < 0 converge faster (e.g. the interaction between the sign of w0 and the particular asymmetric boundary conditions), but this is not probed at all. The authors also present the time evolution of the energy in the system and say that "we can study how the total energy balance of the ice column depends on the four dimensionless numbers that determine the stationary solutions", but then don't study it at all…

In addition to this, there are more general statements, including most of those presented in the abstract, which are based on a small subset of simulations in a small region of parameter space. Numerical solutions, such as those presented in figure 2 are useful to understand how varying parameters affects the behaviour, but general statements about the behaviour over the whole of parameter space cannot be made.
      - They say that "The Peclét number produces the largest changes in the equilibrium solutions", which doesn't even appear to be the case in figure 2 (for example, the variations in gamma are equally large).
      - They say that "..under downwards advective conditions, the thermal basal equilibrium is found irrespective of the specific top boundary condition" (i.e. the solution is independent of beta): this is only true in figure 2f because the range of beta is so small. For larger beta, the solution does vary with beta (see e.g. figure below, which is as in figure 2f but for a larger range of beta). The authors claim that beta is in the range 0-0.125, which no justification, but, even if that were true, for other values of the parameters (e.g. weaker geothermal heat flux, weaker downwards advection), dependence on beta would be seen I suspect.

[Figure]

Finally, I find the EISMINT section very confusing. As far as I can tell, the authors fit a the velocity profile from the output of a numerical ice sheet model to a power law profile (this is not shown anywhere). They then use this exponent to determine the temperature profile, using dimensionless parameters which are also computed from the ice sheet model. They claim (but don't ever show) that this profile matches the output from the numerical ice sheet model. They also claim that this result gives an independent control against which ice sheet models can be verified. I disagree with this, since both the velocity profile and model parameters have to be determined from the ice sheet model itself. Even if so, I'm not really sure what benefit the analytic solution gives (by the way, the incomplete gamma function has to be evaluated numerically, so this isn't truly an analytic solution, and I think I commented on similar in previous revisions) over just solving the equation numerically, which authors have don previously (e.g. https://tc.copernicus.org/articles/16/1221/2022/).

---

## Author Response (AR2)

**Author's response**
**TC-2022-97**

Daniel Moreno-Parada, Alexander Robinson, Marisa Montoya, and Jorge Alvarez-Solas.

July 12, 2023

**Contents**

Note: Reviewers comments are given in blue font whereas the author response reads in black.

**1 Relevant changes made in the manuscript**

Here is a summary of the main changes of the current manuscript version.

- Removed all connection of our study with Heinrich Events timescales (MacAyeal, 1993a, b).
- Reframing of the manuscript.
    - New title.
    - Re-written introduction for a more general thermodynamics approach.
- A more sohpisticated physical description of the system:
    - Vertical advection. Including $z$-dependency $w(z)$.
    - Strain heating term.
    - Hoizontal advection (vertically-averaged).
    - Generalised basal inflow of heat:
        * Geothermal heat flux.
        * Potential basal frictional heat term.
- Structure of the solution: stationary/transient components. Independent study.
- Compairson with the EISMINT benchmarks to provide realistic values to our dimensionless study.
    - Additional Table 2 with dimensionless parameters, ranges and particular EISMINT values for a realistic ice sheet..
    - General power-law formulation of the $z$-dependency of the vertical advective term $w(z)$.
- New detailed appendices with solution derivations.
- GitHub repository (scripts to plot figures and necessary calculations).

**2 Reviewer 1**

I am concerned that the authors have not actually taken on board a number of the referee comments in a significant way. I still think it could be a stronger paper than in current form. Frequently, the authors have only amended the text to acknowledge limitations of their study even if the suggested improvements seemed quite minor. I am fine with the authors explaining that a suggestion was not helpful/beyond the scope/did not reveal anything new but I would like to hear that. Multiple times in the responses to referees, the authors state that a comment 'has been considered' with neither further explanation, nor updated text in the manuscript. For example, I suggested a plot showing the effect of L/L on the timescale, which seems to have been ignored.

Since all the referees highlighted the strong dependence on initial conditions as an issue, perhaps the authors could show some results with something other than a linear initial profile - e.g. piecewise linear or exponential. This would really help to clarify how the finite depth of the ice column plays a role in the time evolution, compared to by changing the initial temperature profile (which, as another referee highlights, is really the most uncertain part of the proposed application). Also, what is a 'lapse rate' as referred to several times when discussing the temperature profiles?

Both referee 2 and I asked about the 2km transition depth, and I think the authors could do better than their current heuristic explanation. They should ideally predict how the transition depth would vary with the parameters that they say it should depend on - it seems like from their argument the transition to infinite time should only require an analysis of heat fluxes in the steady state? Just because a value depends on parameters doesn't mean it cannot be quantitatively analysed.

Similarly, the depth of 3km to return to the infinite depth solution - is this found by eye? What is the threshold being applied, as it looks more like 2.5km for some surface temperatures? What parameters does this depend on? The suggestion by another referee to non-dimensionalise the problem would clarify these points significantly.

Paragraphs at lines 225 to 237: much of this a description of the non-monotonicity, in quite a verbose way, rather than a physical explanation. This could certainly be condensed, probably into a single paragraph. Consider what the important points are. If it possible to explain why some of the curves are monotonic and others are not, that would strengthen the argument.

The authors are deeply grateful to the reviewer for their constructive comments. The current work has strongly benefit from them.

With this new revised manuscript version, we have taken on board all comments in a significant way, as it can be seen in the relevante changes section. Particularly, following the suggested non-dimensionalization of the problem, we have addressed the effect of $\theta_L/L$. Furthermore, we have defined four dimensionless numbers (see Table 2 in the revised manuscript): the Peclét number Pe, the effective geothermal heat flux $\gamma$, Brikman number Br and normalised surface insulation parameter $\beta$.

Moreover, all references to MacAyeal (1993) and periodicity have been removed, so that the

current paper is completely general and does not rely on any of MacAyeal's assumptions. This further solves the issue of the dependency to the initial conditions, given that the solutions are expressed as the sum of a transient and a stationary component.

All discussion on non-monotonicy is thus avoided as we no longer focus on the time required to melt the column base from a particular initial state. Rather, we now describe a much more sophisticated physical system where vertical advection, strain heating and vertically-averaged horizontal advection are also considered.

Minor comments: Throughout the text, 'reads' is used instead of 'is' The term 'periodicity' persists in a few places - double-check and replace. When discussing the convergence towards the $L \xrightarrow{\infty}$ case, variables with tildes are not defined. The structure of the introduction does not reflect the change in title of the paper. Either find a title that is more of a common ground, or at least pretend the paper is about the more general thermodynamics of ice sheets by adding an introductory paragraph.

- The used of 'read' has been reconsidered when necessary.
- The term 'periodicity' has completely disappear as we do not link our analytical solution with MacAyeal (1993) or Heinrich Events.
- Likewise, the convergence study towards MacAyeal (1993) has been omitted.
- Both the title and the introduction have changed to reflect the more general thermodynamic approach.

**3   Reviewer 2**

This referee report follows a major revision of the paper by the authors. I believe that the changes made to this manuscript have improved it, but their remain serious issues with it and I outline these below.

Please note (both author and editor): should this paper be revised and resubmitted, I would be happy to provide another review but only in the case that the **authors completely remove the 'binge-purge' terminology used in their ms**. I appreciate that this name is not their invention, but such language is outdated, trivialises a serious condition and is potentially triggering to a reader (see e.g. https:// www.hopkinsmedicine.org/health/conditions-and-diseases/eating-disorders/bulimia-nervosa for an overview). I also strongly encourage anyone external to this review process who might see this review, to refrain from using such terminology. I do not hold the authors use of this terminology against them in the present review.

As I see it, the authors essentially take the study of MacAyeal and consider how the timescale derived therein would change should I different upper boundary condition be applied. They consider the study of MacAyeal to be infallible, a fact that is evident in both their replies and in their ms (in particular, they dedicate an entire section of the ms to describing how the MacAyeal result can be recovered from their model.) They assume that, because MacAyeal suggested that the timescale of HE events is related to the thawing timescale, then this can safely be assumed. My main issue with the ms (and as raised by reviewer 3 in the previous round) is that the MacAyeal model describes largely the wrong physics for ice temperatures: ice sheet temperatures are set almost entirely by horizontal advection of heat (see e.g. https://doi.org/10.1002/jgrf.20054), rather than diffusion of heat; however, the authors assume that temperatures are entirely set by diffusion. To me, this means that no inference can be drawn between the model in the ms and the timescales of HE (even if this comparison was made in previously published work). Since this is the central premise of the paper (there is even a section 'a new period for the binge/purge oscillator') I am not sure if this is surmountable, but at the least I think the paper needs to be entirely reframed with the connection to the HE timescale removed.

I find the authors response to my comment on non-dimensionalization wholly unsatisfying. In their system, there are 7 parameters: $\kappa$, $T_{\mathrm{air}}$, $L$, $G/k$, $\beta$, $\theta_L$, $\theta_b$, which represents a massive parameter space. Non- dimensionalizing the system (see below for an example on how this would look), reduces the dependence of the 'thawing timescale' to only four dimensionless parameters. This would also allow an 'apples with apples' comparison in many of the figures: in figure 2, for example, configurations with different values of $L$ should be evaluated at different times to properly understand the effect of changing $L$ (in particular, the solution with $L = 2.5$ km should be evaluated $1.5^2 = 2.25$ times later than that with $L = 1$ km because the diffusive timescale is $L^2/\kappa$). The authors say that 'the simple enough that keeping dimensionality will help the reader.': I think the opposite is true — it is a complicated system with 7 parameters; reducing the dimensionality will help the reader understand which parameters truly matter.

Regarding the initial condition: the authors responded by saying that 'the linear profile is introduced for simplicity'. That is OK, but, if so, drawing analogy with actual observed timescales is even more tenuous. In addition, they say that it 'allows us to explicitly determine the impact of the initial basal/surface ice temperature independently'. In fact, the authors shouldn't be considering the initial basal temp and surface temp independently; as the non-dimensionalization shows, these quantities are coupled, e..g changing the surface temperature without varying the basal temperature changes the temperature gradient in the column. This is another advantage of using the dimensionless approach.

I would also stress reviewer three's comment that the timescale determined is most sensitive to the basal temperature. I will not repeat this point here, but I do not think this comment has been satisfactorily addressed in the updated ms.

The authors mention that they have placed the code in a repository (by the way, the repository mentioned in the updated ms is different to that mentioned in the referee responses) but this repository is not open.

I do not provide another line by line breakdown of the paper as I feel their are more fundamental problems to be overcome before another such critique is useful.

We are deeply grateful to the reviewer for their constructive comments and the provided example as part of the report. The current manuscript version has strongly benefit from them.

As the reviewer requested to further proceed with another review, **we have completely removed the 'binge-purge' terminology used in our ms**.

We have entirely reframed the paper and omitted the previous central premise of the manuscript, thus removing any connection with Heinrich Event timescales. As a result, the paper has taken a much more general approach and does not rely on the any of the assumptions found in MacAyeal (1993). We now describe a more realistic physical system with several processes that were previously missing: vertical advection, strain heating and vertically-averaged horizontal advection among others (see Section 1 of this document for a summary of the relevant changes made in the manuscript).

Regarding non-dimensionalisation, we have rigorously followed Reviewer's 2 suggestion. We are deeply grateful for the example provided in their report. As shown in Eq. 6 and Table 2 of the revised manuscript, we only have now 4 dimensionless parameters that determine the analytical solutions: the Peclét number Pe, the effective geothermal heat flux $\gamma$, Brikman number Br and normalised surface insulation parameter $\beta$.

Concerning the initial condition, we have eluded previously raised issues by reframing the current manuscript in two main points. First, we no longer focus on determining the thawing timescale as any connection with MacAyeal (1993) has been removed. Moreover, as noted by Reviewer 2, our dimensionless approach further surmounts this problem as changes in the basal/surface temperatures perturb the temperature gradient in the ice column.

---

## Author Response (AR3)

**Author's response**
**TC-2022-97**

Daniel Moreno-Parada, Alexander Robinson, Marisa Montoya, and Jorge Alvarez-Solas.

January 16, 2024

**Contents**

Note: Reviewers' comments are given in italic font whereas the authors' responses read in regular font.

**1 Relevant changes made in the manuscript**

Following the editor's guidelines, we have included several changes to make the paper of deeper practical use for the cryospheric community.

In particular, from the potential paths for publication suggested by the editor, we have applied our analytical results to test numerical solvers (Table 3) as shown in the new Section 6 (see also Fig. 6 and 7). A suite of benchmark experiments is summarised in Table 2. The current manuscript version exhibits a strong balance between theoretical knowledge and a clear application for glaciology.

Regarding our choice of time-scaling, there are other equally valid scaling options, such as the advection time scale $L/w_0$ noted by the editor. However, this scaling depends on the particular advective regime $w_0$, thus implying different time scales for each $w_0$ value employed. We prefer to non-dimensionalise our model with the same time scaling for all scenarios so that we obtain a unique dimensionless time scale. The particular choice will have no influence on the results so long as the conversion among variables is consistent.

Here is a summary of the main changes of the latest manuscript version:

- Reframed motivation.
  - Clear identification of the current theoretical gap in available analytical solutions.
  - Test for numerical solvers: benchmark experiments.
- Re-written conclusion for a clear message of the work.
- Updates on the physical description of the system:
  - Focus on the downwards advective scenario $w_0 < 0$ (all figures have been accordingly updated).
  - Time scale analysis from eigenvalues as suggested by Reviewer 2 (Fig. 5 and Discussion section).
  - Rewriting of vague statements that were previously based on a small subset of the solutions.
  - EISMINT section removed as requested by the editor.
- Test for numerical solvers: whole new Section 6.
  - Additional suite of benchmark experiments to test numerical solvers (Table 2, Figures 6 and 7). This entails two main results:
    * Best discretisation choices.
    * Minimum resolution for reliable temperature profiles.
  - Discretisation over nonuniform grids as state-of-the-art ice sheet models employ (Table 3).
  - A number of different numerical schemes to test accuracy and reliability as a function of the convergence order (Table 3).

- Resolution study (Fig. 7).

- New detailed appendices with disretisation schemes for numerical solvers (Appendix D).

- Updated GitHub repository (scripts to plot figures and necessary calculations).

**2  Reviewer 1**

The authors are deeply grateful to the reviewer for their constructive comments. The current work has strongly benefit from them. We now provide our answers (regular font) to the main concerns risen by the reviewer (italic).

- *I do agree that analytical results have value, e.g. for easily extracting characteristic timescales as functions of parameters, but this has not been done and instead the work presented here, inspecting the solutions by eye, could have been based on purely numerical results with no difference in the discussion. I think this speaks volumes to the lack of real depth in the analysis.*

  [...]

  *Around l.255 it is stated that there are two different timescales visible in the results. One way to quantify this would be to relate these timescales to the different eigenvalues (which are the exponential decay rates). This is not done.*

We have included an entirely new figure (Fig. 5) to extract the characteristic timescales as a function of relevant parameters. Figure 5a shows the decay time dependency to the particular eigenvalues and Fig. 5b further maps the decay times as a function of the two controlling parameters: $\beta$ and Pe. Results and discussion sections have also been updated accordingly and we have determined what parameters affect the timescales of the system. Moreover, we have completely removed any inspection by eye.

- *The idea of upwards advection (from ice being created at the base?) is unreasonable, yet presented throughout as though it is one of the two equally plausible regimes.*

We have entirely removed the upwards advection regime from the manuscript. Even so, solutions are still applicable to both positive and negative values of $w_0$.

- *The strain-heating term dependent on $u_x$ should, by the incompressibility of Stokes flow, also be expressible in terms of $w_z$ − a quantity which is explicitly calculable from the vertical advection profiles $w(z)$ - yet is instead taken as constant throughout the paperand then is neglected completely in section 7, when $w_z$ varies in depth.*

The strain-heating term is introduced as a vertically-integrated magnitude (following Meyer and Minchew, 2018) to keep the analytical tractability of the solution. An additional vertical dependency of the inhomogeneous term falls beyond the analytical tractability of the equation, since it cannot be expressed as a confluent hypergeometric equation.

   The physical implications of such approximation can be divided into stationary and transitory. In the former, the eigenvalues of the problem (and consequently the relaxation times of the system) are independent of the lateral advection of heat. This would not be true for a numerical solver if an explicit advective horizontal term was to be introduced. Regarding the stationary solution, one could expect small variations in the temperature profile if the lateral advective term was

$z$-dependent, even though the surface and the basal temperatures would be still determined by the boundary conditions.

- *Finally, there are consistent spelling and grammatical errors throughout the manuscript, and the discussion section needs editing for clarity.*

We have thoroughly revised the manuscript to correct such errors and provide greater clarity in the discussion section.

**3  Reviewer 2**

The authors are deeply grateful to the reviewer for their constructive comments. The current work has strongly benefit from them. We now provide our answers (regular font) to the main concerns risen by the reviewer (italic).

- *Line 1 in the abstract: 'of paramount importance' is 'paramount'.*

This typo has been fixed.

- *Line 5 in the abstract: 'sophisticated' can be replaced by 'Robin-type'.*

We have included the suggestion.

- *Line 6 in the abstract: non-equilibrium temperature and non-equilibrium thermodynamics are very different, but I think the authors are referring to the former, here and throughout. I suggest clarifying the language.*

Indeed, we refer to non-equilibrium temperature. We have changed the manuscript accordingly.

- *Line 8 in the abstract: the surface insolation number? As I note later, it is typically called the Biot number.*

The Biot number (Bi) characterizes the relative importance of conduction within a solid compared to convection at its surface. Therefore, it requires additional information of the convective heat transfer coefficient $h$ of the air. The Biot number provides information of whether the solid has a uniform temperature (Bi $\ll$ 1) if convection is dominant. Empirical measurements show the strong vertical dependency of ice temperature profiles, thus implying that glacial ice is mostly in the conductive Biot regime (Bi $\gg$ 1).

We rather aim at capturing the fact that the thermal conductivity of glacial ice $k(\rho)$ is reduced towards the surface since density drops as we approach the ice-air interface (knowing that ice forms through snow densification). Following Carlsaw and Jaeger (1989), we simply apply a general "Newton's Law", where the heat flux across a surface is proportional to the temperature difference between the surface and the surrounding medium. The traditional approach, i.e. imposing a particular ice surface temperature given by the air temperature, is then a limit case of "Newton's law" (where $\beta = 0$).

- *Line 9 in the abstract: there is a typo in 'Brinkman' in many places throughout the text as 'Brikman'.*

We have changed the manuscript accordingly.

- *equation (7): I would put Pe ξ into the equation for w so that the system is closed and all of the parameters are clear.*

We agree. The manuscript has been updated to reflect this.

- *In figure 2, why does changing β have little effect?*

Since we focus on the downwards advective regime, the temperature profile is strongly determined by the downwards transport of colder ice from the surface. Since the upwards transport of heat due to difussion is significantly weaker than the advective counterpart, the upper half temperature of the ice column is mostly dictated by $w_0$. The particular insulating conditions are critical only if the heat inflow from the base (geothermal) can reach the uppermost layers of the ice column, thus yielding a temperature difference between the ice surface and the adjacent atmosphere. We have clarified the manuscript to reflect this mechanism.

- *There are positive and negative vertical velocities presented, what is the physical mechanism of positive vertical velocity?*

Upwards advective conditions are indeed quite rare in real ice sheets and glaciers. As suggested by the editor and other reviewers, we have focused our present work on the downward advective solution $w_0 < 0$, though the solution are still applicable to both scenarios.

- *Brainstorming a few ideas: this analytical solution is likely relatively fast to compute, could this be a good initialization for an ice sheet model? I know there is an interest in Heinrich events from the coauthors, could it be a better analytical model to use in a thermodynamically coupled ice sheet model?*

We really appreciate the brainstorming of new ideas. Indeed, the analytical solution is quite fast to compute. It would be definitely a good approximation to initialize ice sheet models as it captures the transitory component of the solution during the relaxation time. This will surely add value to the motivation of the present work.

Regarding Heinrich Events (HE), this model provides a more comprehensive description of the system. Yet the required analytical tractability of the solution implies simplified physics that reviewers pointed out in prior iterations. As a results, we decided to drop the application to HE and leave the present study to its broader theoretical implication and convenience to test numerical solvers.

**4 Reviewer 3**

The authors are deeply grateful to the reviewer for their constructive comments. The current work has strongly benefit from them. We now provide our answers (regular font) to the main concerns risen by the reviewer (italic).

- *The motivation (although different from before) is very tenuous, the analysis is very limited and includes mistakes (which affect the rest of the paper), their new section (the EISMINT experiments) doesn't, to my mind, offer what they claim, and they make many broad, sweeping statements based on very limited number of results in a small region of parameter*

Regarding the motivation, we have reframed its structure to reflect the two main points that the work provides:

1. Clear identification of the current theoretical gap in available analytical solutions:
   - Absence of a strain heating term due to internal ice deformation.
   - Analytical description of a vertically-integrated exchange of heat due to lateral advection (term previously employed for stationary solutions in Meyer and Minchew, 2018).
   - Lack of time dependency in previous studies. Particularly convenient for initialization of ice sheet models.

2. Test for numerical solvers: benchmark experiments.
   - Additional suite of benchmark experiments to test numerical solvers (Table 2, Figures 6 and 7).
     - Unevenly-spaced grids (both polynomial and exponential) to obtain higher resolutions near the base whilst minimising the total number of grid points, thus reducing computational costs (Table 3).
     - A number of different numerical schemes to test accuracy and reliability as a function of the convergence order (Fig. 6).
   - Resolution study (Fig. 7): minimum resolution for reliable temperature profiles.

3. The EISMINT section has been completely removed.

- *[...] the author's timescale on which solutions approach the steady state is $\kappa/L^2$, which is on the order of minutes, suggesting that the ice column is always in quasi-equilibrium.*

To give an estimation, for a typical vertical ice extent $L \sim 10^3$ m and $\kappa \sim 10^{-6}$ m$^2$/s, the timescale of the transitory solution is of the order $t = L^2/\kappa \sim 10^4$ years (note that the thermal difussivity has been expressed in m$^2$/yr), and not minutes (see also Fig. 5a and 5b). We have included an entirely new figure (Fig. 5) to extract the characteristic timescales as a function of relevant parameters. Figure 5a shows the decay time dependency to the particular eigenvalues and Fig. 5b further maps the decay times as a function of the two controlling parameter: $\beta$ and Pe. Results and discussion

sections have also been updated accordingly and we have determined what parameters affect the timescales of the system.

There are other equally valid scaling options, as the advection time scale $L/w_0$. However, this scaling depends on the particular advective regime $w_0$, thus implying different time scales for each $w_0$ value. We prefer to non-dimensionalise our model with the same time scaling for all scenarios.

- *While the authors have included horizontal advection in their model in a way, they do so via an awkward source term which is then completely ignored in their analysis (they focus only on a strain heating source term). The authors claim that the dimensionless horizontal advection term is in the range of 0-0.01, which I disagree with. To demonstrate this, I have quickly plotted $\Lambda = L^2/(\kappa|T_{\mathrm{air}}|)V$ over the Antarctic ice sheet, where $\kappa = 36m^2/year$ is the thermal diffusivity, $T_{\mathrm{air}} = -20^{o}C$ is the air temperature, $L$ is the ice thickness and $V$ is the horizontal ice velocity, a proxy for the integral they consider.*

Horizontal advection is introduced following Meyer and Minchew (2018) as a vertically-integrated contribution $\Lambda$. This term is now analysed in Fig. 2e, Fig. 3b and Fig. 4c, as well as the Discussion section. We agree that our previous estimation of the nondimensionless range of values was mistaken since the spanned range was too narrow. We thank the reviewer for including an additional figure and we have accordingly updated the manuscript throughout and further updated Table 1.

The physical interpretation of $\Lambda$ is a source/sink of heat due to horizontal advection and its value provides information about the relative strength compared to difussion. The sign of the horizontal temperature gradients along the ice flow will determine whether the contribution is a source or a sink. These gradients are bounded and often supplied by an additional heat source as the boundary conditions or the basal frictional heat. As noted by Robel et al. (2013), for rapidly–moving ice streams, strong advection can preserve the shape of the vertical temperature distribution, with the same basal and surface temperatures as before. Therefore, in strong horizontal advective regimes, temperature gradients along the flow become small and thus the overall heat exchange is bounded (e.g., Dahl-Jensen, 1989). To keep the analytical tractability, it is not possible to explicitly solve for the horizontal dimensions. Hence, the heat exchange in such dimensions must be parametrised by a vertically-integrated quantity that implicitly considers realistic horizontal temperature gradients (as Meyer and Minchew, 2018). This definition does not explicitly capture those gradients, so $\Lambda$ values cannot be solely estimated from velocities, but also limited by a reduced horizontal gradient along the flow. For this reason, the non-dimensional values of $\Lambda$ cannot span 9 orders of magnitude, but rather a more conservative range of $\sim 1$ order of magnitude. Such values yield temperature profiles that already capture the shape of a rapidly-streaming ice sheet flow line (e.g., Chapters § 9.6 and 9.8, Cuffey and Paterson, 2010).

- *Their equation (2), now updated to include a vertical advection term, is still missing a term from flow divergence. The first of equation (2) should read $\theta_t = \kappa\theta_{zz} - w\theta_z - w_z\theta + \Omega$. I am not sure how this would change the rest of the analysis but, given that they claim vertical advection is very important, this term could potentially be very important.*

We disagree that Eq. (2) has a missing divergence term (e.g., Cuffey and Paterson 9.7.1.). As the editor said: "The reason is that in the areas where horizontal advection is not important, thermal

transport is usually dominated by vertical advection and the Peclet number is larger than one (See for example Table 3 in Nereson and Waddington 2017)".

Moreover, as noted by Reviewer 1, we can apply the "incompressibility of Stokes flow" to our problem, thus implying a divergence-free velocity field.

- *The solutions shown in figure 2 are not actually solutions to the problem (9): the solutions shown have the wrong boundary condition at the upper surface.*

This is indeed our mistake. As a preliminary test of our analytical solutions, we set $\beta v_\xi + v = 0$ to retrieve Clarke et al. (1989) stationary solutions. Unfortunately, we included those figures rather than our actual solution for the $\beta v_\xi + v = 1$ surface boundary condition. We have now updated our figures to account for the latter top boundary condition.

- *There are also sign errors in equation (9), Omega has the wrong sign (based on its definition in equation (2)) and so does gamma. The Brinkmann number is also referred to frequently as the Pe (see e.g. the caption of figure 2).*

We agree. The manuscript has been updated accordingly.

- *In addition to this, there are more general statements, including most of those presented in the abstract, which are based on a small subset of simulations in a small region of parameter space. Numerical solutions, such as those presented in figure 2 are useful to understand how varying parameters affects the behaviour, but general statements about the behaviour over the whole of parameter space cannot be made.*

We have completely rewritten the Discussion section to avoid vague statements on small subsets of the solutions. Moreover, the abstract is now updated to reflect those changes.

- *Finally, I find the EISMINT section very confusing. As far as I can tell, the authors fit a the velocity profile from the output of a numerical ice sheet model to a power law profile (this is not shown anywhere). They then use this exponent to determine the temperature profile, using dimensionless parameters which are also computed from the ice sheet model.*

The EISMINT section has been completely removed for clarity. Instead, there is a new section where benchmark experiments for numerical solvers are described in detail (Section 6). Additionally, we have included a number of different numerical schemes (Table 3) to test accuracy and reliability as a function of the convergence order (Figures 6 and 7).

---

## Author Response (AR4)

**Author's response**
**TC-2022-97**

Daniel Moreno-Parada, Alexander Robinson, Marisa Montoya, and Jorge Alvarez-Solas.

May 20, 2024

**Contents**

Note: Reviewers' comments are given in italic font whereas the authors' responses read in regular font.

**1    Relevant changes made in the manuscript**

Following the editor's guidelines, we have included the neccesary changes to make the paper of deeper practical use for the cryospheric community. The current work makes now a stronger case in the paper about what is new and why the method is a useful tool, i.e. the significance of the paper.

Here is a summary of the main changes of the latest manuscript version:

- Reframed introduction.
  - Clear identification of the current theoretical gap in available analytical solutions.
  - Statement on how the current work aims at filling this gap.
  - Suite of benchmark experiments.
- Re-written discussion and conclusion to avoid repetition.
- Brief description on Newton's law of cooling as requested by the reviewer.
- New additional Table 2 with a summary of physical parameters to justify the dimensionless explored range in the temperature solutions.
- Relaxed assumption on the strain rate regimes. Solutions are now applicable to any regime:
  - Shear margins.
  - Centerline of fast-flowing ice streams.
- Figures are now expressed in dimensional terms to ease interpretation.
- Table 1 has been rearranged in a different location to define variables before appearance.
- Expanded discussion on potential depth-averaged lateral advection and depth-independent strain-rates assumptions.
- Clarification on the importance of $\beta$ parameter (see Discussion section).
- Updated GitHub repository (scripts to plot figures and necessary calculations).

**2    Reviewer #5**

The authors are deeply grateful to the reviewer for their constructive comments. The current work has strongly benefitted from them. We now provide our answers (regular font) to the main concerns risen by the reviewer (italic).

**Larger comments**

- *The introduction has a lot of good insight but is a bit disorganized – it begins with a list of the previous temperature modeling studies, goes into the need for validation of numerical models and model initialization, and then starts talking about optimizing for temperature. I think there needs to be more organization and clear takeaways for (1) where the current knowledge gap is, and (2) how this study fills that knowledge gap.*

We have improved the organization of the introduction by emphasizing the current knowledge gap in the literature and how the present work provides new insight. Additionally, the manuscript contains a clear statement on the limitations that the current study presents.

- *Why make the assumption that the lateral strain rate is a dominant component of the strain rate tensor – this seems to be an assumption that is really only valid in shear margins, since outside of fast-flowing glaciers it is very likely that vertical strain-rates are the dominant component, and in the centerline of fast-flowing glaciers it is likely that longitudinal strain rates are dominant. Given that the paper presents this model as very general to lots of ice sheet conditions, this seems to be an assumption that constrains the model. Further, since strain heating is largely treated as a free parameter in the model (through the Brinkmann number) there seems to be no reason to constrain this to just lateral strain rates.*

We thank the reviewer for noting that the solution can be in fact applied to a more general set of conditions. There is no need to further assume that the lateral strain rate is a dominant component and thus the results can be applied to any other conditions: slow-moving regions and centerline of fast-flowing glaciers or icestreams. We have relaxed this assumption so that the strain rate contribution is in fact general and applicable in any other conditions (Section 2 and Eq. 4).

- *There are a number of assumptions within the representation of heat sources and sinks that needs to be further discussed in the Discussion section – for example, the assumption that strain-rates are constant with depth when calculating the shear heating term (certainly not the case everywhere) and the use of a depth-averaged lateral advection term (given the uncertainties underlying lateral advection, I would assume that there is a possibility that there is depth variation in lateral advection). While I don't think this makes the model incorrect by any means, it does affect the results shown in the "stationary solutions" and "full solutions" sections and means that the model as presented here is not necessarily general for all possible ice sheet conditions.*

We have expanded the discussion, particularly on the assumption that strain-rates are constant with depth and the implication of a depth-averaged lateral advective term. As the reviewer points out,

it only has a slight impact on the temperature solutions and it is worthy of discussion. We agree that there are certain conditions under which the analytical solutions herein presented are not fully applicable. Particularly, in regimes where vertical shear dominates and the strain heat dissipation is concentrated near the base, a vertically-averaged contribution appears to be inaccurate. However, as already noted by Rezvanbehbahani et al. (2019), this effect is well captured by an increase in the inflow of heat from the base (i.e., an increased geothermal or frictional heat term) under conditions where most of the vertical shear is concentrated in the basal layers (Fowler, 1992).

- *I was a bit confused about the importance of the beta parameter for the results – in Figure 2, it appears that beta has little effect on the estimated ice temperature profiles (at least for the stationary solutions), but then in lines 328-329, it is stated that beta does affect temperature through the ice column. And if beta doesn't affect temperature, then I wonder why spend so much space early on describing the surface boundary condition?*

The role of the surface insulating parameter $\beta$ is important for the transitory solution. As shown in Figs. 3 and 5, $\beta$ is essential to accurately describe the transient regime of the ice temperatures. However, Figure 2 shows instead solutions at equilibrium (i.e., stationary regime), where $\beta$ is in fact less relevant compared to the remaining parameters. The subtlety thus relies on the fact that $\beta$ determines the transitory behaviour of ice temperatures while leaving nearly unchanged the solutions at equilibrium. This has been clarified in the text.

**Smaller comments**

- *The last paragraph of the introduction is a bit repetitive*

We have entirely rewritten the last paragraph.

- *Some of the variables names need to be clearly defined up front (e.g. l105, it is worth stating outright that theta is ice temperature)*

Variables have been clearly defined in the updated manuscript version (current line 111).

- *Newton's law of cooling is mentioned twice as the model behind the surface boundary condition, so it is valuable to describe the law briefly and explain why it is applicable to this situation*

We have included a brief description of the law and the conditions under which it becomes applicable (see lines 120-125). It can be summarized as follows. Newton's law of cooling describes those

boundary conditions where the heat flux across the interface is proportional to the temperature difference between the surface and the surrounding medium. It is applicable to a large variety of conditions such as a body cooling by forced convection (i.e., a fluid forced rapidly past the surface of a solid) or a a thin surface layer of a poor conductor. Moreover, Newton's law of cooling captures the two simpler boundary conditions as limit cases: (1) prescribed surface temperature and (2) no flux across an interface.

- *Lines 123-126 seem to be a restatement of the previous paragraphs*

We have deleted this additional paragraph to avoid repetition.

- *There are a lot of parameters in this paper, so it seems useful to me to eliminate parameters if they aren't strictly necessary – for example, why use $\mathcal{L}$ instead of just stating [0,L]?*

The interval $[0, L]$ was denoted by $\mathcal{L}$ solely for a more succinct notation. After non-dimensionalization, the interval becomes $[0, 1]$ and we thought that $\mathcal{L}$ (where tildes are dropped) would lighten the notation.

- *L145: S is technically a function of both the stress tensor and the strain rate tensor – one could argue that these are related but Equation 4 puts S in terms of strain rate, so it seems worth it to state the dependence on strain rate explicitly first.*

Indeed. We have stated this explicitly in the revised manuscript.

- *Equation 5: this equation uses xi before it is defined.*

We thank the reviewer for noting this. We have expressed the magnitude in terms of $z$.

- *Table 1: some of these characteristic ranges need some justification – for example, I believe that Br can be much larger than 2. Further, how do you estimate the ranges of the lateral advection parameter?*

We have included an additional table (Table 2) to summarize all the physical parameters employed to justify the dimensionless range in Table 1. Typical ice-sheet values are used as reference. To estimate the ranges of the lateral advection parameter, we explore realistic values of two phsyical magnitudes: horizontal ice velocity and longitudinal temperature gradients (along a flow line; see e.g. Dahl-Jensen, 1989).

- *Lambda isn't defined before it is used in line 178, I believe.*

We have rearranged the location of Table 1 so that the definitions appear before they are stated in the text.

- *l207-208: I didn't quite understand this statement.*

We mean that it is worth describing in detail the solution at thermal equilibrium for different combinations of the dimensionless parameter. We have updated the text for clarity.

- *l214: reduces dimensionality compared to what – other models, or the dimensional form of this model? 5 uncertain parameters still seem like a lot.*

Compared to the dimensional form of this model. Five dimensionless parameters fully determine the solution and it becomes a reasonable number given that we consider six physical processes for heat transfer: diffusion, advection (vertical and horizontal), basal inflow of heat (geothermal and frictional), strain dissipation and surface insulation. This leaves the model with one unique parameter for each physical process therein described, allowing for a straightforward interpretation of the results.

- *l218: "normalized geothermal heat flow also yields…" I had to go back and remember which parameter that corresponded to, since the Figures are only labeled by parameter. It would be useful to include the mathematical parameter symbol when you refer to them in the text. Also, isn't gamma a combination of both geothermal heat fowand basal friction?*

We have included the parameter symbols for clarity. Indeed, it refers to the combination of both geothermal heat flow and basal friction. The text has been modified accordingly.

- *There's a bit of repetition in the Discussion and conclusion sections, in which the results get recapped in both places.*

We have now re-organised the Discussion and Conclusion section to avoid repetition.

- *Figure 2: the colormaps make some of the distinctions between the lines hard to see, especially c and d. In theory, the reader can infer which lines are which but it'd be clearer to switch to a different colormap that distinguishes the lines better.*

We have improved the colour palettes to ease visualization. We thank the reviewer.

- *Is it possible to redo the x-axes for the temperature profiles (and for the timescale in Fig 4) in dimensional terms? It's hard to know the magnitudes of temperature variations/timescales we're talking about in the nondimensional terms*

Yes, we have now expressed it in terms of dimensional variables by inverting the transformation in Eq. 6. Dimensionless parameters have been consequently adjusted. We hope that this will ease the interpretation of temperature variations and timescales.

**3 Reviewer #4**

- *In this paper, the author derive solutions to the heat equation that are relevant to glaciers. The focus is on computing solutions and not solving a particular science question or using their tool to make a prediction. Although I see merit in their analysis and find that there are some interesting insights, the results are not a significant enough advance beyond what is already published to warrant publication in the Cryosphere.*

We strongly differ from the opinion of Reviewer #4. Together with a number of other reviewers of this paper, we consider that this work brings new insight to the description of time-dependent ice temperatures and further provides a set of benchmark experiments to test numerical solvers widely used in state-of-the-art ice-sheet models. To illustrate this, Reviewer #5 provides a clear statement: "*The analytical formulation of a transient ice temperature equation is certainly interesting and provides useful insight, in my opinion, in two ways – firstly, allowing for a simplified way of deriving physical insight into the physics of heat transfer in ice (as demonstrated by their analysis on equilibrium timescales) and secondly, by providing a way of benchmarking numerical solvers for heat transfer*".

From the early works of Robin (1955) and Lliboutry (1967) to the most recent advances such as Rezvanbehbahani et al. (2019), great effort has been made to expand our knowledge on how temperatures behave within the ice. Nevertheless, there is a clear gap in our understanding of the inevitable temporal evolution. The present study aims at filling this gap not only by providing an exact analytical solution of the time-dependent nature of ice temperatures, but also by providing a suite of benchmark experiments to test numerical models. The novelty of the current work is thus clear and Reviewer #4 has chosen to overlook it.

---

## Author Response (AR5)

**Author's response**
**TC-2022-97**

Daniel Moreno-Parada, Alexander Robinson, Marisa Montoya, and Jorge Alvarez-Solas.

July 19, 2024

**1 Reviewer #5**

The authors are grateful to the reviewer for their constructive comments. The current version of the manuscript has strongly benefitted from them. We now provide our answers (regular font) to the suggestions made by the reviewer (italic).

- *While the authors should organize their introduction as it best fits their vision, I would recommend beginning with an introduction to the importance and background of the ice temperature problem in glaciology, identifying the need for a new 1D transient solution, discussing the work done in other fields (e.g. what is currently lines 34-43 in the tracked-changes version of the manuscript), and ending with what the paper specifically aims to accomplish. Particularly given the focus of the journal, this would be the clearest way to set up the work presented in the rest of the paper.*

We agree with the suggested new order of the introduction and we have change it accordingly. The current introduction thus starts by stating importance and background of the ice temperature problem in glaciology. Then, the need for a new 1D transient solution is highlighted with a review of previous work done in other fields. Finally, the specific aim of the paper is presented.

- *Finally, as a small comment, I still disagree with some of the ranges presented in Table 1 – the Brinkmann number can certainly be larger than 5 (back-of-the-envelope calculations give reasonable values up to 50 and possibly higher), and if I'm understanding the authors' response correctly, the lateral advection parameter was taken primarily from Dahl-Jensen 1989, which looked at a flowline near the summit of Greenland, and I'm not sure it's clear that these are relevant magnitudes for fast-flowing glaciers in Antarctica. Even if the authors continue with these ranges, however, it would be valuable for them to include citations for all of these ranges, or an explanation (possibly as another column in the table) for how they arrived at these ranges.*

We have re-calculated the Brinkmann number by including higher strain rates from e.g., Meyer and Minchew (2013). As the editor suggests, values can be higher than 5 and Table 1 has been updated to reflect so. Moreover, following the reviewer comment, we have included an additional column in Table 2 with all references of those values that yield the non-dimensional parameter ranges. The lateral advection parameter is taken from two references: Dahl-Jensen (1989) and Funk et al. (1994). The former spans a variety of horizontal temperature gradients from the ice divide to the ablation region. The latter further describes the two-dimensional temperature fields applied along the central flowline of Jakobshavns Isbnfæ, West Greenland, and along a flowline through the adjacent ice sheet.